# Depositional controls and budget of organic carbon burial in fine-grained sediments of the North Sea – the Helgoland Mud Area as a natural laboratory

Daniel Müller[1,2], Bo Liu[1], Walter Geibert[1], Moritz Holtappels[1,3], Lasse Sander[4], Elda Miramontes[2,3], Heidi Taubner[2,3], Susann Henkel[1,3], Kai-Uwe Hinrichs[2,3], Denise Bethke[1], Ingrid Dohrmann[1], Sabine Kasten[1,2,3]

[1]Alfred Wegener Institute Helmholtz Centre for Polar and Marine Research, 27570 Bremerhaven, Germany
[2]Faculty of Geosciences, University of Bremen, 28359 Bremen, Germany
[3]MARUM - Center for Marine Environmental Sciences, University of Bremen, 28359 Bremen, Germany
[4]Alfred Wegener Institute Helmholtz Centre for Polar and Marine Research, Wadden Sea Research Station, 25992 List/Sylt, Germany

*Correspondence to*: Daniel Müller (daniel.mueller@awi.de)

**Abstract.** The burial of organic matter (OM) within fine-grained continental shelf sediments represents one of the major long-term sinks of carbon. We investigated the key factors controlling organic carbon burial in sediments of the North Sea by using the Helgoland Mud Area (HMA) as a natural test field. The HMA represents the most significant depocentre of fine-grained and organic-rich sediments in the German Bight (SE North Sea). We examined factors including sedimentation and accumulation rate, sediment mixing rate, grain size, total organic carbon (TOC) content and aerobic remineralisation rate. Highest sedimentation rates of up to ~ 4.5 mm yr$^{-1}$ and average TOC contents of 2 wt% were found in the southern part of the HMA which is under the influence of the Elbe river outflow reaching organic carbon burial efficiencies of > 65 %. Four times lower sedimentation rates and lowest TOC contents (0.7 – 1.0 wt%) were found in the shallow, eastern part of the research area, with the lowest organic carbon burial efficiencies being 30 %. High sedimentation rates are known to limit oxygen exposure time thereby enhancing OM preservation. Our data support this finding, demonstrate and confirm that sedimentation rate is the key factor determining organic carbon burial efficiency and long-term sedimentary carbon storage. In the southern part of the HMA, close to the outflow of the Elbe river, the OM being degraded is primarily of terrigenous origin, while in the central and northern part of the HMA a mixture of marine and terrigenous OM is remineralised. At the sites dominated by the degradation of marine organic matter, as found in the western and northwestern HMA, the organic carbon burial efficiency is lower and fluctuates around 55 %. The burial efficiency of OM is highest in sedimentary habitats characterised by high sedimentation rates and OM of terrigenous sources. Sediment mixing rates were highest in the northwestern HMA, where also the highest bottom trawling activity is reported. The comparison of sites similar in depositional characteristics but different in bottom trawling intensity suggests that in the area of intense bottom trawling in the northwestern HMA the sequestration of OM is reduced by around 30 %. The annual burial flux of organic carbon in the HMA amounts to an average of 22.5 g C m$^{-2}$ yr$^{-1}$. Considering the strong tidal currents in the shallow HMA, the burial flux is exceptionally high and even compares with those reported for the deeper Skagerrak and Norwegian Trough (~ 10 to 66 g C m$^{-2}$ yr$^{-1}$), which are the main depocentres for fine-grained and organic-rich sediments in the North Sea. For the entire HMA the total annual organic carbon accumulation amounts 0.011 Tg C yr$^{-1}$. These findings highlight the importance of depocentres for fine-grained sediments as important carbon sinks: while the area of the HMA represents only 0.09 % of the North Sea it stores 0.76 % of the total annual accumulated organic carbon in this shelf sea area.

## 1 Introduction

Marine sediments and sedimentary rocks represent the largest permanent sink for carbon on our planet (e.g., Berner and Berner, 2012) with coastal, estuarine and continental shelf sediments being the most important depocentres as they contain

approximately 90 % of the organic carbon in the marine sedimentary system (Hedges and Keil, 1995). The burial of organic matter (OM) in continental shelf sediments drives the storage of carbon over timescales of thousands of years and, thus, is a key natural process removing carbon from the fast cycling and heavily anthropogenically affected coastal ocean and atmosphere (Berner, 1982; Burdige, 2007). While anthropogenic carbon dioxide ($CO_2$) release continues to rise (IPCC, 2023), human activities in shelf seas, like bottom trawling, sediment dredging and dumping, as well as the construction and use of offshore infrastructures, reduce the natural carbon storage capacity of sediments by enhancing the remineralisation rates of particulate organic carbon back to $CO_2$ (Paradis et al., 2021; Clare et al., 2023).

Depending on the origin and reactivity of OM as well as water depth and settling velocity of particles and aggregates, most OM is remineralised on short time scales in the water column and only a small part is deposited on the seafloor (Suess, 1980; Berner, 1982; Middelburg et al., 1997). The long-term preservation of OM in marine sediments is controlled by post-depositional aerobic and anaerobic remineralisation processes (Froelich et al., 1979; Berner, 1980). Below the oxic zone, where oxygen is consumed by aerobic microbial respiration, other electron acceptors, namely nitrate, manganese oxides, iron oxides and sulfate, are used by microorganisms to further degrade OM within anoxic sediments (Froelich et al., 1979). Despite these different remineralisation pathways, the aerobic degradation of organic matter was shown to be the most energy-yielding and fastest process (e.g., Froelich et al., 1979; Jørgensen, 2006). Thus, post-depositional oxygen exposure is a key factor controlling OM degradation, especially of less reactive refractory OM, and hence its preservation (e.g., Hartnett et al., 1998; Burdige, 2007; Zonneveld et al., 2010; Bogus et al., 2012).

Physical and biological parameters and processes like sedimentation rate, natural sediment mixing/remobilisation by wave activity, tides and during storm events, anthropogenic activities (e.g., bottom trawling, off-shore wind farm constructions, sediment dredging and dumping) or bioturbation of the surface sediment can alter the duration of oxygen exposure and thereby influence the magnitude of the aerobic oxidation of OM (e.g., Aller, 1994; Canfield, 1994; Hartnett et al., 1998; Zonneveld et al., 2010; De Borger et al., 2021a; Daewel et al., 2022). Sedimentation rate is one of the most important factors controlling the preservation of OM in the sediment as it directly influences oxygen exposure time of surface sediments (Canfield, 1994; Jung et al., 1997; Jørgensen, 2006; de Lange et al., 2008; Zonneveld et al., 2010). Besides the impact of oxygen exposure time, the reactivity of the organic material is decisive for remineralisation kinetics. The reactivity of OM is closely related to its molecular structure and depends on the origin/source and formation pathways of OM (e.g., Hedges et al., 1988, 2000; Burdige, 2005; LaRowe et al., 2020; Freitas et al., 2021; Xu et al., 2023). While marine OM is generated locally through primary production in the euphotic zone and generally more reactive and preferably degraded both in the water column and in marine sediments, part of the terrestrial OM is already degraded during transport towards the oceans, resulting in a more refractory composition and lower bioavailability (Henrichs, 1992; Zander et al., 2020). The long-term preservation of OM in the sediment can be assessed based on the content of total organic carbon (TOC) in the sediment and also be described quantitatively as the burial efficiency of organic carbon. This is expressed as the ratio of organic carbon buried in the sediment and OM reaching the seafloor (Canfield, 1994).

Fine-grained coastal and shelf sediments have been shown to be rich in OM (e.g., Bockelmann et al., 2018; Diesing et al., 2021, 2024). In these deposits $O_2$ typically penetrates only a few millimetres into the sediments, leading to the establishment of anoxic conditions at shallow sediment depths and in this way enhancing the burial of OM (e.g., Zonneveld et al., 2010; De Borger et al., 2021b). Therefore, fine-grained coastal and shelf sediments act as an important sink for organic carbon and thus for the regulation of $CO_2$ in the ocean and atmosphere. Carbon retained on the shelf depends on the balance between supply mechanisms and removal by transport or the interplay of deposition, remineralisation and burial (e.g., Legge et al., 2020). Annually, between 16 and 40 Tg of carbon from the atmosphere is taken up into the water column of the Northwest European Shelf, which includes the North Sea. This carbon is stored in the water column, as well as in vegetated and unvegetated coastal and shelf sediments (Legge et al., 2020). The two major outputs of organic and inorganic carbon from the shelf are off-shelf transport, accounting for 60 – 100 %, and long-term burial in coastal and shelf sediments, accounting for 0 – 40% of carbon

outputs (Legge et al., 2020). In order to best protect these seafloor habitats characterized by high organic carbon contents (up to 5 wt%, e.g., Bockelmann et al., 2018) and to preserve their natural $CO_2$ storage capacity, a comprehensive understanding of the various environmental and depositional factors and their influence on long-term organic carbon burial is paramount.

So far, detailed investigations to determine the factors controlling the preservation and long-term burial efficiency of OM in fine-grained North Sea sediments are lacking. As part of the collaborative project "Anthropogenic impacts on particulate organic carbon cycling in the North Sea" (APOC), we chose the Helgoland Mud Area (HMA) as a study site as it represents the most important depocentre of fine-grained and organic-carbon-rich sediments in the German Bight of the North Sea (e.g., Figge, 1981; Diesing et al., 2021). It is known from previous studies that the area hosts a broad variety of sedimentary habitats that differ in key depositional factors, including water depth (e.g., Sievers et al., 2021), sedimentation rates (e.g., Irion et al., 1987; Baumann, 1991; Hebbeln et al., 2003; Boxberg et al., 2020), grain size (Figge, 1981; Laurer et al., 2013; Bockelmann et al., 2018; Sievers et al., 2021), and origin of organic matter (Oni et al., 2015b; Zhou et al., 2024). Based on a new extensive pore-water and solid-phase data set, we use the HMA as a natural laboratory to (1) determine the main depositional drivers controlling the burial of organic carbon for fine-grained parts of North Sea shelf sediments and beyond, (2) assess the efficiency of different sedimentary habitats as long-term natural carbon sinks, and (3) estimate the annual organic carbon accumulation in this significant depocentre in the German Bight of the North Sea.

## 2 Study area

The south eastern North Sea is a shallow shelf sea with water depths ranging from 10 to 40 m, dominated by the presence of unconsolidated sediment of primarily glacial, pro-glacial and fluvial sources (von Haugwitz et al., 1988; de Haas et al., 2002; Sievers et al., 2021). The area was submerged by the transgressive North Sea during the mid-Holocene, following the relative sea-level low-stand of the last Glacial Maximum, and marine depositional processes thus have only occurred over the last ~ 8000 years (Vink et al., 2007). The German Bight of the southern North Sea is characterised by high tidal and wave energy regimes with a tidal range of ~ 1.5 to 3 m and a mean significant wave height of around 1 m in the German Bight (Hagen et al., 2021). Seafloor processes are mostly characterised by local transport and resuspension of material in sand-rich sedimentary environments (Figge, 1981; de Haas et al., 1996; Zeiler et al., 2000; de Haas et al., 2002). These conditions limit sedimentation to only a few regions, with the most important depocentres for fine-grained sediments in the German Bight of the North Sea being the HMA together with the tidal flats of the Wadden Sea and estuaries (Figge, 1981). The HMA covers an area of approximately 500 km$^2$ (after von Haugwitz et al., 1988 and Doll, 2015). It is located southeast of the island of Helgoland at water depths between 10-30 m below mean sea level (Fig. 1a).

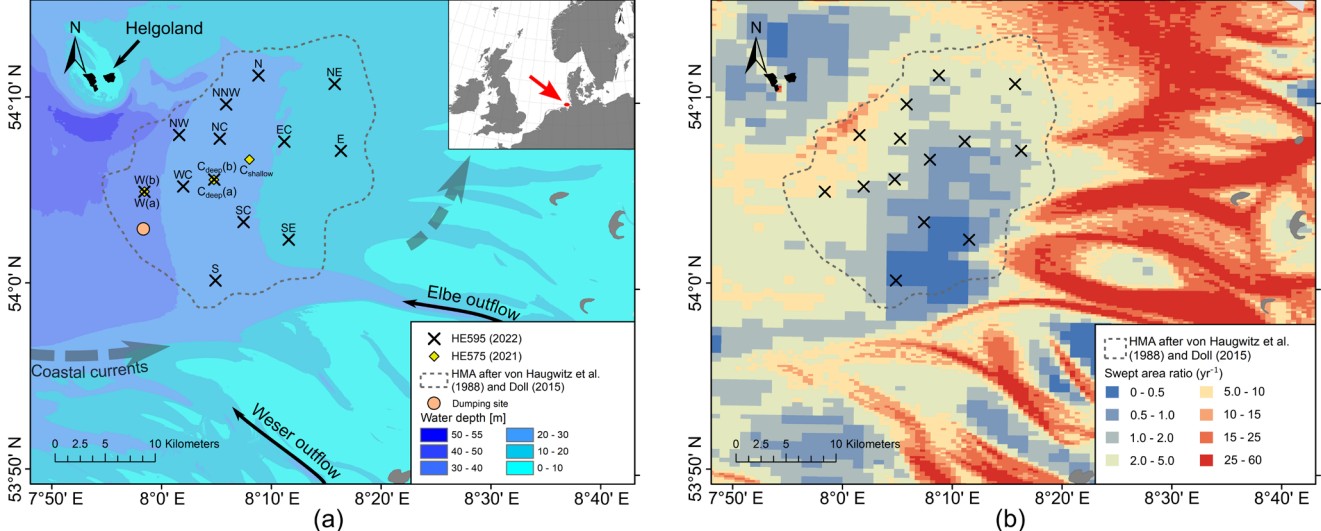

Figure 1: (a) Locations of the multi-corer (MUC) sampling during expeditions with RV *Heincke* HE575 (2021, yellow diamonds) and HE595 (2022, black crosses), areal extent of the Helgoland Mud Area (HMA, dashed line) after von Haugwitz et al. (1988) and Doll (2015), bathymetry from Sievers et al. (2021), overall coastal currents after Hertweck (1983; dashed arrows), dumping site for harbour sludge Tonne E3 (orange dot; Hamburg Port Authority, 2017) and (b) mapped bottom trawling activity as swept area ratio from Thünen Institute (2018) after Hintzen et al. (2012).

The sediments originate from suspended particulate matter (SPM) of riverine input (mainly from the Elbe and Weser rivers with a total of 1,190,000 – 1,900,000 t yr$^{-1}$; e.g., Hertweck, 1983; and references therein), primary production, local sediment redeposition, and coarser-grained layers deposited during storm events (Gadow, 1969; Puls et al., 1999). In the southeastern and eastern HMA Serna et al. (2010) showed a strong influence of the Elbe river on the sediments using stable nitrogen isotopes. In the German Bight, the spring algal bloom starts in late spring, around April, and produces organic material in the water column (e.g., Teeling et al., 2012; Amorim et al., 2024), but the benthic system lags several months behind the appearance of these phytoplankton blooms (e.g., Provoost et al., 2013). Previous estimates of sedimentation rates in the HMA range from 0.5 to 40 mm yr$^{-1}$, using a variety of methods including dating based on storm layers (Reineck et al., 1967; Gadow, 1969), coal and slag layers (Reineck, 1963), total sediment thickness (Reineck, 1969), SPM modelling (McCave, 1970), anthropogenic heavy metal content (Förstner and Reineck, 1974; Irion et al., 1987), $^{210}$Pb/$^{137}$Cs (Dominik et al., 1978; Baumann, 1991; Hebbeln et al., 2003; Serna et al., 2010) and radiocarbon dating (Hebbeln et al., 2003; Boxberg et al., 2020). As many of these methods operate on different timescales, with varying vertical resolutions, and in some cases lack precise spatial localisation in the study area, a consistent evaluation of recent sedimentation rates is impossible. While bottom trawling activity is generally low or absent in most of the HMA, the area in the northwest of the HMA is significantly affected by bottom trawling with average bottom trawling activity of 5 – 15 yr$^{-1}$ (given as swept area ratio in Fig. 1b; Eigaard et al., 2017; Hintzen et al., 2012; Thünen Institute, 2018). However, centuries of ongoing bottom trawling in the German Bight and the North Sea (de Groot, 1984) could contribute significantly to the overall SPM load and residual transport and deposition in the HMA and the shelf sea area. Further direct anthropogenic impacts on sedimentation in the study area include the dumping of sediments at site Tonne E3 at the western rim of the HMA, where harbour sludge that is regularly dredged from the Elbe estuary and the Hamburg harbour are dumped (e.g., Hamburg Port Authority, 2017).

Although this study is not directly involved in biological research, there is a comprehensive understanding of relevant infaunal community dynamics developing within the HMA (Thatje and Gerdes, 1997; Neumann et al., 2013; Shojaei et al., 2016, 2021; Wrede et al., 2017). The general grain-size distribution shows finer sediments with a mud content of more than 90 % in the western, central and southern parts of the HMA and coarser sediments towards the shallow-water eastern part of the HMA (Figge, 1981; Laurer et al., 2013; Bockelmann et al., 2018; Sievers et al., 2021). However, the depositional mechanism of the HMA is complex due to the interaction of wind-driven coastal circulation with two distinct frontal systems – a salinity front

and a tidal mixing front (Becker et al., 1992, Callies et al., 2017). The detailed mechanism of sediment transport, trapping and deposition is subject of ongoing research within the project APOC.

## 3 Materials and methods

Sediment and pore-water samples were collected during two expeditions with the RV *Heincke* (HE575, April 28 to April 30, 2021, and HE595, March 17 to April 3, 2022). While no severe winter storms were recorded in the German Bight during the 2020/2021 season before the expedition HE575 (Abromeit et al., 2021), a total of 17 winter storms occurred during the 2021/2022 season. This includes two severe storm floods and a series of storm floods, culminating in a very severe storm flood on February 19, 2022 – only four weeks prior to the RV *Heincke* expedition HE595 (Abromeit et al., 2022). This offers the possibility of studying the effects of severe winter weather on the HMA.

We have performed a detailed literature search of studies previously performed in and around the HMA and provide this compilation as a publicly accessible database (https://doi.org/10.1594/PANGAEA.968994; Müller and Kasten, 2024; for details see supplementary material). Based on this literature research, sample locations were selected and distributed over the entire HMA in order to represent all depositional environments (differences in: water depths, sedimentation rates, grain sizes, origin of OM) of this depocentre. Sampling sites were named according to their geographical location within the HMA (cf. Table 1, Fig. 1). A total of 16 stations were sampled with a multi-corer (MUC) at water depths between 11 and 31 m to investigate the undisturbed sediment surface and shallow subsurface (upper 18-36 cm). During expedition HE575 in 2021, three stations were sampled in the western and central parts of the study area. As part of expedition HE595 in 2022, 13 stations were sampled across the entire HMA (Fig. 1, Table 1).

**Table 1: Site name, station number, sampling date, coordinates, water depths and lengths of all MUC cores investigated in this study. Two different cores from the same MUC deployment were used for pore-water (PW) and solid-phase (SP) analyses resulting in slightly different core lengths.**

| Site name | Station number | Date | Latitude (N) | Longitude (E) | Water depth (m) | Core length PW (cm) | Core length SP (cm) |
|---|---|---|---|---|---|---|---|
| N | HE595_70-1 | 27.03.2022 | 54° 11.335' | 8° 08.370' | 19.0 | 20 | 26 |
| NNW | HE595_98-3 | 30.03.2022 | 54° 09.759' | 8° 05.423' | 19.1 | 18 | 22 |
| NE | HE595_67-5 | 26.03.2022 | 54° 10.923' | 8° 15.379' | 13.5 | 22 | 22 |
| NW | HE595_1-3 | 17.03.2022 | 54° 08.080' | 8° 01.146' | 26.0 | 32 | 22 |
| NC | HE595_69-2 | 27.03.2022 | 54° 07.911' | 8° 04.885' | 20.3 | 28 | 30 |
| EC | HE595_97-1 | 30.03.2022 | 54° 07.799' | 8° 10.806' | 16.1 | 28 | 26 |
| E | HE595_26-1 | 22.03.2022 | 54° 07.329' | 8° 16.005' | 11.3 | 24 | 22 |
| $C_{shallow}$ | HE575_18-1 | 29.04.2021 | 54°06.802' | 8° 07.630' | 17.5 | 30 | 28 |
| W(a) | HE575_15-1 | 28.04.2021 | 54°05.000' | 7°58.058' | 30.7 | 36 | 36 |
| W(b) | HE595_12-3 | 20.03.2022 | 54° 04.999' | 7° 58.043' | 27.3 | 32 | 28 |
| WC | HE595_6-1 | 18.03.2022 | 54° 05.316' | 8° 01.596' | 25.6 | 26 | 18 |
| $C_{deep}$(a) | HE575_14-2 | 28.04.2021 | 54°05.692' | 8°04.400' | 22.1 | 32 | 32 |
| $C_{deep}$(b) | HE595_9-1 | 19.03.2022 | 54° 05.712' | 8° 04.431' | 20.0 | 24 | 20 |
| SC | HE595_45-1 | 24.03.2022 | 54° 03.442' | 8° 07.166' | 16.5 | 32 | 30 |
| SE | HE595_41-4 | 23.03.2022 | 54° 02.526' | 8° 11.315' | 12.0 | 32 | 26 |
| S | HE595_48-3 | 25.03.2022 | 54° 00.280' | 8° 04.665' | 21.8 | 28 | 28 |

## 3.1 Pore-water and solid-phase sampling

For each MUC deployment, one core was used for pore-water sampling and one core for oxygen micro-profiling prior to solid-phase sampling. No fluffy layer was found on top of the surface sediments, indicating that there was no recent transfer of fresh phytoplankton biomass to the seabed at the study sites. Both pore water and sediment samples were taken at 1-cm-intervals in the top 10 cm and every two centimetres below. The extraction of pore water from the MUC cores was performed according to Seeberg-Elverfeldt et al. (2005) using Rhizon samplers with an average pore size of 0.1 µm. The first millilitre of extracted pore water was discarded to prevent oxidation of the pore-water samples. For the determination of dissolved inorganic carbon (DIC) and its stable carbon isotopic composition ($\delta^{13}$C-DIC), 2 ml of pore water were mixed with 10 µl saturated $HgCl_2$ solution to prevent biological reactions and stored at 4°C until onshore analysis. Sediment samples used to determine porosity, TOC content and $^{210}$Pb, $^{226}$Ra and $^{137}$Cs activities were filled in Whirl-Pak® bags and stored at 4°C until further processing.

## 3.2 Solid phase analyses

### 3.2.1 Grain-size analysis and porosity

Sediment samples were frozen prior to freeze-drying and both, water content and porosity, were calculated by the mass loss, assuming a sediment grain density (quartz) of 2.65 g cm$^{-3}$ (e.g., Anderson and Schreiber, 1965). Eight different sites (NW, EC, E, $C_{shallow}$, W, $C_{deep}$, SC and SE) were selected to determine medium grain size $D_{50}$ and mud content (grain size fraction < 63 µm) in order to obtain two transects, one from north to south and one from west to east, intersecting at site $C_{deep}$ (Fig. 1a). Analyses were performed using a laser-diffraction particle size analyser (CILAS 1180) with a range of 0.04 to 2500 µm. Carbonates and organic matter were removed prior to analysis by a consecutive treatment with 30 % acetic acid and 12 %

hydrogen peroxide. Statistical evaluations were performed using GRADISTAT (Version 9.1) for unconsolidated sediments, developed by Blott and Pye (2001).

### 3.2.2 Radiometric analyses, age model and sediment mixing model

Radiometric analyses were carried out at 11 of the 14 sites. Only two sites in close geographical proximity to other sites (sites: NC and WC, Fig. 1a) and site NNW, where no active sedimentation takes place (von Haugwitz et al., 1988), were not analysed. Samples from eight sites were measured by gamma spectrometry to determine the activity of $^{210}$Pb and additionally $^{226}$Ra and the independent time marker $^{137}$Cs. Samples from additional three sites were measured for $^{210}$Pb by alpha spectrometry to further increase data density.

For $^{210}$Pb$_{xs}$ analysis using gamma spectrometry, MUC cores from sites NW, EC, E, C$_{shallow}$, W(a), C$_{deep}$(a), SC and SE were analysed. The activity measurements of the radioisotopes $^{210}$Pb, $^{226}$Ra and $^{137}$Cs were performed on a planar High Purity Germanium gamma detector (Canberra). Freeze-dried and homogenized sediments were weighed in petri dishes, sealed gas-tight to prevent $^{222}$Rn loss and stored for a minimum of three weeks to guarantee equilibrium of the relevant $^{226}$Ra daughters. Total $^{210}$Pb ($^{210}$Pb$_{tot}$) was detected at 46 keV and $^{137}$Cs at 661 keV. The unsupported, airborne "excess" $^{210}$Pb ($^{210}$Pb$_{xs}$) was calculated by subtracting "supported" $^{210}$Pb ($^{210}$Pb$_{supp}$), which is produced in the sediment through decay within the uranium and thorium decay series, from $^{210}$Pb$_{total}$. $^{210}$Pb$_{supp}$ was determined through measurement of other daughters of $^{226}$Ra: $^{214}$Pb at 295 keV, $^{214}$Pb at 352 keV and $^{214}$Bi at 609 keV. Samples were measured for a maximum of three days or 1000 net counts of $^{210}$Pb at 46 keV. Activities were corrected for the detector efficiency using the reference material IAEA-385 (Pham et al., 2008) with similar geometry and activities of $^{210}$Pb$_{xs}$ and $^{137}$Cs were decay corrected for the time between sampling and measurement.

The generation of the $^{210}$Pb data set was speeded up by using alpha spectrometry in parallel to the gamma detector for the three sites N, NE and S. Here, total acid digestions were performed according to the protocol from Kretschmer et al. (2010) using the MARS Xpress microwave system (CEM). 0.035 ml of a 5.6 dpm g$^{-1}$ $^{209}$Po spike was added to approximately 100 mg of freeze-dried, homogenised sediment. The sediments were digested in a sub-boiling mixture of 65 % distilled HNO$_3$ (3 ml), 32 % distilled HCl (2 ml) and 40 % Suprapur® HF (hydrofluoric acid; 0.5 ml) at ~ 230 °C. The solutions were fumed off to dryness and the residue re-dissolved under pressure in 1 M HNO$_3$ (5 ml) at ~ 200 °C and filled up to 20 ml with 1.5 M HNO$_3$. The polonium plating followed the protocol from Grasshoff et al. (1999): 0.05 ml of a FeCl$_3$ solution (50 mg ml$^{-1}$ Fe$^{3+}$) was added to the samples and 25 % NH$_3$ was added until iron precipitates formed (pH ~ 8-8.5). The suspensions were centrifuged and the supernatants were decanted. The residues were then dissolved in 32 % sub-boiling distilled HCl (0.1 ml), transferred into Teflon beakers with 0.02 M HCl and filled up to ~ 40 ml. The beakers were placed on a magnetic stirring hot plate in a sand bath at 80°C for auto-deposition of polonium. Ascorbic acid was added until the yellow colour (Fe) disappeared. Ethanol-cleaned silver plates were placed in the solutions for at least 4 hours at 80 °C. $^{210}$Pb was indirectly determined via its granddaughter $^{210}$Po, an alpha emitter, using silicon surface-barrier detectors (EG&G ORTEC). Samples were measured for at least 800 counts for $^{209}$Po and $^{210}$Po or three weeks. Supported $^{210}$Pb was approximated from $^{226}$Ra measured using gamma spectrometry in four duplicate samples for each core to calculate $^{210}$Pb$_{xs}$. The activity of $^{210}$Pb$_{xs}$ was decay corrected for the time between sampling and measurement. The reproducibility of the $^{210}$Pb$_{xs}$ measurements between gamma and alpha spectrometry was evaluated based on four samples from each of the three cores analysed by alpha spectrometry. All intercomparison measurements are in agreement within one standard deviation.

To calculate the sediment age and the sedimentation rates the constant rate of supply model (CRS; Appleby and Oldfield, 1978) was applied according to Sanchez-Cabeza and Ruiz-Fernández (2012) and the respective uncertainties were calculated, including the effects of error propagation. The CRS age model was chosen based on the quantification of the impact of sediment mixing (altered $^{210}$Pb$_{xs}$ profiles) on sedimentation rates/MAR calculations by Arias-Ortiz et al. (2018). They showed that the calculated sedimentation rate using the constant flux, constant sedimentation (CFCS) model deviated by 20 to 95 % in a

sediment mixing scenario, while the CRS model - using the inventory to calculate the rates - deviated by only 2 to 5 %. Sediment mixing rates were computed by applying the regeneration model from Gardner et al. (1987) to model steady-state sedimentation and sediment mixing (bioturbation and physical mixing).

### 3.2.3 Total organic carbon contents

The TOC contents of all samples were analysed using a Leco CS744. To determine TOC contents, samples were decalcified with 12.5 % HCl to remove inorganic carbon and dried on a hot plate before the measurements. Approximately 100 mg of freeze-dried, ground and homogenised sediment were weighed into a ceramic cup and combusted in a stream of oxygen by a high-frequency induction furnace. Carbon is measured as $CO_2$ in a non-dispersive infrared cell by absorbing a specific wavelength of infrared energy. Based on replicate analyses of two different standards, the precision (relative standard deviation in percentage) was 1.23 % for the standard of 0.99 wt% (n=15), and 1.34 % for the standard 5.00 wt% (n=7).

## 3.3 Pore-water analyses

### 3.3.1 Concentrations and stable carbon isotopic composition of dissolved inorganic carbon (DIC)

DIC concentrations were measured using a QuAAtro Continuous Segmented Flow Analyzer (Seal Analytical) for the concentration range of 0 to 4 mmol $l^{-1}$. $\delta^{13}C$-DIC values were determined for every second pore-water sample from expedition HE595 and measured by isotope ratio infrared spectrometry (IRIS; Thermo Scientific Delta Ray IRIS with URI connect and Cetac ASX-7100 Autosampler). A modified protocol for IRIS after the method of Torres et al. (2005) was applied. 12 ml Exetainer® vials (Labco) containing 100 µl of phosphoric acid (45 %) were flushed for 3 minutes with $CO_2$-free synthetic air using the Delta Ray system. A syringe was used to inject 600-1000 µl of pore water into a prepared vial and the vial was kept at room temperature for the conversion of DIC to $CO_2$. After 10 hours of equilibration, the carbon isotope composition of the released $CO_2$ in the headspace was analysed against a $CO_2$ reference gas using the Delta Ray system.

In order to estimate the source of the degraded organic material, a Miller-Tans plot analysis was performed, displaying $\delta^{13}C$-DIC × [DIC] vs. DIC concentrations. From this plot, the stable carbon isotopic composition of the degraded OM ($\delta^{13}C_{degraed}$ OM) can be traced by the slope of the linear fit (Miller and Tans, 2003; Wu et al., 2018). The Miller-Tans plot analysis was performed on the pore-water data of the 13 MUC cores from expedition HE595 for which both concentrations and $\delta^{13}C$ of DIC are available.

### 3.3.2 Oxygen profiles and aerobic remineralisation rates

Three to five oxygen micro-profiles were measured at each site during expedition HE595 with a resolution of 100 µm using micro-optodes with a tip diameter of 50 µm (OXR50, High Speed, Pyroscience). Oxygen micro-optodes were calibrated at 100 % air saturation (air-bubbled seawater) and 0 % air saturation (by adding sodium dithionite) at in-situ temperatures and positioned with a motorised micromanipulator (MU1, Pyroscience). The sediment core temperature was recorded by a thermistor (Pyroscience) immersed in the overlying water. Profile data (oxygen concentration at depth) were recorded using a 4-channel FireSting oxygen meter (FSO2-4, Pyroscience) and processed using the Pyroscience 'Profix' software.

The respective oxygen consumption rates were calculated using the diagenetic model of Berg et al. (1998) following the R script of van de Velde et al. (2022), where the model was inversely fitted to the measured pore-water oxygen profiles. Oxygen consumption rates were converted to aerobic remineralisation rates with a stoichiometry of $C_{org}:O_2$ being 1:1. This was then applied to all profiles at each individual site and average rates were calculated. This approach does not only assess and consider aerobic respiration but also includes the re-oxidation of reduced metabolic products originating from anaerobic mineralisation pathways as has been shown by e.g. Glud (2008).

### 3.4 Organic carbon burial efficiency

The organic carbon burial efficiency (OC BE) is defined as the rate of organic carbon buried at depth (burial flux) divided by the rate of organic carbon reaching the sediment-water interface (input flux) (Eq. 1) (Burdige, 2007; van de Velde et al., 2023). Here, the organic carbon input flux ($J_{OC,in}$) is the sum of the burial flux (determined via sediment accumulation rate and TOC content, $J_{OC,bur}$) and the integrated aerobic remineralisation rate ($J_{OC,min}$):

$$OC\ BE\ (\%) = \frac{J_{OC,bur}}{J_{OC,in}} * 100 = \frac{J_{OC,bur}}{J_{OC,bur} + J_{OC,min}} * 100 \tag{1}$$

### 3.5 Statistical analyses

In order to assess the different factors influencing the preservation of organic carbon and the efficiency of organic carbon burial in the HMA sediments, TOC contents and OC BE were analysed using the Pearson correlation coefficient. The respective p-values were calculated. Prior to the tests, the distribution of the residuals was tested using a Kolmogorov-Smirnov test to ensure data normality. Average values for TOC contents, porosities and grain size compositions were calculated above the depth of the sediment layer formed in 1960, derived from the age model. The year 1960 marks the cut-off date because it is the beginning of the continuously increasing global trade of goods resulting in enhanced sediment management and deepening of harbours and estuaries (Levinson, 2016; Baur et al., 2021). Moreover, all of our cores cover the past 60 years. Interpolated maps were generated using the ArcMaps' (software version 10.8.1) inverse distance weighted interpolation tool, which uses the measurements around the prediction location to predict a value for each unsampled location.

## 4 Results

As part of this study, samples were retrieved at 14 sites in the HMA. Table 2 shows which parameters were analysed at each site and are presented in the results section below. Figure S1 shows a conceptual diagram of the work performed in the framework of this study, including pore-water and solid-phase measurements, their contribution to the calculations at different hierarchies, as well as the products.

**Table 2: Overview of all parameters determined at the respective site in the HMA.**

| Site name | Porosity | Grain size | Radiometric analyses | TOC | DIC | $\delta^{13}$C-DIC | Oxygen |
|---|---|---|---|---|---|---|---|
| N | X | | X | X | X | X | X |
| NNW | X | | | X | X | X | X |
| NE | X | | X | X | X | X | X |
| NW | X | X | X | X | X | X | X |
| NC | X | | | X | X | X | X |
| EC | X | X | X | X | X | X | X |
| E | X | X | X | X | X | X | X |
| C$_{shallow}$ | X | X | X | X | X | | |
| W | X | X | X | X | X | X | X |
| WC | X | | | X | X | X | X |
| C$_{deep}$ | X | X | X | X | X | X | X |
| SC | X | X | X | X | X | X | X |
| SE | X | X | X | X | X | X | X |
| S | X | | X | X | X | X | X |

### 4.1 Compilation of previous studies performed in the area of the HMA

In order to compare our results to previously available data, we have screened and compiled data from available studies on sediments carried out in and around the HMA since the mid-20[th] century. We make this compiled data set accessible for

geographic information systems via the PANGAEA repository (https://doi.org/10.1594/PANGAEA.968994; Müller and Kasten, 2024; for details see supplementary material). This includes authors, titles, sample locations and corresponding parameters of 31 studies and 177 respective sampling locations.

## 4.2 Grain size and porosity

Across the HMA, we find low $D_{50}$ values in the deeper western and southern area (sites: W, $C_{deep}$, SC, SE; 17 to 40 µm) with higher mud contents of 66 to 91 %. In the shallower northern and eastern HMA (sites: NW, EC, E, $C_{shallow}$) slightly coarser sediments ($D_{50}$: 50 to 59 µm) with lower mud contents of 53 to 62 % are present (Fig. 2, Table 3).

Sediment porosities range from 0.37 to 0.80 over the research area (Fig. 2, Table 3). The lowest porosity of 0.37 is found at site NNW. In the northern and eastern HMA (sites: N, NE, NW, NC, EC, E, $C_{shallow}$) porosities range between 0.54 and 0.61. Sediments with the highest porosities are located in the western and southern HMA (sites: W, WC, $C_{deep}$, SC, SE, S) ranging from 0.61 to 0.80.

**Table 3: Aerobic remineralisation rates, porosities, grain sizes D$_{50}$, mud contents and sediment mixing rates.**

| Site name | Aerobic remineralisation rate (g C m$^{-2}$ yr$^{-1}$) | Porosity (-) | Grain size $D_{50}$ (µm) | Mud content (%) | Sediment mixing rate (cm$^3$ cm$^{-2}$ yr$^{-1}$) |
|---|---|---|---|---|---|
| N | 11.0 | 0.54 | | | 0.60 |
| NNW | 1.3 | 0.37 | | | |
| NE | 14.9 | 0.61 | | | 0.04 |
| NW | 18.4 | 0.57 | 58 | 55 | 8.35 |
| NC | 7.4 | 0.56 | | | |
| EC | 10.1 | 0.56 | 50 | 62 | 0.06 |
| E | 15.3 | 0.60 | 59 | 53 | 0.56 |
| $C_{shallow}$ | | 0.60 | 50 | 60 | 0.04 |
| W | 24.5 | 0.66 | 26 | 81 | 5.02 |
| WC | 17.5 | 0.62 | | | |
| $C_{deep}$ | 21.5 | 0.62 | 30 | 75 | 4.94 |
| SC | 25.8 | 0.80 | 17 | 92 | 3.15 |
| SE | 16.2 | 0.70 | 40 | 66 | c |
| S | 17.5 | 0.61 | | | 0.53 |

c: Here, the sediment mixing model was not able to reproduce the $^{210}Pb_{xs}$ profile.

## 4.3 Age model, sedimentation rates and sediment accumulation rates

Of the 11 MUC sites analysed for $^{210}Pb_{xs}$, the sediments retrieved at sites $C_{shallow}$ and EC in the centre and east of the centre show the lowest $^{210}Pb_{xs}$ activities at the top of the core with $9.4 \pm 2.1$ Bq kg$^{-1}$ and $8.3 \pm 2.1$ Bq kg$^{-1}$, respectively. At these stations, $^{137}Cs$ is only detectable down to a depth of 3 cm (Fig. 2). The other cores show significantly higher $^{210}Pb_{xs}$ activities in the uppermost sediments with highest values of $76.9 \pm 6.2$ Bq kg$^{-1}$ at site SE. The $^{210}Pb_{xs}$ activity decreases with depth at all sites until the background activity of $^{210}Pb_{supp}$ is reached at different depths for each station. The profiles show a scattered pattern of the decreasing $^{210}Pb_{xs}$ activity. In cores with higher $^{210}Pb_{xs}$ activities compared to the cores with lower activities, $^{137}Cs$ can be detected to greater depths up to a maximum depth that extends over the entire core at site SE (Fig. 2).

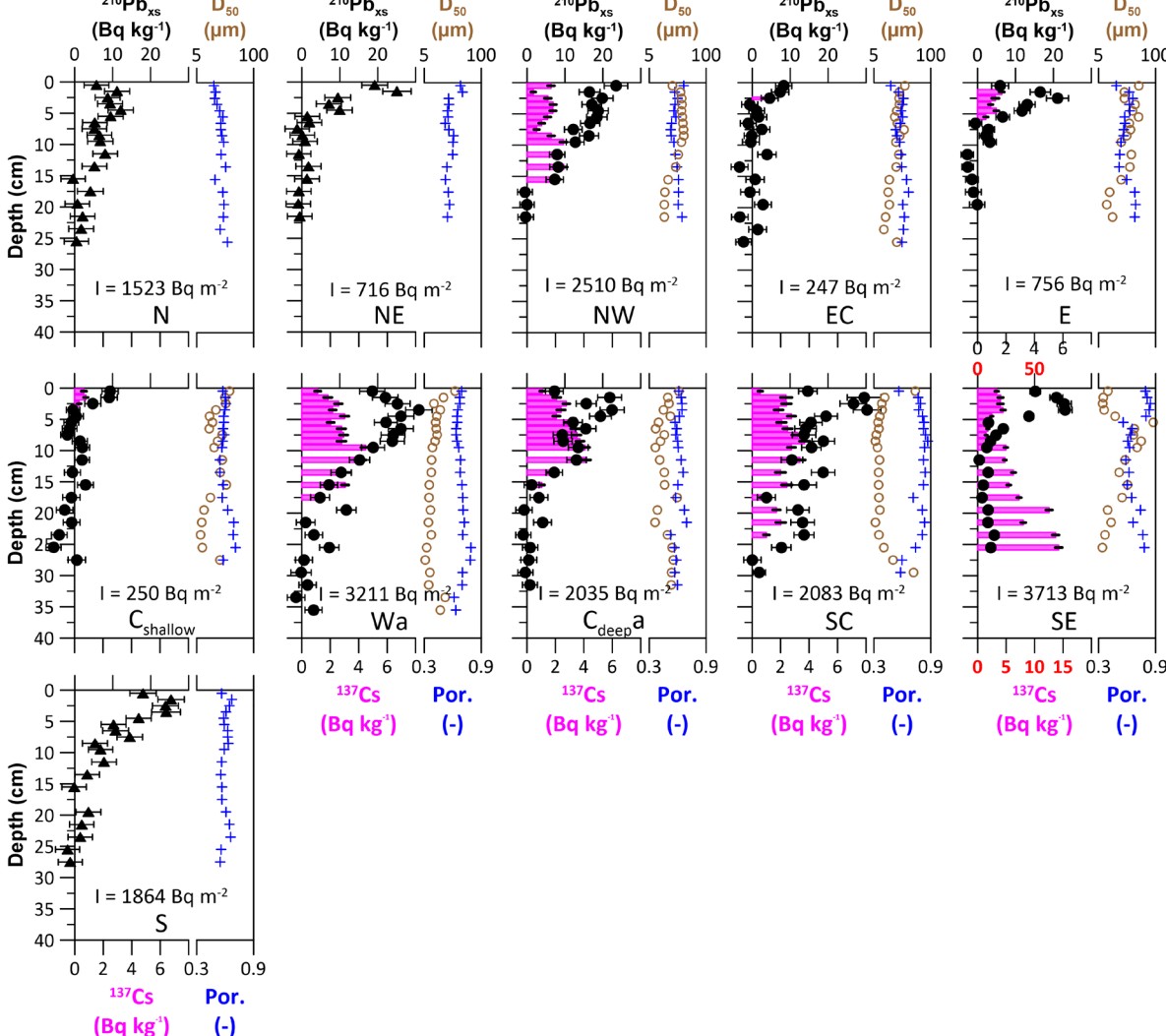

**Figure 2:** $^{210}Pb_{xs}$ activity (black),$^{137}Cs$ activity (pink), porosity (blue) and grain size (brown). The $^{210}Pb_{xs}$ profiles at sites N, NE and S were measured using alpha spectrometry (black triangles), in which $^{137}Cs$ cannot be detected due to the method used, while the others were measured using gamma spectrometry (black dots). Note the different scales at sites SE. $^{210}Pb_{xs}$ inventories (I) are shown in the respective plot area. See Table 1 for site abbreviations.

Figure 3 shows the sediment ages and corresponding sedimentation rates for the HMA, excluding non-accumulation site NNW. The sediment age of a sediment layer represents the time since the respective layer ceased to receive airborne $^{210}Pb_{xs}$, beginning at 0 cm depth, where the CRS age model is defined as 0 yrs. Due to the overall low $^{210}Pb_{xs}$ activities at sites $C_{shallow}$ and EC, sediment ages could only be calculated for the uppermost 2 cm. These stations have the overall highest sediment ages in the upper core, dating 37 and 41 yrs at 2 cm depth, respectively, and the lowest sedimentation rates of about 0.5 mm yr$^{-1}$. Sedimentation rates at most of the other sites vary around 2 mm yr$^{-1}$ with elevated values of up to 7.5 mm yr$^{-1}$ in the uppermost part of the cores at sites NW, E, W and $C_{deep}$. Subsurface maxima, e.g., at 19.5 and 25.5 cm at site W (Fig. 2), are most likely caused by storm events (de Haas et al., 1997; van der Zee et al., 2003). However, their influence on sedimentation rate calculations is only ~ 3 % and can be neglected.

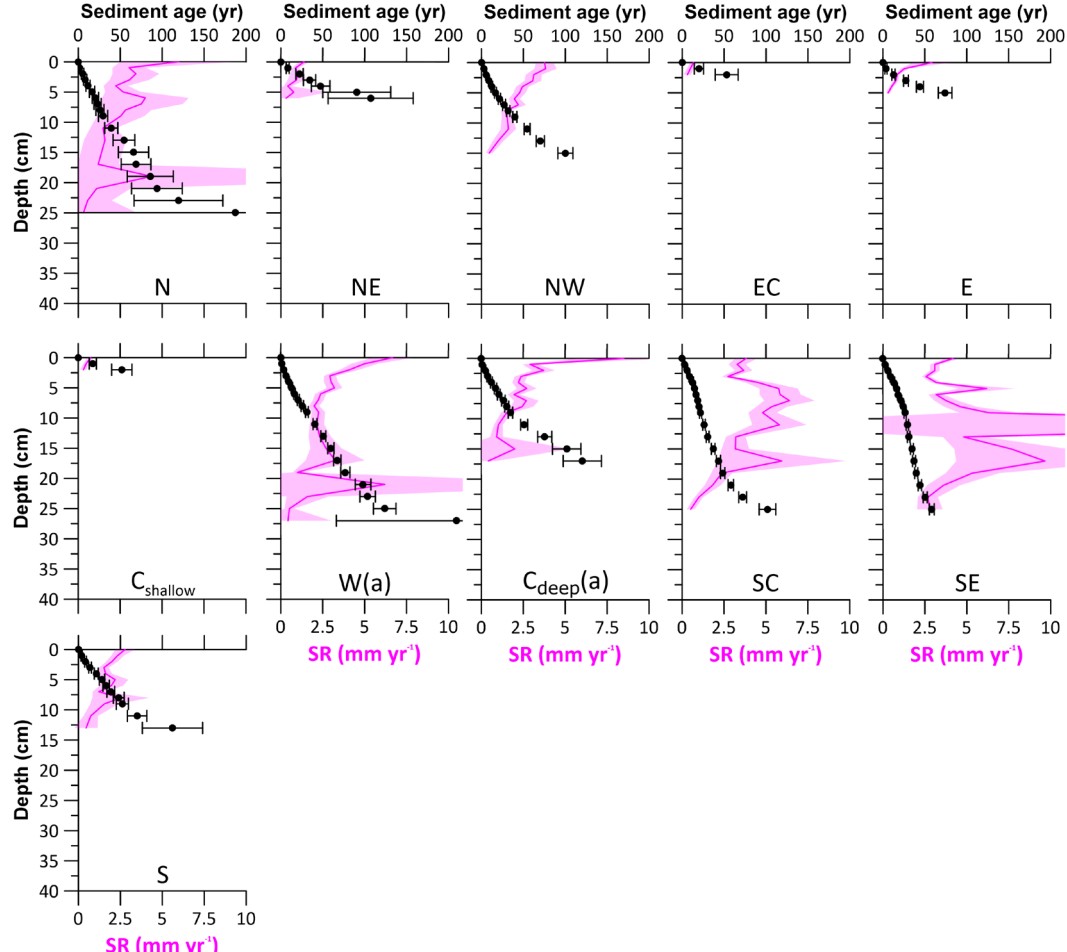

**Figure 3: Sediment ages (black dots) and sedimentation rates (SR, pink line). See Table 1 for site abbreviations.**

The distribution of sedimentation rates across the HMA shows low sedimentation rates from 0.5 to 1.3 mm yr$^{-1}$ in the eastern part (sites: NE, EC, E, C$_{shallow}$), intermediate sedimentation rates around 2.0 to 3.2 mm yr$^{-1}$ in the deeper western and central parts (sites: N, NW, W, C$_{deep}$, S) while the highest rates of 4.3 and 4.5 mm yr$^{-1}$ were found at the southern sites SC and SE.

The pattern of respective sediment mass accumulation rates (MAR) is similar to the pattern of sedimentation rates across the HMA. They range from 550 to 600 g m$^{-2}$ yr$^{-1}$ for the low sedimentation rate sites EC and C$_{shallow}$, 2000 to 3000 g m$^{-2}$ yr$^{-1}$ for sites N, NW, W, C$_{deep}$, SC and S, with only site SE showing a higher MAR of 3815 g m$^{-2}$ yr$^{-1}$ (Table 4).

**Table 4: Total organic carbon (TOC) content, $\delta^{13}C$ of degraded OM ($\delta^{13}C_{degraded}$ OM), sedimentation rate (SR), sediment mass accumulation rate (MAR), total organic carbon accumulation rate (TOCAR) and organic carbon burial efficiency (OC BE): Average values for every site used for the interpolations.**

| Site name | TOC (wt%) | $\delta^{13}C_{degraded}$ OM (‰) | SR (mm yr$^{-1}$) | MAR (g m$^{-2}$ yr$^{-1}$) | TOCAR (g C m$^{-2}$ yr$^{-1}$) | OC BE (%) |
|---|---|---|---|---|---|---|
| N | 0.64 | -20.7 | 2.9 ± 0.5 | 2908 | 18.6 | 63 |
| NNW | 0.16 | -9.0[d] | | | | |
| NE | 0.95 | -3.6[d] | 0.9 ± 0.2 | 920 | 8.7 | 37 |
| NW | 0.75 | -20.3 | 2.5 ± 0.1 | 2761 | 20.7 | 53 |
| NC | 0.75 | -20.8 | | | | 63 |
| EC | 0.70 | -17.0 | 0.5 ± 0.1 | 604 | 4.3 | 30 |
| E | 0.80 | -2.7[d] | 1.3 ± 0.2 | 1455 | 11.7 | 43 |
| $C_{shallow}$ | 0.73 | | 0.5 ± 0.1 | 552 | 4.0 | |
| W | 1.07 | -16.6 | 3.2 ± 0.2 | 2775 | 29.7 | 55 |
| WC | 1.03 | -17.7 | | | | 60 |
| $C_{deep}$ | 0.99 | -27.9 | 2.4 ± 0.3 | 2409 | 24.0 | 53 |
| SC | 2.12 | -9.2[d] | 4.3 ± 0.3 | 2275 | 48.3 | 65 |
| SE | 1.50 | -4.7[d] | 4.5 ± 0.4 | 3815 | 57.3 | 78 |
| S | 0.98 | -25.2 | 2.0 ± 0.2 | 1968 | 19.4 | 52 |

[d]DIC and $\delta^{13}C$-DIC profiles and the results from the Miller-Tans plots indicate pore-water mixing with bottom water.

### 4.4 Sediment mixing rates

Sediment mixing rates were modelled for all sites where $^{210}Pb_{xs}$ measurements were available. Generally, the model was able to reproduce the shape of the $^{210}Pb_{xs}$ activity over depth, except for the data set from site SE, where the residual sum of squares was ~ 30 times higher compared to the average residual sum of squares of the other model results. As no steady-state sediment mixing rate can be calculated for site SE, no mixing rate is given below (Table 3).

We find low sediment mixing rates in the shallow part of the HMA (sites: N, NE, EC, E, $C_{shallow}$) and at site S in the deeper HMA, where the modelled rates range from 0.04 to 0.56 cm$^3$ cm$^{-2}$ yr$^{-1}$ and higher mixing rates in the deeper western HMA (sites: NW, W, $C_{deep}$, SC; 3.15 to 8.35 cm$^3$ cm$^{-2}$ yr$^{-1}$). The highest sediment mixing rate of 8.35 cm$^3$ cm$^{-2}$ yr$^{-1}$ was modelled for site NW (Table 3).

### 4.5 Total organic carbon contents

The TOC contents of the HMA sediments vary both geographically and with core depth in a range from 0.1 to 3.0 wt%. As a general pattern, lowest TOC contents are found in the uppermost centimetres of the sediments and then gradually increase downward (over 2 to 3 cm) to a rather constant value (Fig. 4). For most of the cores, this stable TOC content is around 1 wt% or slightly less and shows only small variations with depth. Exceptions are found in sediments from the northern and central part of the HMA at sites NW, NE, NC and $C_{deep}$, where the surface layer shows slightly higher TOC contents, followed by a downward decrease and a constant or slight scattering around a constant TOC content below. The spatial distribution of the averaged TOC contents over the last 60 years shows the highest TOC contents of 1.5 and 2.1 wt% in the southern HMA at sites SE and SC respectively. While the other sites are found with TOC contents around 0.9 wt%, site NNW shows an exceptionally low TOC content of 0.2 wt%.

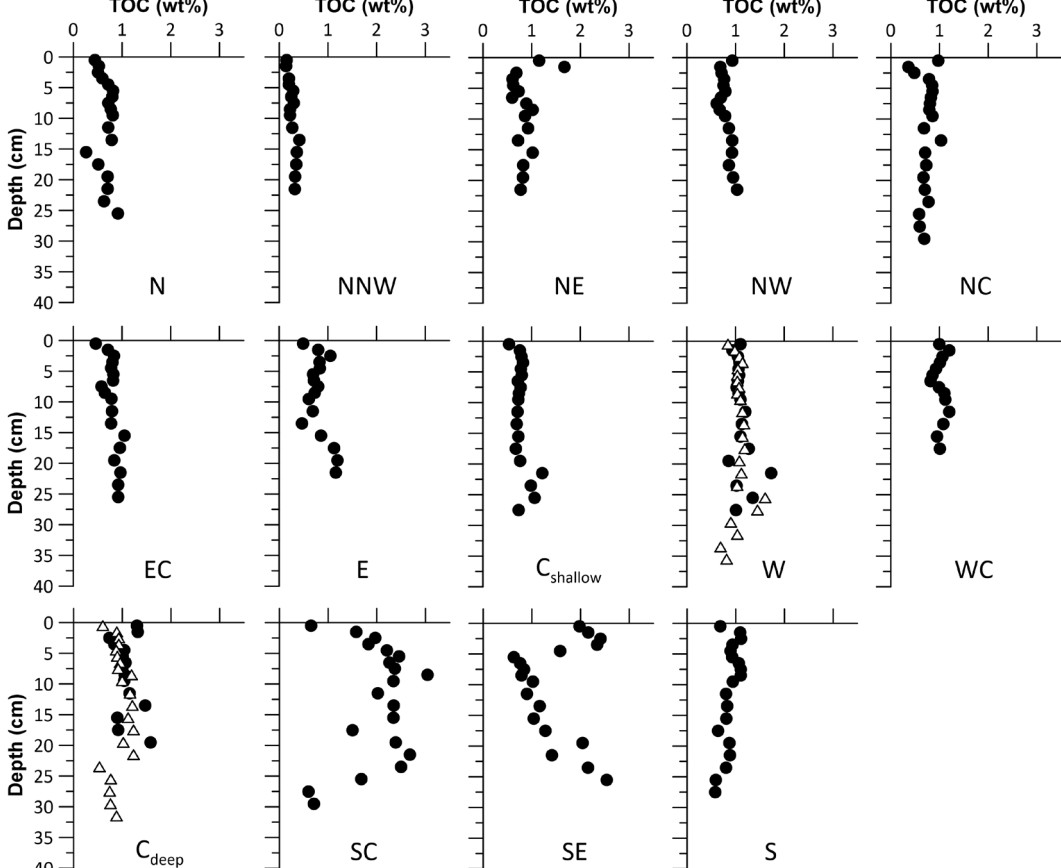

**Figure 4: Total organic carbon (TOC) contents of sediments derived during expedition HE595 (black dots). Please, note the two additional profiles from sampling in 2021 (expedition HE575, white triangles) highlighting the precision of the analysis and consistency of the sampled sites. See Table 1 for site abbreviations.**

## 4.6 Dissolved inorganic carbon and its isotopic composition

Pore-water DIC concentrations increase with depth at all sites (Fig. 5). Overall highest DIC concentrations of $\sim$ 40 mmol l$^{-1}$ are found at sites SC and SE. In contrast to the gradual increase in DIC concentrations with depth at most of the stations, the profiles from the shallow stations in the eastern HMA (sites E, C$_{shallow}$, SC, and SE) show constant, low DIC concentrations in the uppermost 5 to 9 cm of the sediments with a steep increase in DIC concentrations below. $\delta^{13}$C-DIC values generally show a mirrored pattern compared to the DIC profiles, with decreasing $\delta^{13}$C-DIC values with depth in a range of -3.4 ‰ and -24.2 ‰. In contrast to this trend, the profiles at sites SE and NE show no clear trend in $\delta^{13}$C-DIC and no decreasing $\delta^{13}$C-DIC values with depth (Fig. 5).

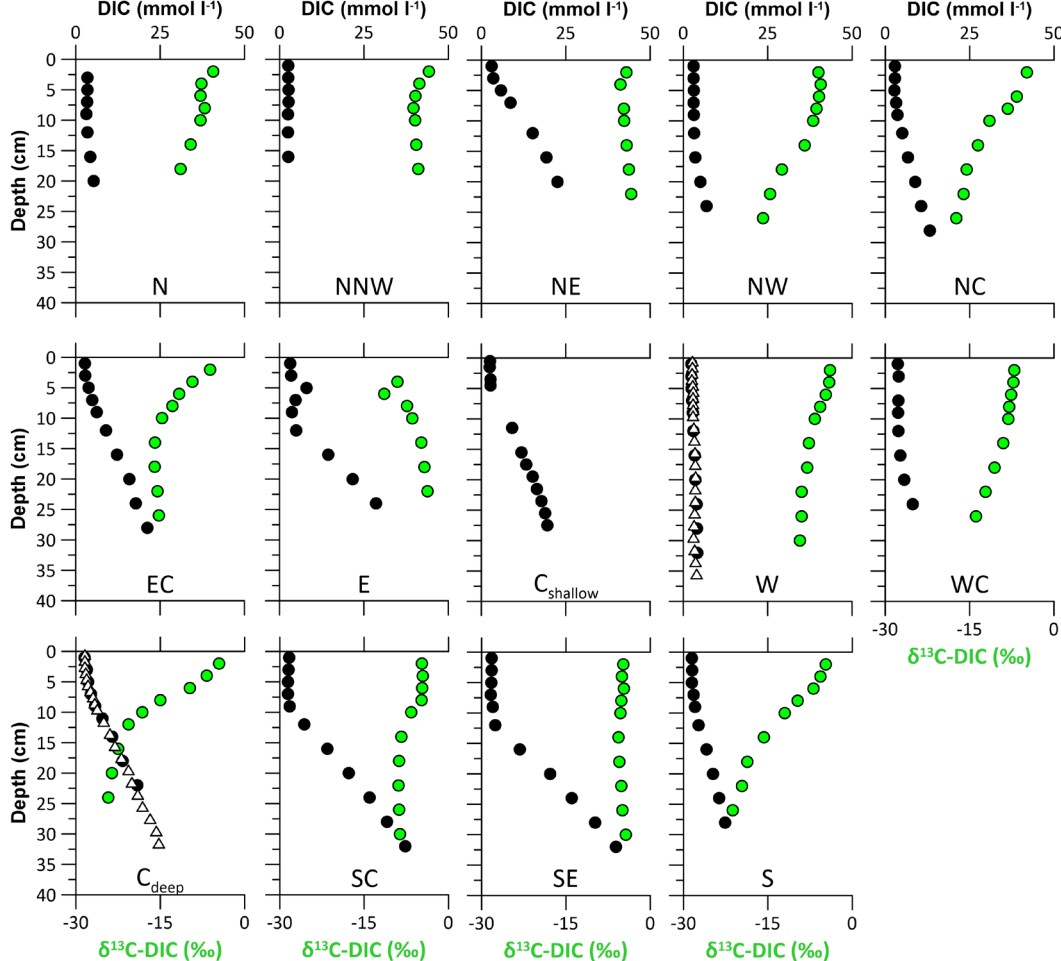

**Figure 5: Pore-water dissolved inorganic carbon concentrations from expedition HE595 in 2022 (black dots) and its stable carbon isotopic composition δ¹³C-DIC (green dots). Note the two additional DIC profiles at sites W and Cdeep from sampling in 2021 (expedition HE575, white triangles), highlighting the precision of the analysis and consistency of the sampled sites. See Table 1 for site abbreviations.**

### 4.6.1 Stable carbon isotopic composition of the degraded OM

The Miller-Tans plots are shown in Fig. 6. The slope m of the fitted linear regressions of δ¹³C-DIC × [DIC] vs. DIC concentrations displays the stable carbon isotopic composition of the degraded OM ($\delta^{13}C_{degraded}$ OM) which is added as DIC to the pore-water pool. A wide range of values was calculated for $\delta^{13}C_{degraded}$ OM in the HMA from -2.6 ‰ to -28 ‰. At sites NNW and E, where the values do not exhibit a linear trend, and in the shallow eastern HMA (sites NNW, NE, E, SC, and SE) $\delta^{13}C_{degraded}$ OM values are close to 0 ‰.

The sites with the lowest $\delta^{13}C_{degraded}$ OM values are located in the south and centre of the HMA (sites: S and $C_{deep}$) with values of -25.2 and -27.9 ‰. In the northern HMA, the values are slightly higher around -20.5 ‰ at sites NC, NW and N. Sites W and WC in the western HMA and for site EC higher $\delta^{13}C_{degraded}$ OM values of around -17 ‰ were calculated (Fig. 6).

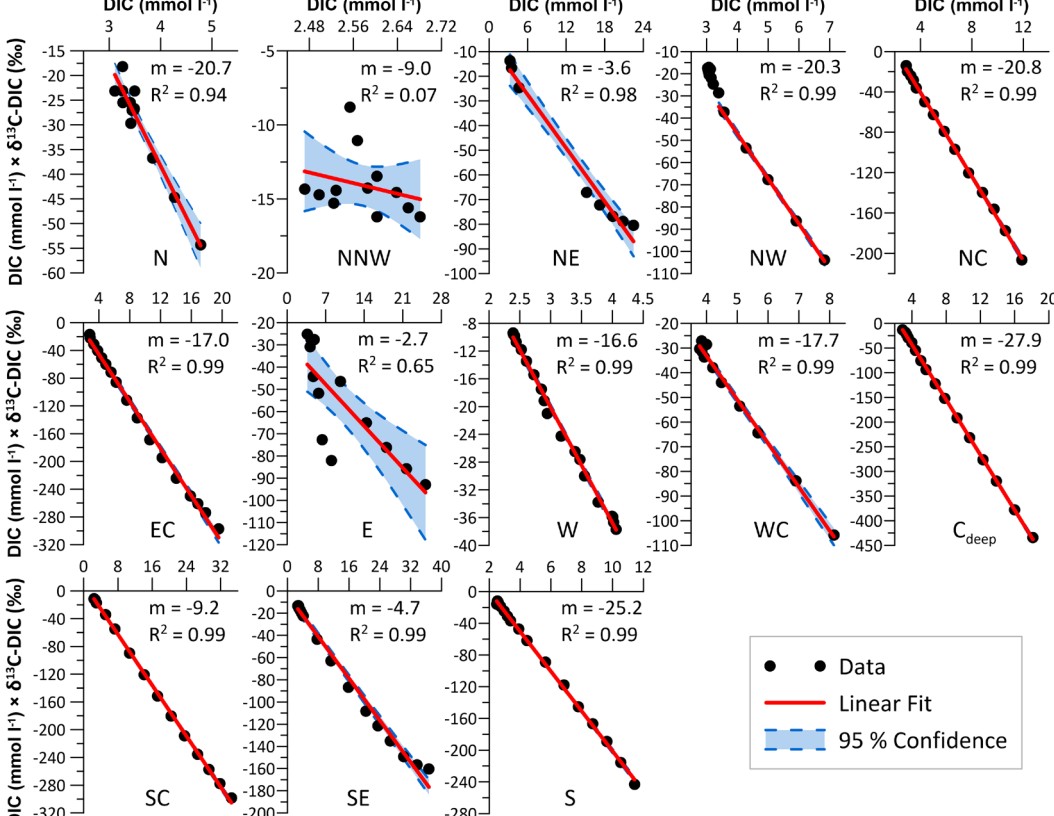

Figure 6: Miller-Tans plots of the pore-water DIC concentrations and δ¹³C-DIC compositions, with m being the slope of the linear fit and R² being the coefficient of determination. Note that at site NW the pore-water mixed layer is excluded from the fit. See Table 1 for site abbreviations.

## 4.7 Aerobic remineralisation of organic matter

The oxygen penetration depth in the western and southern HMA (sites: NW, W, WC, $C_{deep}$, SC, SE, S) is between 5 and 6 mm, while in the central and eastern HMA (sites: N, NE, NC, EC, E) oxygen penetrates slightly deeper reaching down to 7 and 8 mm. The deepest oxygen penetration depth is ~ 20 mm, measured at site NNW (Fig. 7, Supplement S2, S3). Bottom water oxygen concentrations were close to saturation (CTD data, not shown) and averaged 94,8 % relative to saturation at the sediment-water interface, with two lower outliners at sites S and E (65 % and 79 %, respectively).

Oxygen consumption rates range from 0.3 to 5.9 mmol $O_2$ m$^{-2}$ d$^{-1}$ or converted to aerobic remineralisation rates of 1.3 to 25.8 g C m$^{-2}$ yr$^{-1}$. Generally, aerobic remineralisation rates in the deeper western and southern part of the HMA are higher (sites: NW, W, WC, $C_{deep}$, SC, S; average ~ 20.9 g C m$^{-2}$ yr$^{-1}$) compared to the shallower eastern and northern HMA (sites: N, NE, NC, EC, E, SE; average ~ 12.5 g C m$^{-2}$ yr$^{-1}$). By far the lowest rates of 1.3 g C m$^{-2}$ yr$^{-1}$ are found at site NNW (Table 3).

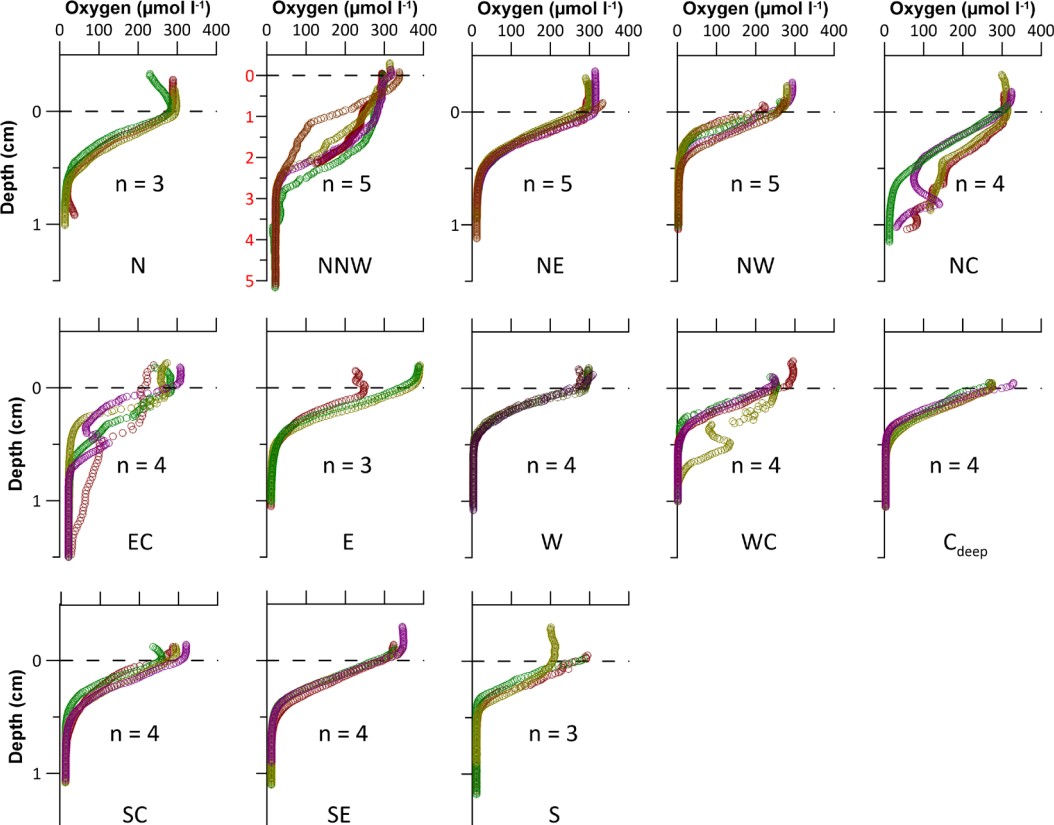

**Figure 7: Pore-water dissolved oxygen concentrations from expedition HE595 in 2022. Symbols in different colours represent different measurements (n) on the same core. See Table 1 for site abbreviations.**

## 5 Discussion

In the following, we first define the characteristics of the depositional conditions in the HMA. Based on these characteristics, we discuss the factors that control the distribution, long-term burial and preservation of OM, as well as the efficiency of organic carbon burial in the different sedimentary habitats of the study area. In the end, we present the organic carbon accumulation rate and the total annual organic carbon accumulation in the HMA.

### 5.1 Depositional conditions

We find large sedimentation rate changes over time at some of our study sites (e.g., sites: NE, W(a), $C_{deep}$(a); Fig. 3), indicating nonsteady-state depositional conditions (e.g., Kasten et al., 2003). Therefore, it needs to be re-evaluated whether the assumed boundary conditions inherent to the model apply in the study area. Common models for dating modern marine sediments with $^{210}Pb_{xs}$ are always based on certain assumptions (e.g., constant supply of sediment and/or $^{210}Pb$, steady-state conditions, absence of mixing) that are rarely met in full and may be completely inappropriate. In shallow marine environments, bioturbation or physical mixing of the sediments is often present, and the model results can overinterpret these processes as changes in sedimentation rates by up to 95 % (Arias-Ortiz et al., 2018). The CRS age model chosen here is particularly robust for averaged sedimentation rates as it uses the $^{210}Pb_{xs}$ inventory and is therefore robust in an environment where the sediments are mixed (Arias-Ortiz et al., 2018). However, sediment mixing alters the $^{210}Pb_{xs}$ profiles by transporting sediments with higher $^{210}Pb_{xs}$ activity further downcore and vice versa. In an environment like the HMA, where physical mixing and bioturbation are present (Hintzen et al., 2012; Oehler et al., 2015; Wrede et al., 2017; Thünen Institute, 2018), this could lead to an overinterpretation of sedimentation rate changes over time, resulting in overestimated sedimentation rates at the top of the core and underestimated rates in deeper core sections (Fig. 3). Therefore, we carefully address the topic of mixing before applying a dating model.

When mixing in sediments is suspected, it is possible to first visually identify mixing from the shape of the profile, as described in detail by Arias-Ortiz et al. (2018). A $^{210}$Pb$_{xs}$ profile affected by mixing can show a reverse pattern at the surface, a homogeneously mixed layer overlying the exponential $^{210}$Pb$_{xs}$ profile or a shift in the slope of the $^{210}$Pb$_{xs}$ profile (Arias-Ortiz et al., 2018; e.g., site NW and W(a), Fig. 3). However, bioturbation and physical mixing can create the same shapes of $^{210}$Pb$_{xs}$

profiles, so it is impossible to identify the exact reason for mixing from the shapes of the $^{210}$Pb$_{xs}$ profiles. By applying the regeneration model presented by Gardner et al. (1987) we successfully reproduce the influence of mixing under steady-state conditions on the shapes of the $^{210}$Pb$_{xs}$ profiles (Supplement Fig. S4). Therefore, we argue that no distinction of changes in sedimentation rates can be made and that reliable average sedimentation rates can be derived from averaged values of the CRS model (Table 4).

The age determination from the CRS age model at our study sites is generally in good agreement with the independent time marker $^{137}$Cs. There is – at maximum – only a 1 or 2 cm difference between the calculated age of 1950 and the start of the detection of $^{137}$Cs in the cores (Fig. 2, 6). The year 1950 marks the onset time of excessive nuclear weapons testing. This date was chosen as a time marker because the highest activity in atomic bomb tests in 1963 is not clearly observable as a peak in the $^{137}$Cs data. This could be due to (1) bioturbation or physical mixing blurring the sharp peak in the sediment by transporting

$^{137}$Cs into deeper layers, or (2) due to riverine sediment input and reservoir effect (e.g., Miguel et al., 2003; Smith et al., 2004; Olley et al., 2013). Our calculated sedimentation rates range between $0.5 \pm 0.1$ and $4.5 \pm 0.4$ mm yr$^{-1}$ (Table 4; Fig. 8a) and are also in good agreement with a $^{210}$Pb$_{xs}$ dated sediment core presented by Hebbeln et al. (2003) of 2.6 mm yr$^{-1}$ located in the northwestern HMA in between our sites NW, C$_{deep}$ (Fig. 8a), with sedimentation rates of $2.5 \pm 0.1$ mm yr$^{-1}$ and $2.4 \pm 0.3$ mm yr$^{-1}$, respectively. However, our sedimentation rates are higher compared to those determined by $^{14}$C dating by Boxberg et al.

(2020) in the central and southern HMA. This is probably due to our use of a radionuclide with a higher time resolution. This highlights the relevance of our high vertical and spatial resolution $^{210}$Pb$_{xs}$ data set to describe recent sedimentary conditions and its ability to resolve the conflicting approaches and outcomes in understanding the depositional conditions in this area.

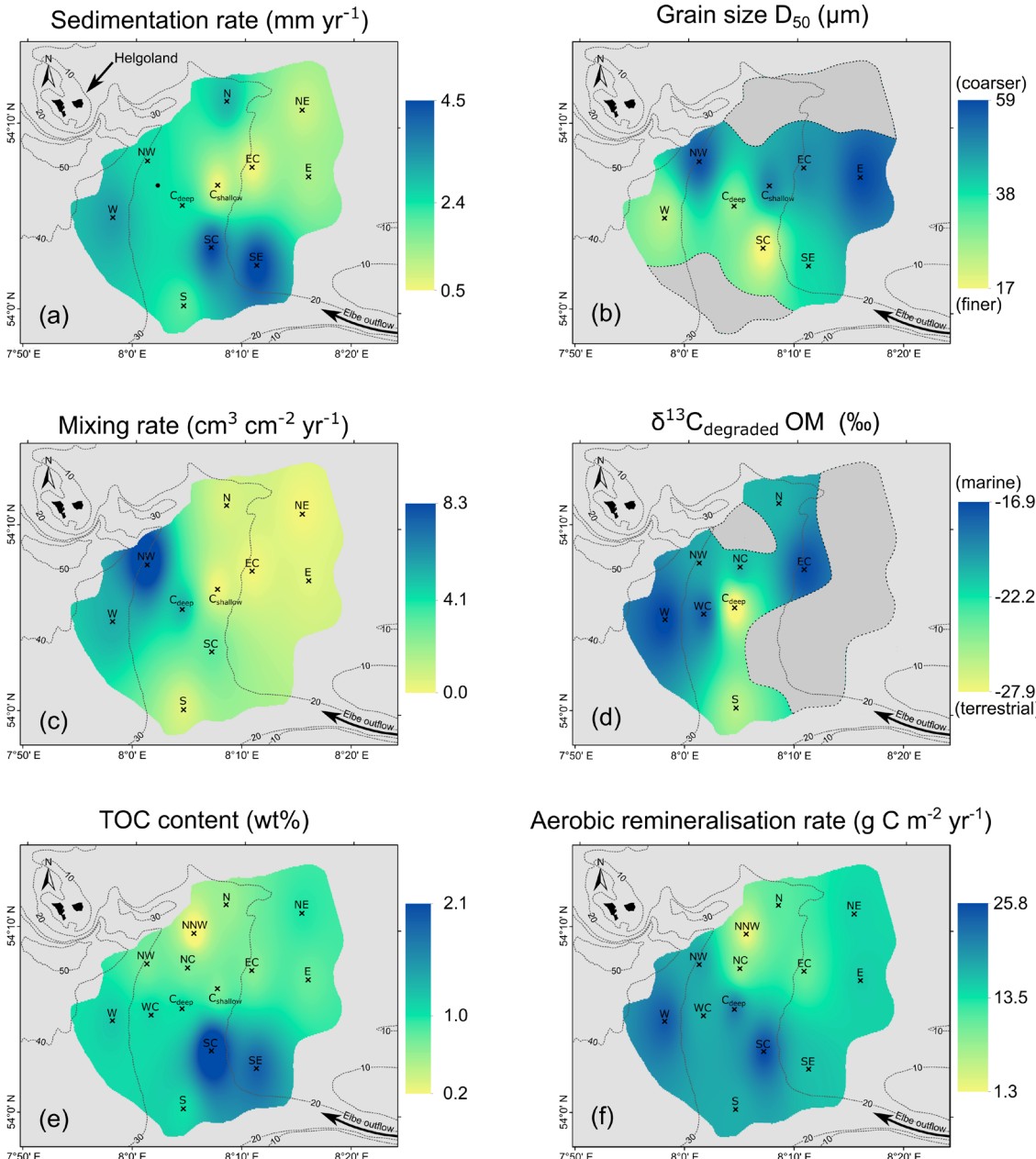

**Figure 8: Synthesis of depth-integrated and interpolated maps for selected parameters (interpolations: ArcMap, inverse distance weighted) of the values from Tab. 2 and 3, showing: (a) sedimentation rates, (b) grain sizes, (c) sediment mixing rates (d) origin of the degraded organic matter ($\delta^{13}C_{degraded}$ OM), (e) total organic carbon (TOC) contents and (f) aerobic remineralisation rates.**

In our interpolation of the recent sedimentation rates across the research area (Fig. 8a), we find highest sedimentation rates up to $4.5 \pm 0.4$ mm yr$^{-1}$ in the southeastern HMA, intermediate rates ranging between $2.0 \pm 0.2$ and $3.2 \pm 0.2$ mm yr$^{-1}$ at the deeper water depth stations and lowest rates around 0.8 mm yr$^{-1}$ in the eastern HMA. The area of highest sedimentation rates is located in the southeast, which is under the strong influence of the Elbe estuary outflow from where SPM is transported into the HMA (e.g., Puls et al., 1997), resulting in high sedimentation of fine-grained riverine and estuarine material (Fig. 8a, b). We further find that - although also being supplied with riverine and coastal water as well as high SPM concentrations - the eastern part of the HMA has lower sedimentation rates due to the higher tidal energy as it is characterized by shallow water depths and slightly coarser sediments (Maerz et al., 2016; Fig. 1; Fig. 8b). We suggest that the eastern part of the HMA represents a more energetic depositional environment that either prevents smaller, muddy particles from settling permanently or leads to resuspension and post-depositional export from the area. In the deeper HMA (western, central and northern sites), the tidal energy is accordingly lower, allowing fine material to settle more effectively. This is further supported by our grain size measurements (Fig. 8b), which coincide with the overall grain-size distribution of the German Bight (Figge, 1981; Laurer et al., 2013; Bockelmann et al., 2018; Sievers et al., 2021), showing high mud contents in the southern and western HMA (sites:

460 W, $C_{deep}$, SC, SE). This results in higher sedimentation rates compared to the eastern HMA, and because of the greater distance from the Elbe estuary, sedimentation rates in the western HMA are lower than those found towards the valley of the Elbe.

The complex hydrodynamic situation and resulting depositional mechanism in the HMA is still subject to discussion. Furthermore, this is a topic of ongoing research and will be addressed by a further study of the hydrodynamics and SPM transport. However, the deposition of fine-grained material could be a result of freshwater outflow from the Elbe and Weser

estuaries towards the HMA and continuing northwards, creating a less dense surface water layer as reported by Hagen et al. (2021). As dense seawater enters the HMA from the deeper west following the general circulation pattern in the German Bight, SPM could potentially be transported along the slope of the HMA, somehow creating a trap for SPM beneath the less dense surface water. The resulting high particle load and lower current velocity would allow particles to settle effectively in this area. This would represent a similar mechanism as described by Meade (1972) or modelled for an idealized estuary by Burchard et

al. (2021), describing the freshwater outflow from an estuary and a current pattern and SPM transport after entering the coastal ocean.

### 5.1.1 Sediment and pore-water mixing

Mixing of surface sediments in the HMA can be induced by various processes – including bioturbation or physical processes like current- and storm-induced resuspension and deposition, as well as bottom trawling and sediment dredging and dumping

(e.g., Baumann, 1991; Hintzen et al., 2012; Hamburg Port Authority, 2017; Eigaard et al., 2017; Wrede et al., 2017; Thünen Institute, 2018). The mixing causes vertical transport of sediment in two directions, while sediment from deeper layers is generally transported to the surface during bioturbation and abandoned burrows are refilled with surface sediment (Gardner et al., 1987), physical mixing by bottom trawling resuspends a layer of sediment which is mixed, eventually redeposited or subsequently refilled with SPM or sediment (e.g., Depestele et al., 2016; De Borger et al., 2021a). This sediment refilling of

trawl marks takes only a few days in the German Bight as a result of the high current energy (Bruns et al., 2020).

Based on our results we can distinguish three different environments of mixing in the HMA (Fig. 8c): (1) in the coarser-grained, shallowest areas in the HMA (sites: N, NE, EC, E, $C_{shallow}$), where the lowest mixing rates are modelled, (2) deeper, fine-grained southern area (sites: SC, S) with moderate mixing rates and (3) the deeper, fine-grained northern and western area (site: NW, W, $C_{deep}$), with the highest mixing rates, in particular at site NW. Comparing these three environments with the

485 mapped bottom trawling activity in the area from Thünen Institute (2018) after Hintzen et al. (2012) (Fig. 1b) we can propose the dominant driver behind the observed mixing of sediment. In (1) low bottom trawling activity is mapped and with low modelled sediment mixing this area represents an environment of low trawling activity and low sediment mixing by burrowing benthic organisms. In contrast, in the deeper southern HMA (2), where bottom trawling activity is less frequent or absent, higher mixing rates are found (Fig. 1b, 8c), presenting an area of potentially higher activity of burrowing organisms. In (3) we

find the highest sediment mixing rates at the site of highest mapped trawling activity at site NW (Fig. 1b) and comparably lower mixing in areas of lower mapped trawling activity (sites: W, $C_{deep}$).

Besides the mixing of sediment, pore water can be mixed in the sediment. Event-driven mixing of pore water with bottom water (e.g., storm events, bottom trawling) could be an additional and important process affecting $O_2$ and nutrient exchange and fluxes in the surface sediments and in the way influencing the biogeochemical cycle (e.g., Aller, 1994; De Borger et al.,

2021a). We visually identified pore-water mixing by evaluating the profile shapes of DIC and $\delta^{13}$C-DIC. At sites E, SC and SE in the shallow eastern HMA, we identify mixing based on DIC concentrations being low and constant in the uppermost $\sim$ 10 cm of the sediments and the isotopic signature being close to or similar to bottom water values of around 0 to -2 ‰ (Kroopnick, 1985; Fig. 3). In addition to the visual comparison, the result of pore-water and bottom-water mixing is also evident in the results of the Miller-Tans plots. At sites E and NE there is a poor fit of the linear regression and at sites SC and

SE $\delta^{13}C_{degraded}$ OM values are close to the bottom water signature of 0 ‰ (Fig. 6). Both indicate pore-water mixing with bottom water. At the sites where pore water is mixed with bottom water, no evidence of a similarly mixed sediment layer is found

based on $^{210}Pb_{xs}$ profiles (see Fig. 2). As a consequence, we propose that at sites NE, E, SC and SE solely mixing of pore water has occurred without significantly affecting the sediment particles. Since a series of storm events were reported for the German Bight only a few weeks before sampling in 2022 (Abromeit et al., 2022) and only the shallowest eastern stations are affected by pore-water mixing, it could be a seasonal effect of pore-water mixing during the winter storm season. The data set presented in this study allows only for speculation about the mechanism of the pore-water mixing. However, processes causing pore-water mixing without disturbing the sediment include wave pumping or migration of gas bubbles (e.g., Santos et al., 2012). Pore water could be mixed with bottom water by wave pumping during times of larger pressure gradients associated with storm events. However, following the empirical relationship between grain size ($D_{50}$) and permeability presented by Neumann et al. (2017) the sediments of the HMA were classified as impermeable sediments (permeabilities below $10^{-14}$ m$^2$). Pore-water mixing as a result of gas bubble migration seems unlikely since this would cause a smoother gradient in the pore-water profiles and also be evident in deeper parts of the sediment (e.g., Haeckel et al., 2007).

**5.2 Distribution and preservation of organic carbon**

In the HMA, areas of high sedimentation rates coincide with high preserved TOC contents (Fig. 8a, e). While the correlation between TOC vs. sedimentation rate is significant (Pearson correlation (cor) = 0.73, p-value (p) < 0.05), the correlation between TOC vs. MAR is not significant (cor = 0.37, p = 0.26; Fig. 9a). These two parameters, although closely interlinked, differ because, unlike the calculation of MAR, porosity is taken into account in the calculation of sedimentation rate (e.g., Sanchez-Cabeza and Ruiz-Fernández, 2012). As the porosities differ between sites (Table 3), sites with similar MAR and different porosities result in different sedimentation rates. Since the time in which OM is exposed to $O_2$ is decisive for its preservation (e.g., Jung et al., 1997; Hartnett et al., 1998) sedimentation rates describe the parameter which represents the time in which OM is exposed to $O_2$, and thus correlate significantly with preserved OM.

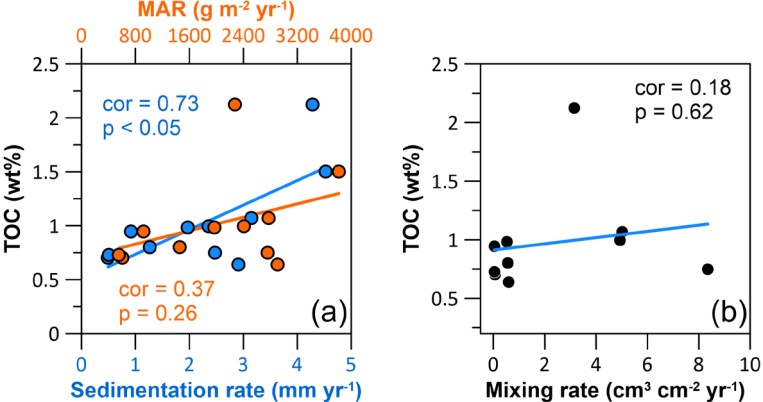

**Figure 9: Cross plots with Pearson correlation coefficients (cor) and corresponding p-values (p) showing the correlation between: (a) total organic carbon (TOC) content vs. the sedimentation rate (blue) and TOC content vs. sediment mass accumulation rate (orange) and (b) TOC content and the sediment mixing rate.**

Besides oxygen exposure time, the remineralisation of OM and hence its preservation is also dependent on the origin and reactivity of the OM (e.g., Hedges et al., 1988, 2000; Burdige, 2005; LaRowe et al., 2020; Freitas et al., 2021; Xu et al., 2023). It is therefore of interest to determine the reactivity of the OM available for degradation in the HMA and its role for preservation. OM of terrestrial and marine origin differ in their reactivity, with marine OM being more labile and preferentially degraded in the sediment (e.g., Hedges et al., 1988). Those two endmembers also differ in the stable carbon isotopic composition ($\delta^{13}$C) of TOC, with $\delta^{13}$C-TOC values for marine OM of -18 to -21 ‰ and terrestrial OM of -25 to -33 ‰, respectively (Lamb et al., 2006; and references therein). Applying this classification to our results of the Miller-Tans plots, we can use the Miller-Tans plots as a proxy to determine the reactivity and source of organic matter currently being degraded in the sediments (Wu et al., 2018). For the sediments at all study sites, except the mixed pore water sites in the eastern part of the HMA, we find a clear linear trend in the Miller-Tans plots (Fig. 6). Since the anaerobic oxidation of methane (AOM) (e.g.,

Niewöhner et al., 1998) typically results in a 'bend profile' (steeper slope) of the profile at higher DIC contents, we can exclude a contribution of DIC originating from AOM in the signal (Wu et al., 2018). Moreover, the sulfate/methane transition zone i.e. the depth where the anaerobic oxidation of methane by sulfate occurs – was shown to be situated in deeper subsurface sediments of the HMA (Oni et al., 2015a). Our results are between -17 ‰ and -28 ‰ and well within the range after Lamb et al. (2006) and thus show significant differences in the origin of the degraded OM, with marine OM being predominantly degraded in the western HMA and terrestrial OM in the southern to central HMA (Fig. 8d). Based on previous findings marine OM is generally more reactive than terrestrial OM (e.g., Hedges et al., 1988). This is consistent with observations from Oni et al. (2015b), who described the degradation of easily degradable marine OM at the uppermost ~ 50 cm of the sediments also in the western HMA using the stable carbon isotopic composition of TOC ($\delta^{13}$C-TOC). Similarly, for a site close to our site WC, Hebbeln et al. (2003) report a trend in the $\delta^{13}$C-TOC signatures from higher (-22 ‰) to lower (-25 ‰) values with depth. This indicates the degradation of marine OM within the uppermost ~ 40 cm of the sediments, which is consistent with our results showing the degradation of marine OM at site WC. As a consequence, we suggest that sediments in the western HMA receive significant amounts of marine OM which is preferentially degraded while a higher proportion of terrestrial OM is buried on a longer term. Accordingly, in the southern and central HMA, where terrestrial OM is predominantly degraded, the proportion of marine OM must be lower at time of deposition. Further, we find $\delta^{13}C_{degraded}$ OM values indicating mixing of a lower marine and higher terrestrial OM fraction at intermediate water depths from the terrestrially influenced south to the north of the HMA, strengthening the idea of a SPM trap along the slope of the HMA, as described above. The terrestrial OM most likely supplied by the outflow of the Elbe river is subsequently mixed with material of marine origin from south to north (see Fig. 8d).

No significant correlation was found between the origin of degraded OM and TOC content (cor = -0.16, p = 0.70). However, this is likely a result of the pore-water mixing, which limits the number of sites that can be compared (grey area Fig. 8d). For the overall preservation of OM in the HMA we conclude that while in the western HMA a mixture of marine and terrestrial OM is deposited, the marine fraction is degraded in the sediment and a majority of the preserved OM is of terrestrial origin. In the southern HMA, where we find the overall highest TOC contents, the majority of the OM deposited is already of terrestrial origin. This highlights the overall strong terrestrial influence on the sediments in the HMA and its potential and importance for the long-term preservation of OM.

**5.2.1 The effect of sediment mixing and bottom trawling on OM preservation**

The extent of bioturbation and/or physical mixing does not show an obvious effect on the long-term preservation of OM in sediments in the HMA. Hence no significant correlation between sediment mixing rates and TOC contents is observed (cor = 0.18, p = 0.62; Fig. 9b). Although we only have a few study sites in the northwestern HMA, which is most affected by high bottom trawling activity (Fig. 1b) we will briefly evaluate the effect of intense bottom trawling on a separate basis. The three sites W, $C_{deep}$ and NW, located in close geographical proximity to each other and at similar water depths in the northwestern HMA are characterized by similar sedimentation rates (2.5 to 3.2 mm yr$^{-1}$). However, they are affected by bottom trawling to significantly different degrees (Fig. 1b), as reflected in our modelled sediment mixing rates, with the highest value of 8.35 cm$^3$ cm$^{-2}$ yr$^{-1}$ at site NW and lower values (5.02 and 4.94 cm$^3$ cm$^{-2}$ yr$^{-1}$) at sites W and $C_{deep}$ (Table 3, Fig. 8c). The sites in the northwest with the highest mixing rates also coincide with the highest bottom trawling activity reported in the HMA (as swept area ratio in Fig. 1b). Similarly to our modelled sediment mixing rates, the bottom trawling activity is lowest at sites W and $C_{deep}$, the southernmost of these three sites and highest at site NW (Fig. 1b). By comparing the TOC contents of the three sites W, $C_{deep}$ and NW, we find ~ 27 % lower TOC contents at site NW (TOC contents of sites W, $C_{deep}$, NW: 1.07, 0.99, 0.75 wt%, Table 4) indicating that high bottom trawling activity could have substantially reduced the long-term preservation of OM. This result is well within the range for the reduction of OM by 20 to 60 % reported for a long-term bottom-trawled area by Paradis et al. (2019). Also, a model approach by De Borger et al. (2021) showed significant OM reduction due to frequent bottom trawling, by resuspension, and transport of oxygen and nutrients from the water column deeper into the sediment. While these

processes might have no or only little impact on the long-term preservation of OM in a moderate sediment mixing environment (e.g., bioturbation) in the HMA, we find that the intensive sediment mixing at site NW seems to lower the long-term OM preservation in sediments by about 30 %. This also agrees with recent findings by Zhang et al. (2024), who showed that intense and persistent bottom trawling in the North Sea lowers sedimentary organic carbon, especially in the fine sediment fraction. In addition, Kuderer and Middelburg (2024) recently showed that strong sediment mixing increases the spectrum of OM reactivities in each layer of the sediment, which can stimulate overall OM degradation. The differences in grain size between the three sites in the HMA may be also a result of recurrent bottom trawling, reducing the resilience of the sediment to resuspension and increasing sediment remobilization, preferentially affecting fine particles (O'Neill and Summerbell, 2011; Mengual et al., 2016; Bruns et al., 2023). However, more studies, including different environments, will be needed to provide reliable estimations on the impact of bottom trawling on OM preservation in sediments.

**5.2.2 Seasonal effects on OM preservation**

When assessing the individual TOC profiles of each station, we do not observe the classical downward decrease in TOC contents that would go along with progressing OM degradation under steady-state conditions (e.g., Arndt et al., 2013; Fig. 4). We found, that TOC contents in the surface sediments are either constant or increase with depth (e.g., sites W, EC, E, SC) – a pattern that has also been reported earlier by van der Zee et al. (2003) who sampled the central HMA in spring season. Since our samples were also retrieved during the same season, this pattern seems to be prevalent at this time of the year, whereas studies investigating samples retrieved in August show a downward decrease in TOC profiles (Dauwe and Middelburg, 1998; van der Zee et al., 2003).

During the algae bloom season that starts in late spring in the German Bight (Teeling et al., 2012), large amounts of freshly produced marine OM reach the seafloor, where bioturbation mixes the uppermost surface layer and produces depth-decreasing TOC profiles. After the algae bloom season is over around the beginning of November (Teeling et al., 2012) OM consumption in this uppermost surface layer continues, which then causes such reverse profiles as we observed in spring at sites W, EC, E and SC (van der Zee et al., 2003; Oehler et al., 2015). Due to the seasonally changing fluxes of marine OM to the seafloor the TOC contents oscillate in the upper centimetre with lower contents in winter/early spring and higher contents in later spring/summer. Below this certain depth, a rather constant TOC content is preserved.

**5.3 Organic carbon burial efficiency**

A quantitative approach to assess the long-term preservation of OM in marine sedimentary environments and to assess their long-term carbon storage capacity is to determine the organic carbon burial efficiency (OC BE). It has been demonstrated that aerobic remineralisation is the key process controlling post-depositional OM remineralisation and preservation (e.g., Hartnett et al., 1998; Jørgensen, 2006). We have therefore used high-resolution in situ oxygen concentration profiles to calculate rates of post-depositional aerobic remineralisation. The distribution of aerobic remineralisation rates in the HMA shows high values in the deeper, western and southern parts of the HMA, while the shallower eastern and northern stations show slightly lower rates with a minimum at the site NNW, where an underlying sandy structure was sampled (Fig. 8f). Our calculated aerobic remineralisation rates range from 1.3 to 25.8 g C m$^{-2}$ yr$^{-1}$ and overlap with the range of rates derived from benthic chamber measurements in muddy sediments of the HMA of 8.8 to 35.9 g C m$^{-2}$ yr$^{-1}$ in March 2013 and 2014 (Oehler et al., 2015). The aerobic remineralisation rates presented here are, however, a conservative value for the study area as we sampled in spring, before the algae bloom. Site NNW with no active sedimentation (von Haugwitz et al., 1988) is excluded from the following discussion on OC BE since it is not possible to describe a TOC burial flux here. For sites WC and NC, we used interpolated sedimentation rates for the calculations of the OC BE. This was done on the basis that site WC is located between our two sites W and C$_{deep}$, which have similar characteristics, and site NC, which, in addition to the high density of our measurements, is close to the well-dated site from Hebbeln et al. (2003).

Overall, lowest OC BE of ≤ 40 % were determined for sites NE, EC and E in the shallow HMA (Fig. 10a). Intermediate OC BE values of around 50 % are found at sites NW, W, WC, $C_{deep}$ and S in the deepest western and southern part of the HMA, whereas the highest OC burial efficiencies of ≥ 60 % occur in the southern, central and northern parts of the HMA, namely at sites N, NC, SC and SE (Fig. 10a). We find that a comparably high OC BE is not necessarily characterised by a high OC burial flux (e.g., site N; Fig. 10b), but OC burial fluxes and OC BE are interrelated.

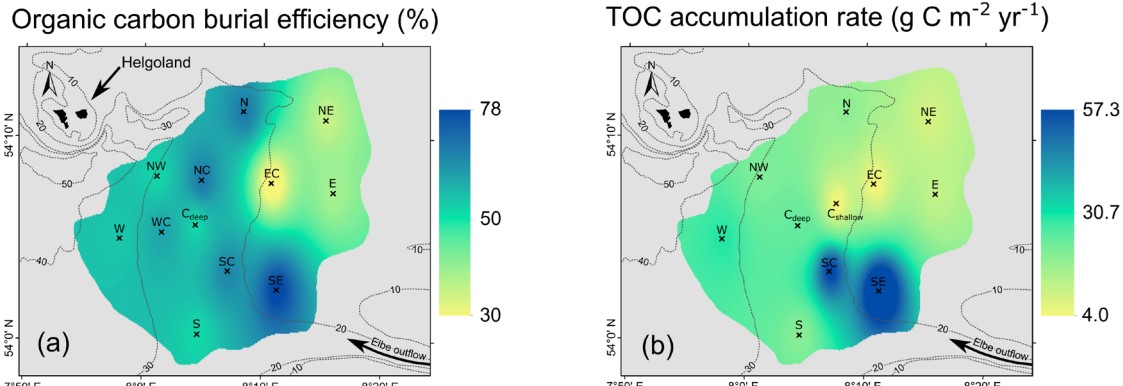

Figure 10: Interpolated maps using ArcMap (inverse distance weighted), showing: (a) organic carbon burial efficiencies and (b) total organic carbon (TOC) accumulation rates.

The observed pattern of OC burial efficiencies can be explained by the variation in depositional conditions and OM characteristics found in the HMA. We find highest OC BE (Fig. 10a) at the sites with small grain sizes (Fig. 8b) and highest TOC accumulation rates (sites: SC, SE; Fig. 10b). Lower sedimentation rates, as found in the eastern HMA, increase oxygen exposure time, while the coarser grain sizes indicate that lower amounts of OM are deposited in association with fine particles (e.g., Bockelmann et al., 2018). This environment of low sedimentation rates and lower OM deposition results in lower OC BE. Total organic carbon accumulation rates (Fig. 10b) and aerobic remineralisation rates (Fig. 8f) are the two parameters used to calculate the OC BE (Eq. 1). By comparing the sedimentation/sediment mass accumulation rate and the aerobic remineralisation rate with the OC BE, we can assess which of the factors has a greater impact on the OC BE. Our data show, that the sedimentation/sediment mass accumulation rate (cor = 0.96/0.93, p < 0.05 for both; Fig. 11a) is the dominant factor for efficient OM preservation, whereas the aerobic remineralisation rate does not have a major impact on the OC BE (cor = 0.21, p = 0.54; Fig. 11b). Although there is no linear relationship between OC BE and the origin of the degraded OM (cor = -0.17, p = 0.69, Fig. 11c), due to the exclusion of pore-water mixing sites (Fig. 8d, grey area), we find an influence of the OM origin on the OC BE in the HMA. At sites with high OC BE, including the sites with high sedimentation rates in the south, $\delta^{13}C_{degraded}$ OM values show terrestrial or mixed terrestrial and marine origin. This area of high OC BE also represents the location of a potential trap for sediment – from south to north on the slope of the HMA. Here, the combination of higher sedimentation rates and the deposition of less reactive terrestrial OM are the most favourable combination for the efficient burial of OM in the study area. Sites characterised by high sedimentation rates and marine OM being primarily degraded, as found in the western and deeper HMA, show ~10 % lower OC BE.

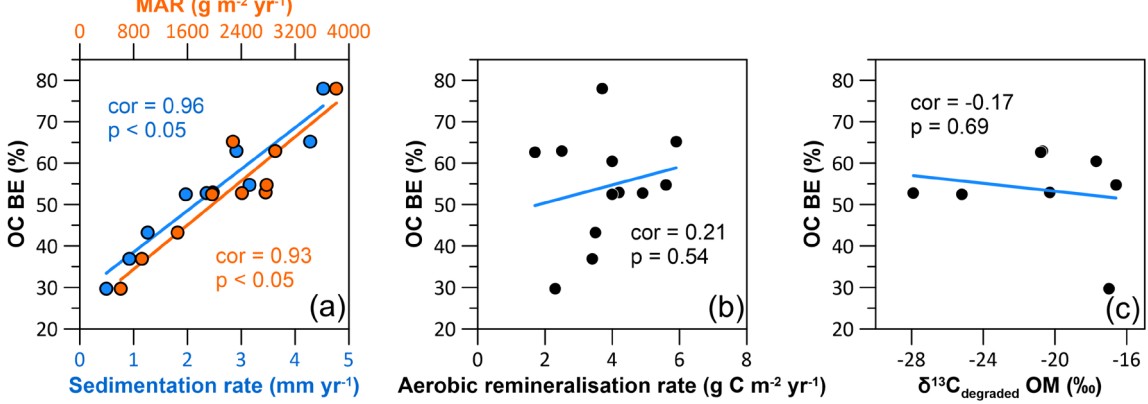

**Figure 11: Cross plots with Pearson correlation coefficients (cor) and corresponding p-values (p) showing the correlation between: (a) organic carbon burial efficiency and the sedimentation rates (blue) and organic carbon burial efficiency and sediment mass accumulation rate (orange), (b) organic carbon burial efficiency and aerobic remineralisation rate and (c) organic carbon burial efficiency and stable carbon isotopic composition of the degraded OM ($\delta^{13}C_{degraed}$ OM).**

Comparing our results of the OC BE with the compilations by Canfield (1994) and Burdige (2007), we find that the OC BE in the HMA are well within the range for marine sediments with oxygen saturated bottom waters (Fig. 12, black line). Our sites characterised by low sedimentation rates/MAR, are average for OC BE in marine sediments, while our highest sedimentation rates/MAR sites are exceptionally efficient at burying OM compared to the compilation presented by Canfield (1994) and Burdige (2007) (Fig. 12). On the Monterey Bay shelf off California at 100 m water depth with sedimentation rates of 2 mm

655     yr$^{-1}$ the OC BE is ~ 33 %, slightly lower than in the HMA (Berelson et al., 2003). Similar values are reported by Song et al. (2016) at 50 to 100 m water depth on the Yellow Sea shelf that is influenced by the Yellow River. There, the OC BE averages at 32 % with an average sedimentation rate of 1.6 mm yr$^{-1}$. Higher OC BE of 45 % are found at sites on the East China Sea shelf, where sedimentation rates are more than five times higher, averaging 8.4 mm yr$^{-1}$. Compared to the HMA, where the sedimentation rates are equally high, the OC BE in the South China Sea is still low, which might be attributed to a smaller

fraction of terrestrial, less reactive OM accumulating (< 4 %) on the East China Sea shelf. A similar trend has been reported on the Louisiana shelf by Sampere et al. (2011), sampling a transect from the Mississippi delta offshore the shelf at water depths between 50 and 100 m. Proximal to the delta, with high sedimentation rates up to 59 mm yr$^{-1}$ and strong riverine influence OC BE are up to 92 %. At sites further from the shore with similar sedimentation rates (~ 5 mm yr$^{-1}$) as in the HMA, OC BE are about 80 % and decreases further offshore on the shelf and distal to the Mississippi delta. Furthermore, Jørgensen

et al. (1990) and Ståhl et al. (2004) reported OC BE between 9 and 73 % in water depths ranging from 17 to 562 m and showed, similar to the HMA, increasing OC BE with increasing MAR and sedimentation rates in sediments of the Baltic Sea - North Sea transition and into the deeper Skagerrak.

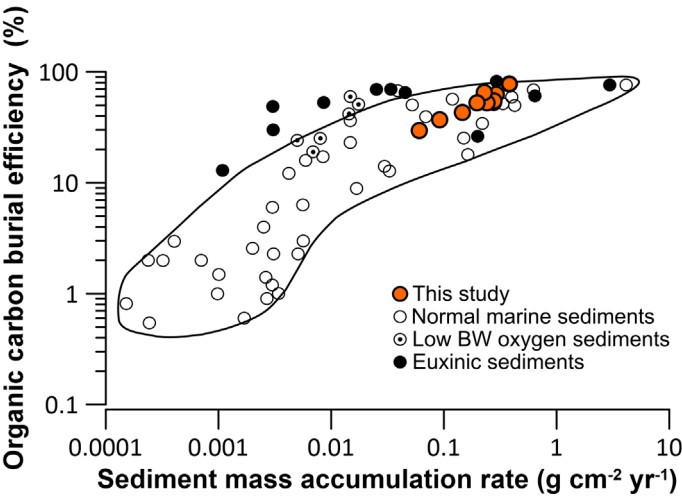

**Figure 12: Organic carbon burial efficiencies (OC BE) against sediment mass accumulation rates, and the range for normal marine**
**sediments (black line) after Canfield (1994) and Burdige (2007) and references therein.**

## 5.4 Annual organic carbon accumulation in the Helgoland Mud Area

Based on the distribution of total organic carbon accumulation rates in our study area (Fig. 10b), we calculated an areal mean TOC accumulation rate of 22.5 g C m$^{-2}$ yr$^{-1}$. The annual organic carbon accumulation for the entire HMA was calculated to amount to 0.011 Tg C yr$^{-1}$. Comparing our value for the annual organic carbon accumulation for the entire HMA with the one-point estimate from de Haas et al. (1997) of 0.022 Tg C yr$^{-1}$, we find that de Haas et al. (1997) overestimated the TOC content by ~ 20%, sedimentation rate by ~ 30% and used a ~ 22% larger area for the HMA. By using the spatial distribution of the relevant parameters, which were calculated individually for each site, and subsequent interpolation (Fig. 10b), our study presents a more robust estimate of the annual organic carbon accumulation for the HMA. Our estimated TOC accumulation rates and annual organic carbon accumulation for the HMA represent, however, a conservative value for the study area as we sampled in spring, before the algae bloom. A rough estimate of a summer sampling scenario after the algae bloom with increased TOC contents in the first two cm of the sediments of 2.9 wt% and 2.1 wt% respectively (Dauwe and Middelburg, 1998; van der Zee et al., 2003) would result in a mean TOC accumulation rate of 27.3 g C m$^{-2}$ yr$^{-1}$ and an annual organic carbon accumulation for the HMA of 0.013 Tg C yr$^{-1}$. Comparing our results with the modelled results from Diesing et al. (2021) based on quantile regression forests we find that the HMA, which covers only 0.09 % of the area of the North Sea accounts for 0.76 % of its total annual organic carbon accumulation. Furthermore, the average TOC accumulation rate in the HMA falls into the range of the Skagerrak and Norwegian Trough (~ 10 to 66 g C m$^{-2}$ yr$^{-1}$) that represent the main depocentres for fine-grained and organic-rich sediments in the North Sea (Diesing et al., 2021). Further hotspots for TOC accumulation on the northwest European shelf have been described by Diesing et al. (2024), who found high TOC accumulation averaging 9.3 ± 6.89 g C m$^{-2}$ yr$^{-1}$ in glacial trough sediments.

## 6 Conclusions

In this study, we present a new high-resolution solid-phase and pore-water data set with both high spatial and vertical resolution to determine the factors controlling the burial and preservation of organic matter (OM) in sediments from the Helgoland Mud Area (HMA). We used this comprehensive data set to investigate variations in sedimentation rates, sediment mixing rates, grain sizes, total organic carbon (TOC) contents and aerobic remineralisation rates. We found that in the HMA, the dominating factor for TOC preservation and organic carbon burial efficiency is the sedimentation rate, resulting in burial efficiencies of up to 78 % for a corresponding high sedimentation rate of 4.5 ± 0.4 mm yr$^{-1}$ as found in the southern part of the HMA. In this area of the southern HMA, which is influenced by the Elbe river outflow, the OM being degraded is primarily of terrigenous origin, while in the central and northern part of the HMA, a mixture of marine and terrigenous OM is remineralised. At the sites dominated by the degradation of marine organic matter, as found in the western and northwestern HMA, the organic carbon burial efficiency is lower and fluctuates around 55 %. Our modelled sediment mixing rates are highest in the northwestern HMA and coincide with the highest reported bottom trawling activity. The comparison of sites in the northwestern HMA, which are characterised by similar depositional conditions but different intensities of bottom trawling suggests that intense bottom trawling reduced OM sequestration by about 30 %. Overall, the average total organic carbon accumulation rate is 22.5 g C m$^{-2}$ yr$^{-1}$, which totals 0.011 Tg C yr$^{-1}$ for the entire HMA. While the area of the HMA covers only 0.09 % of the area of the North Sea it accounts for 0.76 % of its total annual organic carbon accumulation, highlighting the significance of depocentres of fine-grained sediments for the burial of particulate organic carbon.

## Data availability

The data presented in this study are available via the PANGAEA repository, including oxygen micro profiles (https://doi.org/10.1594/PANGAEA.970325; Müller et al., 2024a), pore-water dissolved inorganic carbon (DIC) and its stable

carbon isotopic composition ($\delta^{13}$C-DIC) (https://doi.org/10.1594/PANGAEA.969033; Müller et al., 2024b) and solid-phase porosity, grain size, TOC, $^{210}$Pb$_{xs}$ and $^{137}$Cs data (https://doi.org/10.1594/PANGAEA.968969; Müller et al., 2024c).

**Author contribution:**

DM, SK and MH designed the study; MH applied for the expedition HE595; DM, MH, DB, ID and SK performed the sampling and analyses on board during the cruises with RV *Heincke* HE575 and HE595; DM, DB, ID, LS, HT and EM carried out laboratory measurements at AWI Bremerhaven, AWI Sylt, MARUM and Faculty of Geosciences, University of Bremen; BL modelled the oxygen consumption rates; DM prepared the figures, tables, wrote the original manuscript and the revised version, with essential contributions of BL, WG, MH, LS, EM, HT, SH, KUH and SK.

**Competing interests**

The authors declare that they have no conflict of interest.

**Acknowledgements**

We thank the captains, crew and scientific teams on board RV *Heincke* cruises HE575 (Grant Number: HE-575) and HE595 (Grant Number: HE-595) for their technical and scientific support. Thanks to Fabrizio Minutolo (Hereon) for the support in on-board sampling. For analytical support in the home laboratory and during data analysis, we thank Ingrid Stimac, and Maja Leusch (both AWI). Brit Kockisch is thanked for performing the TOC measurements and Muriel Böschen for the grain size measurements. We thank Dr. Jessica Volz for the constructive discussions. We thank Dr. Sebastian Müller from the MPI for Biological Cybernetics in Tübingen for support with the MATLAB implementation. Finally, we thank the anonymous reviewers for their helpful and constructive comments and suggestions on the manuscript.

**Financial support**

This research was funded by the German Federal Ministry of Education and Research (BMBF) MARE:N project "Anthropogenic impacts on particulate organic carbon cycling in the North Sea (APOC)" (03F0874A). We acknowledge additional financial support from the Helmholtz Association (Alfred Wegener Institute Helmholtz Centre for Polar and Marine Research) in the framework of the Helmholtz Research Program "Changing Earth – Sustaining our Future" in PoF IV and Germany's Excellence Strategy Cluster of Excellence EXC-2077-390741603.

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
