# Peer review of "Depositional controls and budget of organic carbon burial in finegrained sediments of the North Sea – the Helgoland Mud Area as a natural laboratory"

_EGUsphere, 2024_

## Author Comment (AC1)

Dear Stefano Ciavatta and anonymous Reviewer #1,

We thank the reviewer for the thoughtful and constructive feedback on our manuscript. We agree that it will be useful to provide additional information in the supplementary material on the literature review. Regarding the methodology of the organic carbon burial efficiency in the comparison part, a detailed reply on this topic is given in the reply to Reviewer #2. During revision of the manuscript, the other minor specific comments will be addressed.

Kind regards

Daniel Müller, on behalf of all co-authors

---

## Author Comment (AC2)

Dear Stefano Ciavatta and anonymous Reviewer #2,

We thank the reviewer for the detailed and insightful comments, which will help to improve the manuscript. In the following, we would like to comment on the major concerns of Reviewer #2. Further minor or individual comments will be addressed during the revision stage of the manuscript.

1) We do not agree with the statement that our paper/study is mainly of a regional interest. As described in our manuscript, to our knowledge, there is no study available so far that has performed a detailed assessment of the factors that control the burial/preservation of organic matter (OM) in fine-grained sediments of the North Sea. We have therefore used the Helgoland Mud Area, which is characterized by a broad range of natural depositional conditions (water depth, sedimentation rates, origin/reactivity of organic matter, etc.) - as a test field or natural laboratory, respectively, to determine the key factors that control OM burial in the North Sea. Only in this way we were able to determine the key factors in the required – and at the same time unprecedented - high spatial resolution – including the extremely time-consuming radiometric analyses ($^{210}$Pb, $^{226}$Ra, $^{137}$Cs). In this respect the results obtained in the framework of this study present key findings for the whole North Sea as well as for shelf seas in general/globally– as these represent the most important seafloor areas for the longterm burial of organic carbon and thus key natural sinks for $CO_2$. Moreover, the North Sea is a crucial area in Europe due to its important role in economy (e.g. offshore energy, fishing activities, navigation) and ecology. Our results are not only important for better understanding carbon fluxes and burial in the North Sea, with important implications for estimations of European blue carbon wealth, but also provides key process-understanding that can be applied to all mud belts on continental shelves worldwide.

We thank the reviewer for the suggestion to extend the comparison with similar environments, in order to discuss and complement the global comparison. We will include and discuss more studies in similar environments in addition to the global comparison and Figure 12.

2) Question on the reproducibility of TOC measurements

The presented high-quality TOC measurements were performed in the group of Prof Elda Miramontes at the University of Bremen Faculty of Geosciences and MARUM, a laboratory with a multi-year expertise in C measurements. Based on replicate analyses of two different standards, the precision (relative standard deviation in percentage) was 1.23 % for the standard of 0.99 wt% (n=15), and 1.34 % for the standard 5.00 wt% (n=7). We will add those details to the material and method section.

Further we revisited two sites from the cruise in 2021 (HE575) in 2022 (HE595) and the TOC profiles from both years match (as described in Fig. 7 in the manuscript), highlighting the precision of the analysis and consistency of the sampled sites.

3) Concern regarding the calculations/intervals used for the determination of total organic carbon (TOC) contents and TOC accumulation rates.

[Figure]

**Figure 1**: TOC contents (orange squares) of (a) the interval used in the study and the bottom 5 cm of the core, (b) the interval used in the study and the uppermost 5 cm of the core and (c) the bottom 5 cm of the core and the uppermost 5 cm of the core. The black line is the 1:1 relation.

We can demonstrate that the TOC contents show no or only little variation over depth and remain almost constant (Fig. 1a-c). Within the natural variability we see no clear/systematic deviation from the 1:1 line. Therefore, we consider the values presented in the manuscript and the following calculations using these values to be sound.

4) Comment regarding the organic carbon burial efficiencies (OC BE).

As outlined in the Materials and Methods chapter we calculated the organic carbon burial efficiency as described by Burdige (2007) and van de Velde et al. (2023): we assume aerobic respiration to contribute most of the total organic carbon remineralisation since no further decrease in TOC over depth can be found in our data (see Fig. 1), similar to the work by Mouret et al. (2010), Sobek et al. (2011) and Oguri et al. (2022). It is correct, that we did not consider the contribution of anaerobic remineralisation to the total organic carbon remineralisation in this paper. Previous studies have demonstrated that in depositional settings characterised by oxic bottom-water conditions, aerobic respiration is the dominant pathway of organic matter degradation. Moreover, diffusive oxygen uptake was shown to indirectly also include anaerobic degradation pathways because part of the oxygen is consumed by oxidation of reduced reaction products liberated into pore water as a consequence of anaerobic mineralization pathways (e.g., Wenzhöfer and Glud, 2004; Glud, 2008 and references therein). We have quantified the possible offset by the fraction of anaerobic remineralisation and found it justified to use just our present data for comparison within the HMA and with the literature, especially as there is no standard approach (for various approaches see Henrichs and Reeburgh, 1987; Aller, 1998; Ogrinc et al., 2003; Sobek et al., 2011; Sampere et al., 2011; Baloza et al., 2022). In our opinion, the application of a full diagenetic model to address anaerobic remineralisation processes exceeds the scope of this paper and is subject of a follow-up manuscript currently in preparation.

5) Comment on the preservation of TOC in an area which is fuelled with particles of different origins

It is correct that (1) we compare the TOC contents and therefore the preservation of OC in the study area and (2) that different sources deliver OC to different areas within the HMA. However, this is exactly what we discuss, that a difference in origin and hence reactivity is - besides the oxygen exposure time/sedimentation rate - a key factor for the preservation and burial (efficiency) of OC on shelf sediments.

6) Question regarding the influence of bottom trawling on organic carbon preservation

Thank you for this comment. It is true that the grain sizes differ between the sites (coarser at the frequently trawled site, finer at the other two). However, we think this is no reason not to evaluate the impact of bottom trawling on the preservation of OC at these sites. Bottom trawling not only enhances the oxygen exposure time and hence aerobic remineralisation but also causes resuspension of the sediments, preferentially affecting fine particles as shown by e.g. O'Neill and Summerbell (2011). Thus, bottom trawling reduces the resilience of the sediment to resuspension (Bruns et al., 2023). Recurrent bottom trawling at the same site and natural events will then lead to increased sediment remobilization (Bruns et al., 2023) and result in coarser residual sediments (Mengual et al., 2016). We will add this aspect and discuss this in this section accordingly, to make this point clear.

Kind regards

Daniel Müller, on behalf of all co-authors

**References**

Aller, R. C.: Mobile deltaic and continental shelf muds as suboxic, fluidized bed reactors, Mar. Chem., 61, 143–155, https://doi.org/10.1016/S0304-4203(98)00024-3, 1998.

Baloza, M., Henkel, S., Geibert, W., Kasten, S., and Holtappels, M.: Benthic carbon remineralization and iron cycling in relation to sea ice cover along the eastern continental shelf of the Antarctic Peninsula, J. Geophys. Res. Ocean., 127, https://doi.org/10.1029/2021JC018401, 2022.

Bruns, I., Bartholomä, A., Menjua, F., and Kopf, A.: Physical impact of bottom trawling on seafloor sediments in the German North Sea, Front. Earth Sci., 11, 1–13, https://doi.org/10.3389/feart.2023.1233163, 2023.

Burdige, D. J.: Preservation of organic matter in marine sediments: controls, mechanisms, and an imbalance in sediment organic carbon budgets?, Chem. Rev., 107, 467–485, https://doi.org/10.1021/cr050347q, 2007.

Glud, R. N.: Oxygen dynamics of marine sediments, Mar. Biol. Res., 4, 243–289, https://doi.org/10.1080/17451000801888726, 2008.

Henrichs, S. M. and Reeburgh, W. S.: Anaerobic mineralization of marine sediment organic matter: Rates and the role of anaerobic processes in the oceanic carbon economy, Geomicrobiol. J., 5, 191–237, https://doi.org/10.1080/01490458709385971, 1987.

Mengual, B., Cayocca, F., Le Hir, P., Draye, R., Laffargue, P., Vincent, B., and Garlan, T.: Influence of bottom trawling on sediment resuspension in the 'Grande-Vasière' area (Bay of Biscay, France), Ocean Dyn., 66, 1181–1207, https://doi.org/10.1007/s10236-016-0974-7, 2016.

Mouret, A., Anschutz, P., Deflandre, B., Chaillou, G., Hyacinthe, C., Deborde, J., Etcheber, H., Jouanneau, J.-M., Grémare, A., and Lecroart, P.: Oxygen and organic carbon fluxes in sediments of the Bay of Biscay, Deep Sea Res. Part I Oceanogr. Res. Pap., 57, 528–540, https://doi.org/10.1016/j.dsr.2009.12.009, 2010.

O'Neill, F. G. and Summerbell, K.: The mobilisation of sediment by demersal otter trawls, Mar. Pollut. Bull., 62, 1088–1097, https://doi.org/10.1016/j.marpolbul.2011.01.038, 2011.

Ogrinc, N., Faganeli, J., and Pezdic, J.: Determination of organic carbon remineralization in near-shore marine sediments (Gulf of Trieste, Northern Adriatic) using stable carbon isotopes, Org. Geochem., 34, 681–692, https://doi.org/10.1016/S0146-6380(03)00023-8, 2003.

Oguri, K., Masqué, P., Zabel, M., Stewart, H. A., MacKinnon, G., Rowden, A. A., Berg, P., Wenzhöfer, F., and Glud, R. N.: Sediment Accumulation and Carbon Burial in Four Hadal Trench Systems, J. Geophys. Res. Biogeosciences, 127, https://doi.org/10.1029/2022JG006814, 2022.

Sampere, T. P., Bianchi, T. S., Allison, M. A., and McKee, B. A.: Burial and degradation of organic carbon in Louisiana shelf/slope sediments, Estuar. Coast. Shelf Sci., 95, 232–244, https://doi.org/10.1016/j.ecss.2011.09.003, 2011.

Sobek, S., Zurbrügg, R., and Ostrovsky, I.: The burial efficiency of organic carbon in the sediments of Lake Kinneret, Aquat. Sci., 73, 355–364, https://doi.org/10.1007/s00027-011-0183-x, 2011.

van de Velde, S. J., Hylén, A., Eriksson, M., James, R. K., Kononets, M. Y., Robertson, E. K., and Hall, P. O. J.: Exceptionally high respiration rates in the reactive surface layer of sediments underlying oxygen-deficient bottom waters, Proc. R. Soc. A Math. Phys. Eng. Sci., 479, https://doi.org/10.1098/rspa.2023.0189, 2023.

Wenzhöfer, F. and Glud, R. N.: Small-scale spatial and temporal variability in coastal benthic O2 dynamics : Effects of fauna activity, 49, 1471–1481, https://doi.org/https://doi.org/10.4319/lo.2004.49.5.1471, 2004.

---

## Author Response (AR1)

**Reviewer #1**

Muller et al., explores the carbon storage and accumulation in the Helgoland mud area and show high carbon accumulation especially in areas under the direct influence pf the Elbe. The rates of carbon accumulation in these muds is equivalent to those in the Skagerrak and Norwegian Trough two of the key depositional environment in the North Sea. This work highlights that the carbon storage and accumulation potentially of Helgoland mud area have been overlooked and shows the muddy depositional area need to be investigated more closely.

This manuscript will be of interest to a broad audience and after some minor revisions will be ready for publication.

**Author's response:** We would like to thank the reviewer for the thoughtful and constructive feedback on our manuscript.

Line 83 - As part of the BMBF-funded collaborative project APOC (Anthropogenic impacts on particulate organic carbon cycling in the North Sea), this is an acknowledgment and not required in the main text.

**Author's response:** The statement regarding the project funding was suppressed in the main text and is in the acknowledgement. The project name remains in the main text because it is referred to later in the text.

Line 86 -Should be in the methods. If you mention a detailed literature review, it is now expected that you provide search terms etc. in the supp mat. Since section 4.1. of the results is focused on this data compilation more details are required even with the data archived in Pangea.

**Author's response:** Line 86 was moved to the materials and methods section (lines 143-145, lines correspond to the manuscript version without track changes). Additional information on the literature review is now given in the supplementary material.

Line 94 - (3) estimate the carbon budget of this significant depocentre in the German Bight of the North Sea. I agree that the first two objectives have been archived but the third on is a push, to make this stronger focus on objective 1 and 2.

**Author's response:** Thank you for this comment. We reformulated the aim accordingly (line 94-95) and renamed the respective chapter in the discussion: 5.4 Annual organic carbon accumulation in the Helgoland Mud Area.

Line 115 - Although the swept area ratio is low in the HMA the intensity outside is high. Though I suspect impossible to fully quantify how much sediment is resuspended and deposited in the HMA from these activities. Might be useful to mention this process at this point.

**Author's response:** Indeed, a quantification of the impact of bottom trawling and respective resuspension and transport on the deposition of sediment in the HMA is not possible in the framework of this study. However, the overall contribution of bottom trawling on SPM load, transport and deposition is added in lines 126-128.

Line 135 - multiple corer (MUC) should this be multi-corer?

**Author's response:** "multiple corer" (MUC) was changed to "multi-corer" in line 148.

Line 147 - Samples were taken at 1-cm-intervals in the top 10 cm and every second centimetre below. Can you clarify if this is the porewater sampling or the intervals the cores were sliced at. I assume it is the porewater if so add a sentence describing the sediment sampling.

**Author's response:** The intervals are similar for pore-water and solid-phase sampling and clarified in line 159.

Line 187 - Could more details be added to the grain size prep, sample mass, wet or dry, etc.

**Author's response**: We kept the description of the employed methods brief in the manuscript for better overall readability. The description of chemical sample treatment is found in the following line, which we perceived as the most essential bit of information. All samples were measured in water in the instrument's wet cell and no dispersing agents were used. The sample mass varied and was chosen to fit the technical requirements (the instrument uses a relative scale to represent sample concentration/laser attenuation). For coarser samples, we used an approx. volume of up to 10 ml freeze-dried and unprocessed sample. For finer samples, a much smaller volume was used.

Line 189 - Sediment samples were freeze-dried and both, moisture and porosity, were calculated by the mass loss, assuming a sediment grain density (quartz) of 2.65 g cm-3 (e.g., Anderson and Schreiber, 1965). Was the grain size analysis preparation carried out on dry sample. If so move this be moved earlier in the paragraph.

**Author's response:** The sentence was moved to lines 168-169.

Line 225 - Why was the CRS model chosen over the others, please provide clarification.

**Author's response:** The CRS model was chosen since the CFCS is more affected by sediment redistribution due to e.g. bioturbation or physical mixing. Arias-Ortiz et al. (2018) showed that the calculated sedimentation rate using the constant flux, constant sedimentation (CFCS) model deviated by 20 to 95% in a sediment mixing scenario, while the CRS model - using the inventory to calculate the rates - deviated by only 2 to 5%. The statement was added to the materials and methods section in lines 213-216.

Results - The results section provide a good overview of the data and the figures are of high quality and communicate the work well.

**Author's response:** Thank you very much for the comment.

Figure 8 - The IDW should be introduced in the statistical section of the methods.

**Author's response:** IDW was added to the statistical section in line 266-267.

Line 527 - Section heading?

**Author's response:** Line 547 is now numbered as Section 5.2.1.

Figure 12 - were the burial efficiencies calculate in the same manner as this study are they comparable?

**Author's response:** Please find a detailed description regarding this topic in the reply to reviewer #2 (General comment section).

Section 5.4 - A new paper by Diesing et al, 2024 will be useful in further highlighting the importance of the HMA in comparison to the wider north west European shelf. https://www.nature.com/articles/s43247-024-01502-8

**Author's response:** Thank you very much for this suggestion. A comparison was added in lines 675-677.

Line 644 - total organic carbon (TOC), why is this being defined in the conclusion, has it not be used before?

**Author's response:** The abbreviation TOC is now used in line 682.

**Reviewer #2**

**Review of the manuscript egusphere-2024-1632**

Muller et al. aim to estimate the capacity of the Helgoland Mud Area (HMA) to store and recycle organic carbon and the factors controlling the efficiency of this preservation. The study site is particularly interesting to achieve this objective because the different zones of the area are subject to different forcing (e.g., trawling, river inputs, origin of organic matter). The authors finally showed the importance of such systems for organic carbon sequestration by estimating the relative contribution of the HMA to organic carbon burial at the scale of the North Sea.

The manuscript presents an interesting data set to improve the understanding and the budget of organic carbon burial on continental shelves. Overall, it is well written but sometimes confusing. It lacks of organization and some figures are not relevant. Although the data set is interesting, the work has too many deficiencies and inaccuracies to be suitable for publication in Biogeosciences. Moreover, it lacks of comparisons with similar environments to be of an international interest.

**Author's response**: We would like to thank the reviewer for the detailed and insightful comments, which helped to improve the manuscript. We have revised the introduction and the discussion (see lines 640-653 and 675-677, lines correspond to the manuscript version without track changes) to better highlight the implications of our study and its broad international relevance. However, we do not agree with the statement that our paper/study is mainly of a regional interest. As described in our manuscript, to our knowledge, there is no study available so far that has performed a detailed assessment of the factors that control the burial/preservation of organic matter (OM) in fine-grained sediments of the North Sea. We have therefore used the Helgoland Mud Area, which is characterized by a broad range of natural depositional conditions (water depth, sedimentation rates, origin/reactivity of organic matter, etc.) - as a test field or natural laboratory, respectively, to determine the key factors that control OM burial in the North Sea. Only in this way we were able to determine the key factors in the required – and at the same time unprecedented - high spatial resolution – including the extremely time-consuming radiometric analyses ($^{210}$Pb, $^{226}$Ra, $^{137}$Cs). In this respect, the results obtained in the framework of this study present key findings for the whole North Sea as well as for shelf seas in general/globally– as these represent the most important seafloor areas for the longterm burial of organic carbon and, thus, key natural sinks for $CO_2$. Moreover, the North Sea is a crucial area in Europe due to its important role in economy (e.g., offshore energy, fishing activities, navigation) and ecology. Our results are not only important for better a understanding of carbon fluxes and burial in the North Sea, with important implications for estimations of European blue carbon wealth, but also provide key process-understanding that can be applied to all mud belts on continental shelves worldwide.

We thank the reviewer for the suggestion to extend the comparison with similar environments, in order to discuss and complement the global comparison. We included and discuss more studies in similar environments (lines 640-653 and 675-677) in addition to the global comparison and Figure 12.

**General comments**

Reviewer: The paper is mainly of a regional interest and I think that an entire section of the discussion would have be devoted to a comparison with other environments. In the Figure 12,

the burial efficiencies calculated in the HMA are compared with those measured in other marine environments. This comparison is interesting but too general. Indeed, the burial of organic carbon on continental shelves has been studied for some decades and several studies have quantified it in environments that are morphologically similar to the HMA. A comparison with such systems would have been relevant to complement the global comparison presented in the Figure 12.

**Author's response**: Thank you for this suggestion. Indeed, extending the comparison to similar environments helped to highlight the global relevance of this study. We included more studies in similar environments (lines 640-653 and 675-677) in addition to Figure 12.

Reviewer: Another major concern is related to the estimates of organic carbon accumulation rates and burial efficiencies. Firstly, TOC accumulation rates are estimated from a mean TOC value for each site. However, such estimates are usually made using either surface organic carbon contents to calculate a part of organic carbon inputs to the sediments or organic carbon contents measured at the core bottom to estimate organic carbon burial rates.

[Figure]

Figure 1: TOC contents (orange squares) of (a) the interval used in the study and the bottom 5 cm of the core, (b) the interval used in the study and the uppermost 5 cm of the core and (c) the bottom 5 cm of the core and the uppermost 5 cm of the core. The black line is the 1:1 relation.

**Author's response**: We can demonstrate that the TOC contents show no or only little variation over depth and remain almost constant, (Fig. 1a-c). Within the natural variability, we see no clear/systematic deviation from the 1:1 line. Therefore, we consider the values presented in the manuscript and the following calculations using these values to be sound.

Reviewer: Secondly, the organic carbon inputs, used to estimate organic carbon burial efficiencies, are calculated as the sum of aerobic respiration and organic carbon burial rates. For this calculation, it is not clearly indicated which values of TOC are used (i.e., the bottom value, an average of values below the oxygen penetration depth, an average of the last points of the profiles). In addition, such calculations are typically made by summing organic carbon burial rates, calculated from bottom TOC contents, and total organic carbon remineralization rates. However, aerobic respiration rates do not represent the total benthic remineralization. The data show that aerobic respiration is limited to the first 1-2cm of sediment. However, the degradation of organic matter continues below this surface layer as indicated by the profiles of DIC shown in the Figure 3. In addition, since the relative importance of the different pathways of organic matter remineralization is not presented, there are no data to justify that aerobic respiration is the main pathway for organic matter remineralization in these sediments. Accordingly, the values of organic carbon burial efficiencies and the subsequent interpretations are not reliable.

**Author's response**: As outlined in the Materials and Methods chapter we calculated the organic carbon burial efficiency as described by Burdige (2007) and van de Velde et al. (2023): we assume aerobic respiration to be responsible for most of the total organic carbon remineralisation since no further decrease in TOC over depth can be seen in our data (see Fig. 1). This is an approach similar to the one in Mouret et al. (2010), Sobek et al. (2011) and Oguri et al. (2022). It is correct, that we did not consider anaerobic remineralisation independently. However, previous studies have demonstrated that in depositional settings characterised by oxic bottom-water conditions, aerobic respiration is the dominant pathway of organic matter degradation. Moreover, diffusive oxygen uptake was shown to indirectly also include anaerobic degradation pathways because part of the oxygen is consumed by oxidation of reduced reaction products liberated into pore water as a consequence of anaerobic mineralization pathways (e.g., Wenzhöfer and Glud, 2004; Glud, 2008 and references therein). As reviewed by Glud (2008), this allows to use oxygen consumption rates as a reasonable estimate for organic carbon mineralisation. We have quantified the possible offset by the fraction of anaerobic remineralisation and considered it justified to use our present data for comparison of OC BE within the HMA and with literature data, in particular as there is no standard approach (for various approaches see Henrichs and Reeburgh, 1987; Aller, 1998; Ogrinc et al., 2003; Sobek et al., 2011; Sampere et al., 2011; Baloza et al., 2022). In our opinion, the application of a full diagenetic model to address anaerobic remineralisation processes exceeds the scope of this study and is subject of a follow-up manuscript currently in preparation.

Reviewer: Finally, the efficiency of TOC preservation is partly discussed by comparing the mean TOC contents of different stations. As it is indicated that the area is fueled with particles of different origins, this approach is not reliable.

**Author's response**: It is correct that (1) we compare the TOC contents and therefore the preservation of OC in the study area and (2) that different sources deliver OC to different areas within the HMA. However, this is exactly what we discuss: the difference in origin, and hence reactivity, is - besides the oxygen exposure time/sedimentation rate - a key factor for the preservation and burial (efficiency) of OC on shelf sediments.

Reviewer: For example, the influence of bottom trawling on organic carbon preservation is estimated by comparing TOC contents at three stations (W, Cdeep, NW). However, the three sites have different grain-size and the site with the lowest TOC contents is the one where the sediments are coarser (D50~58μm versus 26-30μm for the others). Therefore, it is likely that the difference in TOC contents is due to the difference in grain size rather than to an influence of bottom trawling on organic matter remineralization.

**Author's response**: Thank you for this comment. We would like to point out that the comparison of the 3 stations in light of bottom trawling is a side aspect in the discussion where we merely provide evidence for an impact in context of the current literature. Regarding the reviewer's argument, we agree that the grain sizes indeed differ between the sites (coarser at the frequently trawled site, finer at the other two). However, we think this is no reason not to evaluate the impact of bottom trawling on the preservation of OC at these sites. Bottom trawling not only enhances the oxygen exposure time and hence aerobic remineralisation but also causes resuspension of the sediments, preferentially affecting fine particles as shown by e.g. O'Neill and Summerbell (2011). Thus, bottom trawling reduces the resilience of the sediment to resuspension (Bruns et al., 2023). Recurrent bottom trawling at the same site and natural events will then lead to increased sediment remobilization (Bruns et al., 2023) and result in coarser residual sediments (Mengual et al., 2016). We added this aspect in this section accordingly, to make this point clear (lines 569-572).

**Specific comments**

Title: I suggest to replace "test field" by "case study" or to modified the title such as: "Depositional controls and budget of organic carbon burial in fine grained sediments of the the Helgoland Mud Area (North Sea)"

**Author's response**: The title has been changed from "Depositional controls and budget of organic carbon burial in fine-grained sediments of the North Sea – the Helgoland Mud Area as a test field" to "Depositional controls and budget of organic carbon burial in fine-grained sediments of the North Sea – the Helgoland Mud Area as a natural laboratory" to reflect the intention of this study to use the most important depocentre in the German Bight as a representative natural laboratory to assess of the factors controlling the organic carbon burial in fine-grained coastal and shelf sea sediments.

l. 20 – « lowest TOC contents ». Precise the range of values between brackets.

**Author's response**: The TOC content ranges from 0.7 – 1.0 wt% in the shallow eastern HMA and has been added in line 21.

l. 55 – 57 – The authors only mentioned the free-energy yield of aerobic respiration to justify that it is a key process for organic carbon preservation. However, some labile compounds are as efficiently degraded in oxic as in anoxic conditions. The sentence should be modified to precise that the effect of oxygen exposure time is mainly relevant for refractory organic compounds.

**Author's response**: An addition to the statement has been made in line 59.

l. 58: The bottom trawling should be mentioned as a process that could enhance the exposure time of organic matter to dissolved oxygen.

**Author's response:** Bottom trawling has been added as a process enhancing oxygen exposure time in lines 61-64.

l. 62 - 65: The sentence from "Natural sediment" to "prolonged aerobic respiration" develops the same idea that the sentence at the beginning of the paragraph. The paragraph should be revised to avoid repetitions.

**Author's response:** Thank you for this comment. The paragraph was revised accordingly.

l. 75 – 77 : The sentence "In these cohesive sediments diffusion is the dominant transport process and oxygen only penetrates a few millimetres into the sediments, leading to the establishment of anoxic conditions at shallow sediment depth that enhance the build-up of OM" develops an idea with no links with the previous and following sentences and so confuses the paragraph. I recommend to suppress this sentence to clarify the text.

**Author's response:** Thank you for this comment. The statement regarding the build-up of OM in fine-grained sediments has been revised and contextualised to clarify the text (lines 76-79).

l.79: "these seafloor habitats characterized by high organic carbon burial efficiencies". Precise the typical range of burial efficiencies observed in these environments to justify this affirmation.

**Author's response:** The statement was reformulated in the text (lines 80-83).

l.100: Precise the tidal range and the mean significant wave height to justify that the area is subject to a high hydrodynamics.

**Author's response:** The tidal range and significant wave heights have been added in the text. Lines 101-104 read now: The southern North Sea is characterised by high tidal and wave energy regimes with a tidal range of ~ 1.5 to 3 m and significant wave height during storm events up to 5 m in the German Bight (Hagen et al., 2021). Seafloor processes are mostly characterised by local transport and resuspension of material in sand-rich sedimentary environments (Figge, 1981; de Haas et al., 1996; Zeiler et al., 2000; de Haas et al., 2002).

On the Figure 1a, the blue point representing the dumping site is not well visible because the bathymetry is also represented in blue.

**Author's response:** The colour of the dot representing the dumping site in Fig. 1 was changed to orange for better visibility.

In the introduction, it is indicated that sedimentation rates have already been estimated in the studied area. The range of mean sedimentation rates previously estimated has to be presented in the section "Study area".

**Author's response:** The range of previously estimated sedimentation rates and the respective methods have been added to the section "Study area" (lines 117-124).

l. 113: Precise the discharge of the Elbe and Weser rivers to give an idea of their potential importance on sediment inputs.

**Author's response:** The discharge was of the Elbe and Weser rivers was added in lines 115-116.

l. 123-124: The reason of the deposition of fine sediments in the study area is discussed with no clear conclusions. These sentences should be revised based on recent literature. For example, Walsh and Nittrouer (2009) have detailed several types of fine-grained depositional environments influenced by rivers and presented systems with the same morphology of the HMA.

**Author's response:** Due to the complex hydrodynamic situation in the German Bight, the HMA cannot be classified according to the classic estuarine and shelf deposition presented by e.g., Walsh and Nittrouer (2009). However, this is an ongoing research topic within the project APOC. The statement has been reformulated (lines 134-135).

l.134-135: Precise what are the different depositional environments studied, at least in brackets.

**Author's response:** The different depositional environments have been added in brackets in lines 146-147.

It is not necessary to precise the place where the analyses were performed. This makes the text heavy. The description of the methods is enough.

**Author's response:** These parts of the text have been suppressed in the method section.

There is no indication that oxygen concentrations were monitored during the measurements of oxygen profiles. Was the bottom water saturation at 100%? If not, the oxygen saturation in the overlying water of the core has to be controlled/monitored to ensure that *in situ* conditions are reproduced.

**Author's response:** Indeed, bottom water oxygen concentrations were close to saturation (CTD data, not shown) and bottom water oxygen contents averaged at 94,8 % at the sediment-water interface, with two outliner measurements at 65 % and 79 % at sites S and E, respectively. Bottom water oxygen saturation has been added to the text (lines 377-379).

l. 184: Please add a symbol on the Figure 1a to highlight the stations selected for grain size analyses. However, it is quite difficult to visualize the transects.

**Author's response:** As requested, a table showing for which sites grain size analyses (a.o.) were carried out has been added at the beginning of the results section. Furthermore, the sites for which grain size measurements were performed are shown in Figure 8b. In order to maintain clarity of Fig. 1a, no additional information was added to Fig. 1.

It is indicated that the samples for particulate analyses were freeze-dried (l. 189, l. 200) but they were stored at 4°C and not frozen (l. 152-153).

**Author's response:** Sediment sample aliquots were frozen before freeze-drying. This was added in line 168.

Precise the reproducibility of radionuclides measurements between the gamma and the alpha detector.

**Author's response:** The reproducibility of the radionuclide measurements between gamma and alpha spectrometry was evaluated for four samples from all three cores analysed by alpha spectrometry. All cross-measurements agree within 1 sigma uncertainty (Fig. 2). An addition has been made to the text in lines 208-210.

[Figure]

Figure 2: Reproducibility measurements between alpha and gamma spectrometry. The red dashed lines are the 1:1 relation and 0 Bq kg$^{-1}$.

l. 230-234: The reproducibility of TOC measurements has to be precise.

**Author's response:** The reproducibility of the TOC measurements was added to the manuscript: Based on replicate analyses of two different standards, the precision (relative standard deviation in percentage) was 1.23 % for the standard of 0.99 wt% (n=15), and 1.34 % for the standard 5.00 wt% (n=7) (lines 224-225).

l. 250-253: This has already been indicated in the introduction. As the sentences do not bring additional information, they can be suppressed.

**Author's response:** We decided to keep this paragraph in the results since this section provides the results of the literature survey in more detail (in which time frame, number of studies/cores). As Reviewer #1 suggested, line 86 has been moved to the materials and methods section (lines 143-145) and more details were added to the supplementary material.

I recommend placing the Figure 2 in supplementary material. Moreover, a table synthetizing the parameters measured at each station would be useful because all the parameters were not measured at all sites.

**Author's response:** Figure 2 was moved to the supplementary material. Table 2 shows now the overview of all parameters determined at each site.

The Results section is confusing partly because of the order in which the data are described. I recommend starting with a description of the physical properties of the sediments, then the age models, then the TOC contents and finally the origin of organic matter and aerobic remineralization.

**Author's response:** The Results sections have been reorganised accordingly. The Materials and Methods sections have also been reorganised to ensure consistency.

Rather than presenting oxygen profiles in supplementary materials, it would be more relevant to make a synthetic figure with the profiles and to include it in the results section.

**Author's response:** A synthetic figure (Fig. 7) of the oxygen profiles has been added to the results section.

Table 2: Explicit the unit of porosity because porosity usually has no unity.

**Author's response:** Porosity has no unit, it is given as rel. = relative. It has been changed to (-) (Table 3).

Figure 5: Explain more clearly the reasons why there are subsurface $Pb_{xs}$ maximums on several cores.

**Author's response:** Please, see the detailed comment below in which we also discuss this topic.

The profiles of $^{210}Pb_{xs}$ are sometimes noisy or present several maximums (e.g., N, W, Cdeep, SC, SE). Some anomalies may be related to grain size changes. Accordingly, the profiles of grain-size should be presented next to the $^{210}Pb_{xs}$ profiles or, if they are not available, the profiles of porosity. Due to these anomalies that are not explained in the manuscript, it would be better to estimate a sedimentation rate from a linear regression of the $^{210}Pb_{xs}$ profiles and to

adjust the model thanks to Cs data rather than estimating sedimentation rates according to ages as it is done in the Figure 6. Indeed, these sedimentation rates likely integrate the noise of the profiles and do not represent realistic variations of the sedimentation. If the sedimentation is fairly constant, the linear regression give robust sedimentation rates that could be extrapolated on the entire core.

**Author's response:** Thank you for this comment, it shows that our careful assessment of different age models was not completely clear in the text. As requested, the porosity and grain size profiles are now shown next to the radionuclides (Fig. 2).

To the pattern of $^{210}Pb_{xs}$ and subsurface maxima: It is true, that the profiles are somehow noisy and show subsurface maxima. The small-scale maxima (e.g., site W at 19.5 and 25.5 cm) could be the result of storm events as described by van der Zee et al. (2003) using $^{210}Pb$ data and an X-radiograph. The upper subsurface maxima - close to the sediment surface - is most likely a sediment-mixing signal. Arias-Ortiz et al. (2018) describe such $^{210}Pb_{xs}$ profiles in detail. The profiles can be altered by sediment mixing ("backwards bend" at the top of the sediments, similar activity over a mixing interval or a linear rather than exponential pattern), or be a result of rapid MAR (mass accumulation rate) changes (added in line 213-216 and line 397-399). Since various studies indicate sediment mixing in the German Bight/HMA (Hintzen et al., 2012; Oehler et al., 2015; Wrede et al., 2017; Thünen Institute, 2018) we attribute these changes to sediment mixing rather than rapid changes in MAR.

To the calculation of the sedimentation rate: The underlying truth/assumption of any age model is to assign a certain age to a certain depth in the sediment and applies to both the CRS and the CFCS. The calculations performed here follow the compilation of formulas presented by Sanchez-Cabeza and Ruiz-Fernández (2012). For the CRS in short: an age is assigned to every layer and the cumulated mass is calculated, delivering a respective MAR. Using the MAR and porosity the sedimentation rate is calculated (for more details see Sanchez-Cabeza and Ruiz-Fernández, 2012).

To the suggestion to use a linear regression model: The linear regression model assumes constant flux and constant sedimentation (CFCS). Arias-Ortiz et al. (2018) quantified the impact on age models/sedimentation rates of different processes altering $^{210}Pb_{xs}$ profiles. They showed that the calculated sedimentation rate using the CFCS model deviates by 20 to 95 % in a sediment mixing scenario, while the CRS (constant rate of supply, used in this study) using the inventory to calculate the rates deviated only by 2 - 5 %. This was added to the text in lines 213-216. Since we assume mixing to be a relevant process in the HMA, we chose the CRS model and used the $^{137}Cs$ as an independent time marker, rather than fixing our model to the $^{137}Cs$ activity. In our case, at site W (affected by mixing), the CFCS model deviates by ~15 % from the CRS model.

To the integration of the "noisy" signal: We quantified the impact for site W with the two subsurface maxima at 19.5 and 25.5 cm. If we use an interpolation excluding those higher activities the sedimentation rate using the CRS age model changes by only 3 % (or 0.1 mm yr$^{-1}$) and can thereby be neglected. This has been added in lines 310-312.

l. 382-384: Describe the typical shape of the profiles affected by mixing.

**Author's response:** Examples of $^{210}Pb_{xs}$ profiles affected by sediment mixing are now given in lines 407-409.

l. 394-395: It is indicated that no prediction on changes on sedimentation rates can be made with the method used. However, it is what it is represented in the Figure 6. According to the discussion, I recommend to remove the Figure 6. The values of interest are already presented in the Table 3.

**Author's response:** It is true that it is not possible to describe changes in sedimentation rates without over-interpreting the system. Nevertheless, Fig. 3 provides a substantial contribution to this insight, as the characteristic changes in sedimentation rate at the top of the core further indicate sediment mixing. Therefore, Fig. 3 is crucial for interpretation/the argument, whether changes in sedimentation rates, or rather sediment mixing is present.

l. 365-403: These paragraphs would be move in the material and methods or results sections. Indeed, while they could clarify the section 4.5 they confuse the discussion.

**Author's response:** Thank you for this comment. We clarified the section by transferring additional information to the materials and methods section and streamlining this part of the discussion. Section 5.1 is now more focussed on the discussion of our data and general statements are suppressed. However, the interpretation of sedimentation rate changes can only be performed once the data are created. We think that the discussion of whether sedimentation rates change over time or sediments are mixed is a crucial point of the discussion/interpretation of our data.

The interpolation of the Figure 8b is too extensive toward the north and the south where there are no data point.

**Author's response:** Thank you for this comment. The interpolation was reduced in the north and south (Fig. 8b).

l. 478: It is mentioned the potential occurrence of an advective transport at some stations. However, it is inconsistent with the use of a diffusive model to estimated fluxes of dissolved oxygen at the sediment-water interface from oxygen profiles. In such a case, permeability measurements have to be presented to justify the use of this method to estimate oxygen fluxes

**Author's response:** Thank you for this comment. We corrected the statement regarding the pore-water mixing processes (lines 496-497). The largest $D_{50}$ grain size is $\sim 60$ µm (Site E). Using the empirical relation of permeability and median grain size for North Sea sediments presented by Neumann et al. (2017a) permeabilities of our sediments are below $10^{-14}$ m$^2$, which are classified as impermeable sediment (Neumann et al., 2017b; Ahmerkamp et al., 2017). Therefore, the diffusive model used to estimate the oxygen fluxes is valid.

On the Figure 9, the significant correlation between MAR and TOC seems mainly related to one extreme point (I supposed the station SC). Is the correlation still significant without this extreme point? Moreover, the linear relationship between the parameters was tested using a Pearson correlation without indicating if the data follow a normal distribution. This has to be indicated prior performing a parametric analysis. Otherwise, the correlation has to be tested with a non-parametric analysis like a Spearman correlation.

**Author's response:** The relation of MAR and TOC (Fig. 9) is neither in the Figure (orange) nor in the text stated to be significant (cor = 0.37, p = 0.26, Fig. 9a, line 503).
However, for all correlations/regressions, the normal distribution of residuals was tested using the Kolmogorov-Smirnov test, showing that all residuals of the linear regressions follow a

normal distribution. To clarify this, the Materials and Methods section was edited (lines 261-262).

l. 529: The title of the section has no number.

**Author's response:** The section number was added in line 546.

l. 550: The section is numbered 5.2.1 but there is no section 5.2.2.

**Author's response:** According to the suggestion above, section "5.2.1" was changed to "5.2.2" (line 574).

l. 521-564: Variations in TOC contents may also be related to grain size changes. It would be relevant to discuss this point in this paragraph, which is rather speculative.

**Author's response:** A detailed comment regarding this topic is given in the general comment section above. A statement regarding the differences in grain sizes has been added in lines 569-572.

**Technical corrections**

l. 147: Replace "second centimeter" by "two centimeters"'

**Author's response:** "second centimetre" was changed to "two centimetres" in line 159.

l. 189: Use "water content"' rather than "moisture"

**Author's response:** "Moisture" was changed to "water content" in line 168.

**References:**

Ahmerkamp, S., Winter, C., Krämer, K., de Beer, D., Janssen, F., Friedrich, J., Kuypers, M. M. M., and Holtappels, M.: Regulation of benthic oxygen fluxes in permeable sediments of the coastal ocean, Limnol. Oceanogr., 62, 1935–1954, https://doi.org/10.1002/lno.10544, 2017.

Aller, R. C.: Mobile deltaic and continental shelf muds as suboxic, fluidized bed reactors, Mar. Chem., 61, 143–155, https://doi.org/10.1016/S0304-4203(98)00024-3, 1998.

Arias-Ortiz, A., Masqué, P., Garcia-Orellana, J., Serrano, O., Mazarrasa, I., Marbà, N., Lovelock, C. E., Lavery, P. S., and Duarte, C. M.: Reviews and syntheses: 210Pb-derived sediment and carbon accumulation rates in vegetated coastal ecosystems – setting the record straight, Biogeosciences, 15, 6791–6818, https://doi.org/10.5194/bg-15-6791-2018, 2018.

Baloza, M., Henkel, S., Geibert, W., Kasten, S., and Holtappels, M.: Benthic carbon remineralization and iron cycling in relation to sea ice cover along the eastern continental shelf of the Antarctic Peninsula, J. Geophys. Res. Ocean., 127, https://doi.org/10.1029/2021JC018401, 2022.

Bruns, I., Bartholomä, A., Menjua, F., and Kopf, A.: Physical impact of bottom trawling on seafloor sediments in the German North Sea, Front. Earth Sci., 11, 1–13, https://doi.org/10.3389/feart.2023.1233163, 2023.

Burdige, D. J.: Preservation of organic matter in marine sediments: controls, mechanisms, and an imbalance in sediment organic carbon budgets?, Chem. Rev., 107, 467–485, https://doi.org/10.1021/cr050347q, 2007.

Glud, R. N.: Oxygen dynamics of marine sediments, Mar. Biol. Res., 4, 243–289, https://doi.org/10.1080/17451000801888726, 2008.

Henrichs, S. M. and Reeburgh, W. S.: Anaerobic mineralization of marine sediment organic matter: Rates and the role of anaerobic processes in the oceanic carbon economy, Geomicrobiol. J., 5, 191–237, https://doi.org/10.1080/01490458709385971, 1987.

Hintzen, N. T., Bastardie, F., Beare, D., Piet, G. J., Ulrich, C., Deporte, N., Egekvist, J., and Degel, H.: VMStools: Open-source software for the processing, analysis and visualisation of fisheries logbook and VMS data, Fish. Res., 115–116, 31–43, https://doi.org/10.1016/j.fishres.2011.11.007, 2012.

Mengual, B., Cayocca, F., Le Hir, P., Draye, R., Laffargue, P., Vincent, B., and Garlan, T.: Influence of bottom trawling on sediment resuspension in the 'Grande-Vasière' area (Bay of Biscay, France), Ocean Dyn., 66, 1181–1207, https://doi.org/10.1007/s10236-016-0974-7, 2016.

Mouret, A., Anschutz, P., Deflandre, B., Chaillou, G., Hyacinthe, C., Deborde, J., Etcheber, H., Jouanneau, J.-M., Grémare, A., and Lecroart, P.: Oxygen and organic carbon fluxes in sediments of the Bay of Biscay, Deep Sea Res. Part I Oceanogr. Res. Pap., 57, 528–540, https://doi.org/10.1016/j.dsr.2009.12.009, 2010.

Neumann, A., Möbius, J., Hass, H. C., Puls, W., and Friedrich, J.: Empirical model to estimate permeability of surface sediments in the German Bight (North Sea), J. Sea Res., 127, 36–45, https://doi.org/10.1016/j.seares.2016.12.002, 2017a.

Neumann, A., van Beusekom, J. E. E., Holtappels, M., and Emeis, K.-C.: Nitrate consumption in sediments of the German Bight (North Sea), J. Sea Res., 127, 26–35, https://doi.org/10.1016/j.seares.2017.06.012, 2017b.

O'Neill, F. G. and Summerbell, K.: The mobilisation of sediment by demersal otter trawls, Mar. Pollut. Bull., 62, 1088–1097, https://doi.org/10.1016/j.marpolbul.2011.01.038, 2011.

Oehler, T., Martinez, R., Schückel, U., Winter, C., Kröncke, I., and Schlüter, M.: Seasonal and spatial variations of benthic oxygen and nitrogen fluxes in the Helgoland Mud Area (southern North Sea), Cont. Shelf Res., 106, 118–129, https://doi.org/10.1016/j.csr.2015.06.009, 2015.

Ogrinc, N., Faganeli, J., and Pezdic, J.: Determination of organic carbon remineralization in near-shore marine sediments (Gulf of Trieste, Northern Adriatic) using stable carbon isotopes, Org. Geochem., 34, 681–692, https://doi.org/10.1016/S0146-6380(03)00023-8, 2003.

Oguri, K., Masqué, P., Zabel, M., Stewart, H. A., MacKinnon, G., Rowden, A. A., Berg, P., Wenzhöfer, F., and Glud, R. N.: Sediment Accumulation and Carbon Burial in Four Hadal Trench Systems, J. Geophys. Res. Biogeosciences, 127, https://doi.org/10.1029/2022JG006814, 2022.

Sampere, T. P., Bianchi, T. S., Allison, M. A., and McKee, B. A.: Burial and degradation of organic carbon in Louisiana shelf/slope sediments, Estuar. Coast. Shelf Sci., 95, 232–244, https://doi.org/10.1016/j.ecss.2011.09.003, 2011.

Sanchez-Cabeza, J. A. and Ruiz-Fernández, A. C.: 210Pb sediment radiochronology: an integrated formulation and classification of dating models, Geochim. Cosmochim. Acta, 82, 183–200, https://doi.org/10.1016/j.gca.2010.12.024, 2012.

Sobek, S., Zurbrügg, R., and Ostrovsky, I.: The burial efficiency of organic carbon in the sediments of Lake Kinneret, Aquat. Sci., 73, 355–364, https://doi.org/10.1007/s00027-011-0183-x, 2011.

Thünen Institute: Fine-scale footprint of bottom trawling in the German EEZ of the North Sea (2012-2016), https://hub.hereon.de/server/rest/services/NOAH_geoDB/TI_SARnested/MapServer, 2018.

van de Velde, S. J., Hylén, A., Eriksson, M., James, R. K., Kononets, M. Y., Robertson, E. K., and Hall, P. O. J.: Exceptionally high respiration rates in the reactive surface layer of sediments underlying oxygen-deficient bottom waters, Proc. R. Soc. A Math. Phys. Eng. Sci., 479, https://doi.org/10.1098/rspa.2023.0189, 2023.

Walsh, J. P. and Nittrouer, C. A.: Understanding fine-grained river-sediment dispersal on continental margins, Mar. Geol., 263, 34–45, https://doi.org/10.1016/j.margeo.2009.03.016, 2009.

Wenzhöfer, F. and Glud, R. N.: Small-scale spatial and temporal variability in coastal benthic O2 dynamics : Effects of fauna activity, 49, 1471–1481, https://doi.org/https://doi.org/10.4319/lo.2004.49.5.1471, 2004.

Wrede, A., Dannheim, J., Gutow, L., and Brey, T.: Who really matters: influence of German Bight key bioturbators on biogeochemical cycling and sediment turnover, J. Exp. Mar. Bio. Ecol., 488, 92–101, https://doi.org/10.1016/j.jembe.2017.01.001, 2017.

van der Zee, C., van Raaphorst, W., Helder, W., and de Heij, H.: Manganese diagenesis in temporal and permanent depositional areas of the North Sea, Cont. Shelf Res., 23, 625–646, https://doi.org/10.1016/S0278-4343(03)00024-4, 2003.

---

## Referee Report (RR1)

**Review** of the manuscript *Depositional controls and budget or organic carbon burial I  fine-grained sediments of the North Sea – the Helgoland Mud Area as a natural laboratory* , by Müller et al, submitted to **Biogeosciences** (egusphere-2024-1632).

**Manuscript overview**

The manuscript presents the results of an extensive sediment core analysis in the Helgoland Mud Area, with the aim of quantifying the burial efficiency of organic carbon in the area. This includes presentation per site of porosity, grain size and mud content data, as well as Dissolved Inorganic Carbon (DIC) pore water profiles and Total Organic Carbon (TOC) and $O_2$ sedimentary profiles and isotope analysis. Derived from these data are sediment age profiles, aerobic remineralisation rates, sedimentation rate, mass accumulation rate, total organic carbon accumulation rate, organic carbon burial efficiency and the source of the organic material (marine or terrestrial).

The presented data covers 14 sites southeast of Helgoland in the German Bight (North Sea, European Shelf), of which cores were taken in spring 2022 (ranging from mid-March to the beginning of April). Data from a 2021 cruise to the same area (end of April) is also used, covering 3 stations of which 2 were resampled in 2022. The data has been made publicly available by the authors. They conclude that sedimentation rate is the main factor driving the burial efficiency and that there is a south-north gradient in the source of the organic material, with the southern part characterised by less degradable material from terrestrial/riverine sources. They also conclude that the area stores a higher percentage of carbon for the North Sea than its size alone would merit.

**Review overview**

First of all I would like to state that I am not an expert on the applied methods, and that I realise the paper has been reviewed before. It is therefore likely that some of my comments apply to text changed on behalf of other reviewers' comments. I did not consider the other reviews comments in the generation of this review report.

The paper follows the standard methods, results, conclusions outline, yet core elements such as the physical and biological environment are not mentioned until late in the manuscript. Without these interpretation of the presented results is difficult. For instance, only at the end do the authors state that the area sediments are classified as impermeable (line 496) and that the sampling occurred before the spring bloom (line 600). No evidence is given for the latter statement, and as some sites were sampled in April I would like to see some confirmation for this. Tian et al (2011) show that the Helgoland Roads data indicate a spring bloom start (varying with the physical conditions) between the end of March and the first two weeks of April. They considered only 2002-2005 data, but Amorim et al (2024) show positive trends in satellite-derived Chla observations in the study area (and elevated Chla values in March in part of the area) over the 1998-2020 period. In their figure 8 full bloom dynamics occur in April, so any assertion that the reported cruises were before the spring bloom (sample dates cover 17-03 to 29-04) should be substantiated (the $O_2$ profiles indicate rapid consumption at the core top). As this is important for the derived sedimentation rates and origin of organic material, information regarding this should be included in the site description at the beginning. This also applies to the variability (seasonal, interannual) in riverine discharges to the area (Elbe and Weser) for suspended particular matter (SPM), dissolved inorganic carbon (DIC) and dissolved organic carbon) DOC: discharges have been declining, how does this affect the results in this paper? See for instance Patsch (2024), Schultz (2023) and Van Beusekom (2019).

Apart from the physical and biological setting the authors omit to consider interannual variability (river discharges, stratification and onset of spring bloom, circulation patterns), and state that there is ongoing discussion about the depositional mechanism in the area which will be the subject of a further

study. I would argue that at least a physical discussion should be part of this manuscript, as the depositional mechanism determines the sedimentation rates, origin of organic material and thus burial efficiency. The wealth of data presented here will be important for that discussion, but for a true interpretation of the gathered observational data the physical setting remains key. Yet the current manuscript does not contain a detailed hydrodynamic overview or an analysis of riverine loads (SPM, carbon compounds) to the area (and their interannual variability, e.g. in years with high riverine discharge the terrestrial matter will reach a larger part of the area and push the coastal current more offshore). A lot of work has gone into this study and the results certainly merit publication, but they should be presented within context. Now the manuscript contains some bold statements (overestimation by de Haas et al (1997)) and many suggestions (in area 1 the main driver for mixing is benthos and in area 2 it is very active benthos) without much evidence beyond the direct measurements. For the manuscript it would be better if these suggestions were more substantiated. Alternatively, the authors could wait with publication until their further studies are concluded and publish a double paper, but funding agencies usually want to see intermediate results. More detailed comments are listed below.

**Recommendation**

Moderate revision (no new analysis or figures needed but textual changes required)

**Detailed Comments**

1.  Abstract: The abstract amounts to 28 lines of text while the Conclusions only hold 16 lines. This is not an abstract, this is an Introduction. A true abstract is short while capturing the set-up of the work and the main take home messages.

2.  Introduction: I miss an overview here of shelf processes which store carbon long-term, and a reference to studies that examine them like Legge et al (2020). This would provide more context for the presented work: for the North Sea area, how much carbon is stored in the sediments compared to for instance off-shelf transport? And therefore, how much does this particular site add to this on-shelf storage?

3.  Line 64: "*Sedimentation rate is one of the most important factor controlling the preservation of OM in the sediment*" is followed by 4 references. But this is also presented as a conclusion of this manuscript in line 24 (albeit as simple the most important factor). I think the authors should make it more clear that their work confirms the earlier work with respect to the process controlling carbon burial.

4.  Lines 76-83: fine coastal sediments may have a low $O_2$ penetration depth but they are also characterised by high anthropogenic activity (dredging/dumping, shipping, fishing/trawling, offshore renewable energy, cabling) and biotic activity (more nutrients close to the coast). I miss a discussion on how for instance increased use by offshore wind farms will affect the processes described here or their interpolation to the larger area. It doesn't have to be in-depth but any extrapolation to the North Sea scale should mention this at least.

5.  Lines 91-95: The analysis is focussed on the HMA, which is by no means a common environment type for the North Sea, as shown by the referenced Bockelmann et al (2018). So I don't quite see why the authors claim to determine the main depositional drivers for carbon burial in the *North Sea*.

6.  Figure 1: The abbreviation MUC should be explained in the caption. And I would prefer more detail in the bathymetry and possibly a different colour scheme) so that the Elbe channel is visible in more detail.

7. Line 102: the reference here is to Hagen et al (2021), but this study solely relates to the German Bight, not to the whole of the Southern North Sea.

8. Line 134: why can't the site be classified as such? At the very least an indication of the complex hydrodynamic conditions should be provided here, to sketch the situation and aid interpretation of the results. This should also include a few lines on biotopes (indicating biological activity on or in the sea bed), riverine influences and interannual variability. How far does the influence of the Elbe and Weser reach in general? This could be indicated by salinity gradients over the area or by studies such as Lenhart & Große (2018). Does the muddy Ems influence this region at all? How about the storm floods, did this increase terrestrial material input to the area? The storms are now mentioned at the start of the methods section, for me it would be more logical if presented here with the site overview. I realise the authors use a steady-state model for the derivation of some quantities, but the marine environment is not steady state and the direct observations require an indication of the dynamic setting.

9. Table 1: can the trawling pressure from Figure 1b be included here for each station? Because it seems to me that in the figure site W has a higher (or equal) trawling activity than site NW, but the text later claims that W has a lower trawling activity than NW (line 556). Or are you referring to the proximity of higher trawled areas? In that case I would like to see residual current patterns for the area.

10. Line 170: what language is the GRADISTAT application in? Python, R, Matlab, IDL, Julia, …?

11. Line 210: "*1 sigma uncertainty*", do you mean one standard deviation?

12. Line 253: the zone of rapid remineralisation is not specified. Is this the oxic zone, which varies per site but is usually 0.5-1 cm? It seems so, as the input flux is reconstructed using the integral over the aerobic remineralisation. Why is anaerobic remineralisation disregarded? Please provide a reasoning and/or evidence for doing so here.

13. Table 2: this baffles me, why present a table with only presence/absence information when you can insert the observational values themselves? Most of the table 2 information is also include in table 3. Please create one table, either using the depth-mean values for parameters with a depth-profile (O2, DIC, TOC, ect.) or a marker to state the analysis was performed for that site. Figure S1 could be inserted here if more figures are allowed.

14. Section 4.2 and onwards: please refer to figure 8 (spatial maps) when discussing the observational evidence.

15. Table 3: if SE has no sediment mixing rate due to model failure then please do not include a value in the table. Just a marker will do.

16. Lines 305-308: I'm not an expert on sediment age modelling, but this part seems very subjective to disturbance for me. According to Figure 1b all sites are subject to trawling activity. Is this visible in Figure 3, either by increased sedimentation because a trawl happened nearby or a direct physical disturbance? Is a storm event much different from a direct trawl, apart from the origin of the particles settling afterwards? And how do burrowing animals affect the age calculations?

17. Figure 3: please refer to the spatial maps in Figure 8 here as well.

18. Line 328: if the model was unable to reproduce the profile for site SE, then why are results for SE included in the table? Shouldn't this site be discarded in terms of derived quantities like sedimentation rate, MAR and mixing rate?

19. Line 340: I would not call the TOC profile of site $C_{deep}$ constant below.

20. Figure 6: here the pore-water mixed layer is not used for the linear fit for station NW, even though only stations NNW, NE, E, SE, SC were listed as having a top layer where pore water

mixed with the overlying water column (table 4) and for those stations all points are used. Shouldn't all these sites be treated identically?

21. Figure 7: this is in pore water I assume.

22. Line 393: larger than what? You cannot expect to find identical sedimentation rates across the area, so what has set these stations apart? Are the sedimentation rates large compared to previous estimates, for instance?

23. Line 394: "*it needs to be evaluated*", I would hope the authors mean re-evaluated here, as the initial evaluation should have been done before applying the model.

24. Line 397: I don't see the use of "*however*" here, as this is exactly what you refer to in the previous sentence.

25. Line 411: Figure S4 shows not very good fits for stations E and SE, can you comment on this in the manuscript? How does this affect the sedimentation rate results for these stations.

26. Line 427: "*to fill a significant gap in the understanding of depositional processes in the area*", so what gap is that? The authors state that future work will address this, but an indication would be nice here. Or are they referring to the dominant driver again of carbon burial? This would not be a significant gap as others have already indicated sediment deposition rates to be very important in carbon sequestration (line 64).

27. Line 432: the authors state that the highest sedimentation rates occur in the southern HMNA, but station S has a relatively low sedimentation rate. The discussion here would benefit from a brief overview of (residual) current patterns in the area and the marine footprint of the Elbe and Weser in the site description.

28. Figure 8: the dark blue colour makes the station identification hard to read.

29. Line 445-446: this sentence could use some comma's to improve readability.

30. Line 447: is their any reference that could support this?

31. Line 448: this really should have been done before the current analysis was presented, to avoid speculation now present in the manuscript.

32. Line 494: "*to speculate*" → for speculation

33. Line 494-499: the authors suggest here that because the sediments are classified as impermeable that wave pumping is not a likely explanation of the mixing of pore waters with bottom waters. I would rather question the classification.

34. Line 545: I would say strong terrestrial influence on sediments in the southeastern HMA, and as terrestrial organic matter is already more degraded than marine organic matter this result is fully expected given the geometry of the location and the size of the Elbe river. It would have been strange to find a different result.

35. Line 553: trawling occurs throughout the whole area it seems, but without further evidence you cannot attribute the mixing rate differences solely to them. Especially as no information on biological habitats and ecosystems is provided for the area.

36. Line 558: if more sites were used this result would be more supported. Now the results could be due to other differences between the stations (physical, biological, chemical).

37. Line 565: the term "*massive*" requires a context, is this in relation to other reported values or just to values within this study?

38. Line 566: Zhang et al (2023) has been published.

39. Line 638: "*with*" → while

40. Figure 12: no need to repeat the legend in the caption, rather list the studies that provided the extra dots.

41. Line 659-662: repetitive, this can be shortened.

42. Line 663: the value from de Haas et al (1997) for annual organic carbon accumulation is based on reported literature values for the sedimentation rate in the area and own and reported observations for carbon content and dry bulk density. Naturally the current study presents a more detailed estimate by using 14 sites compared to their 1 site, but as that study is based on data from 1994-1996, do the authors think differences since then in current patterns, trawling activity and biological activity may have added to the difference in organic carbon accumulation values between the current study and the one from de Haas et al? In other words, is their value really an overestimate or is it partly a sign of a different era?

43. Line 667: here the authors state their estimate to be conservative, as representing before-bloom conditions. So maybe the overestimation by de Haas et al (1997) was not so much an overestimation at all? And how would seasonal riverine discharges effect the reported burial efficiencies? The spring bloom is not the only seasonal effect in this area.

44. Line 672: what is the value by Diesing for this area, and what is the 0.79% of total annual organic carbon accumulation based on (i.e. what is their reported value for the total North Sea)?

45. Line 682: TOC is not explained here, though the conclusions should be readable as a stand-alone piece.

**References**

Amorim, F. D. L. L. D., Balkoni, A., Sidorenko, V., & Wiltshire, K. H. (2024). Analyses of sea surface chlorophyll a trends and variability from 1998 to 2020 in the German Bight (North Sea). *Ocean Science*, *20*(5), 1247-1265.

Bockelmann, F. D., Puls, W., Kleeberg, U., Müller, D., & Emeis, K. C. (2018). Mapping mud content and median grain-size of North Sea sediments–A geostatistical approach. *Marine geology*, *397*, 60-71.

de Haas, H., Boer, W., and van Weering, T. C. E. (1997) Recent sedimentation and organic carbon burial in a shelf sea: the North Sea, Mar. Geol., 144, 131–146, https://doi.org/10.1016/S0025-3227(97)00082-0

Lenhart, H. J., & Große, F. (2018). Assessing the effects of WFD nutrient reductions within an OSPAR frame using trans-boundary nutrient modeling. *Frontiers in Marine Science*, *5*, 447.

Legge, O., Johnson, M., Hicks, N., Jickells, T., Diesing, M., Aldridge, J., ... & Williamson, P. (2020). Carbon on the northwest European shelf: Contemporary budget and future influences. *Frontiers in Marine Science*, *7*, 143.

Pätsch, J. (2024). *Daily loads of nutrients, total alkalinity, dissolved inorganic carbon and dissolved organic carbon of the European continental rivers for the years 1977-2022*. Report, Inst. für Meereskunde. Available at https://wiki.cen.uni-hamburg.de/ifm/ECOHAM/DATA_RIVER?action=AttachFile&do=view&target=RIVER_Jun_2024.pdf

Schulz, G., van Beusekom, J. E., Jacob, J., Bold, S., Schöl, A., Ankele, M., ... & Dähnke, K. (2023). Low discharge intensifies nitrogen retention in rivers–a case study in the Elbe River. *Science of the Total Environment*, *904*, 166740.

Tian, T., Su, J., Flöser, G., Wiltshire, K., & Wirtz, K. (2011). Factors controlling the onset of spring blooms in the German Bight 2002–2005: light, wind and stratification. *Continental Shelf Research*, *31*(10), 1140-1148.

Van Beusekom, J. E., Carstensen, J., Dolch, T., Grage, A., Hofmeister, R., Lenhart, H., ... & Ruiter, H. (2019). Wadden Sea Eutrophication: long-term trends and regional differences. *Frontiers in marine science*, *6*, 370.

Zhang, W., Porz, L., Yilmaz, R., Wallmann, K., Spiegel, T., Neumann, A., ... & Schrum, C. (2023). Intense and persistent bottom trawling impairs long-term carbon storage in shelf sea sediments.

---

## Editor Decision (ED1)

**Review** of the manuscript *Depositional controls and budget or organic carbon burial I fine-grained sediments of the North Sea – the Helgoland Mud Area as a natural laboratory* , by Müller et al, submitted to **Biogeosciences** (egusphere-2024-1632).

**Manuscript overview**

Given before, this is the second review round for this particular reviewer

**Review overview**

Following the initial presented manuscript (which had been reviewed before) the authors have made textual changes, as suggested. Some of my comments were is direct opposition to previous comments by other reviewers, so I accept the choices made by the authors to keep all/most reviewers happy. I do have some comments on the revised version though:

I know the sedimentation rates do not include seasonality, but my point was more on interannual variability. As the sedimentation rate estimates are based on longer timescales I would expect a shift in values as long term trends in e.g. SPM or DOC loads from the Elbe become visible. This hinders comparison to values obtained decades ago, for instance, if the trends are significant (which was my point for the comparison to de Haas et al (1997), this is nearly 30 years ago). This caveat should be included here, as marine trends have become stronger due to climate change in this region. Not necessarily for SPM, DIC or DOC loads from the Elbe (there is no obvious trend yet in the data from Patsch (2024), which is worth mentioning), but in general I think this point is worth including in the text. The authors have not addressed this point in the revised text.

8. I also understand the focus is not on biology here, yet a few references to relevant work such as Neumann et al (2013), Shojaei et al (2016), Shojaei et al (2021) and particularly Thatje & Gerdes (1997) and Wrede et al (2017, already referenced elsewhere) would not be amiss given the speculation later of benthic activity differences. True, the spatial scale is not the same, but Thatje & Gerdes do classify a large part of the HMA as 1 ecosystem type, which is relevant for the presented work when bioturbation is discussed, and Wrede et al (2017) have quantified bioturbation potential in the area. So mentioning this in the site description seems worthwhile to me.

27. A reference to Callies et al (2017) would not be amiss here, just for readers who wish to have a better overview of local residual current patterns.

Purely textual:
- Line 91: remove "as a natural laboratory", as the sentence is grammatically wrong this way and it is repetitive with regard to line 95.
- Line 553: "this is *likely* a result of", as you cannot be sure of the cause.
- Line 651: "sites *are* exceptionally" or replace while with "with" in the line above.

**Recommendation**

Accept after minor revision

**References**

Callies, U., Gaslikova, L., Kapitza, H. and Scharfe, M., 2017. German Bight residual current variability on a daily basis: principal components of multi-decadal barotropic simulations. *Geo-Marine Letters*, *37*, pp.151-162.

de Haas, H., Boer, W., and van Weering, T. C. E. (1997) Recent sedimentation and organic carbon burial in a shelf sea: the North Sea, Mar. Geol., 144, 131–146, https://doi.org/10.1016/S0025-3227(97)00082-0

Neumann, H., Reiss, H., Ehrich, S., Sell, A., Panten, K., Kloppmann, M., Wilhelms, I. and Kröncke, I., 2013. Benthos and demersal fish habitats in the German Exclusive Economic Zone (EEZ) of the North Sea. *Helgoland Marine Research*, *67*, pp.445-459.

Shojaei, M.G., Gutow, L., Dannheim, J., Schröder, A. and Brey, T., 2021. Long-term changes in ecological functioning of temperate shelf sea benthic communities. *Estuarine, Coastal and Shelf Science*, *249*, p.107097.

Shojaei, M.G., Gutow, L., Dannheim, J., Rachor, E., Schröder, A. and Brey, T., 2016. Common trends in German Bight benthic macrofaunal communities: Assessing temporal variability and the relative importance of environmental variables. *Journal of Sea Research*, *107*, pp.25-33.

Thatje, S. and Gerdes, D., 1997. The benthic macrofauna of the inner German Bight: present and past. *Archive of fishery and marine research, 45 (2)*, *1997*, pp.93-112.

Wrede, A., Dannheim, J., Gutow, L., and Brey, T.: Who really matters: influence of German Bight key bioturbators on biogeochemical cycling and sediment turnover, J. Exp. Mar. Bio. Ecol., 488, 92–101, https://doi.org/10.1016/j.jembe.2017.01.001, 2017.

---

## Author Response (AR2)

**Reviewer #3**

**Review** of the manuscript "Depositional controls and budget or organic carbon burial in fine-grained sediments of the North Sea – the Helgoland Mud Area as a natural laboratory", by Müller et al, submitted to Biogeosciences (egusphere-2024-1632).

Manuscript overview

The manuscript presents the results of an extensive sediment core analysis in the Helgoland Mud Area, with the aim of quantifying the burial efficiency of organic carbon in the area. This includes presentation per site of porosity, grain size and mud content data, as well as Dissolved Inorganic Carbon (DIC) pore water profiles and Total Organic Carbon (TOC) and O2 sedimentary profiles and isotope analysis. Derived from these data are sediment age profiles, aerobic remineralisation rates, sedimentation rate, mass accumulation rate, total organic carbon accumulation rate, organic carbon burial efficiency and the source of the organic material (marine or terrestrial).

The presented data covers 14 sites southeast of Helgoland in the German Bight (North Sea, European Shelf), of which cores were taken in spring 2022 (ranging from mid-March to the beginning of April). Data from a 2021 cruise to the same area (end of April) is also used, covering 3 stations of which 2 were resampled in 2022. The data has been made publicly available by the authors. They conclude that sedimentation rate is the main factor driving the burial efficiency and that there is a south-north gradient in the source of the organic material, with the southern part characterised by less degradable material from terrestrial/riverine sources. They also conclude that the area stores a higher percentage of carbon for the North Sea than its size alone would merit.

Review overview

First of all I would like to state that I am not an expert on the applied methods, and that I realise the paper has been reviewed before. It is therefore likely that some of my comments apply to text changed on behalf of other reviewers' comments. I did not consider the other reviews comments in the generation of this review report.

Author's response: We would like to thank the anonymous reviewer #3 for the thoughtful and constructive feedback on our manuscript. We have tried to follow a balanced approach in cases of new review aspects that are conflicting with changes suggested by the earlier reviewers.

The paper follows the standard methods, results, conclusions outline, yet core elements such as the physical and biological environment are not mentioned until late in the manuscript. Without these interpretation of the presented results is difficult. For instance, only at the end do the authors state that the area sediments are classified as impermeable (line 496) and that the sampling occurred before the spring bloom (line 600). No evidence is given for the latter statement, and as some sites were sampled in April I would like to see some confirmation for this. Tian et al (2011) show that the Helgoland Roads data indicate a spring bloom start (varying with the physical conditions) between the end of March and the first two weeks of April. They

considered only 2002-2005 data, but Amorim et al (2024) show positive trends in satellite-derived Chla observations in the study area (and elevated Chla values in March in part of the area) over the 1998-2020 period. In their figure 8 full bloom dynamics occur in April, so any assertion that the reported cruises were before the spring bloom (sample dates cover 17-03 to 29-04) should be substantiated (the O2 profiles indicate rapid consumption at the core top). As this is important for the derived sedimentation rates and origin of organic material, information regarding this should be included in the site description at the beginning. This also applies to the variability (seasonal, interannual) in riverine discharges to the area (Elbe and Weser) for suspended particular matter (SPM), dissolved inorganic carbon (DIC) and dissolved organic carbon) DOC: discharges have been declining, how does this affect the results in this paper? See for instance Patsch (2024), Schultz (2023) and Van Beusekom (2019).

Author's response: We have added a description of the spring bloom in the section study area (lines 124-127; lines correspond to the manuscript version with track changes). This clarifies that the onset of the spring bloom may overlap with parts of our sampling in late March (e.g., Teeling et al., 2012; Amorim et al., 2024). However, the signal was not transferred to the benthic system, as in our case no fluffy layer was observed at the top of the sediment cores while sampling (added in lines 173-174) and no elevated TOC contents were observed at the top of the core (discussed in lines 598-604). This is consistent with findings for the southern North Sea by e.g., Provoost et al. (2013), which show that the benthic system lags behind the spring bloom in the water column for several months (added in lines 127-128). This does not affect the sedimentation rate calculations, as (1) organic carbon contributes only a small fraction to the total sediment (maximum TOC contents ~2 wt%) and (2) the signal derived from the sediment analyses integrates over larger time scales. This is due to the sampling of sediment layers of 1 cm, averaging seasonal signals at every datapoint (in our case one sediment layer integrates over >2 to 20 years). While this does not allow the investigation of seasonal effects within the sediments, it does filter the seasonal signal from the dataset, which is an advantage in this case and differs from sampling in the e.g., highly dynamic water column. As we further use a 60 year average for our interpolations/interpretations (see lines 279-280), all sites are at least comparable with respect to longer-term environmental changes. This also applies to the determination of the origin of the degraded organic carbon in the sediments, as this is similarly beyond the filter of seasonal effects.

Apart from the physical and biological setting the authors omit to consider interannual variability (river discharges, stratification and onset of spring bloom, circulation patterns), and state that there is ongoing discussion about the depositional mechanism in the area which will be the subject of a further study. I would argue that at least a physical discussion should be part of this manuscript, as the depositional mechanism determines the sedimentation rates, origin of organic material and thus burial efficiency. The wealth of data presented here will be important for that discussion, but for a true interpretation of the gathered observational data the physical setting remains key. Yet the current manuscript does not contain a detailed hydrodynamic overview or an analysis of riverine loads (SPM, carbon compounds) to the area (and their interannual variability, e.g. in years with high riverine discharge the terrestrial matter will reach a larger part of the area and push the coastal current more offshore). A lot of work has gone into this study and the results certainly merit publication, but they should be presented within

context. Now the manuscript contains some bold statements (overestimation by de Haas et al (1997)) and many suggestions (in area 1 the main driver for mixing is benthos and in area 2 it is very active benthos) without much evidence beyond the direct measurements. For the manuscript it would be better if these suggestions were more substantiated. Alternatively, the authors could wait with publication until their further studies are concluded and publish a double paper, but funding agencies usually want to see intermediate results. More detailed comments are listed below.

Author's response: We agree with the reviewer that a multidisciplinary approach including processes from the atmosphere, water column and sediments will always provide the most comprehensive understanding of a system. We focus here on the perspective of sediment biogeochemistry and including in the same framework a detailed investigation of e.g., the hydrodynamic mechanism in the study area would extend well beyond the scope and intention of the present manuscript. This perspective will be the context for further studies addressing hydrodynamics and sediment transport. Our interpretation of sedimentary effects (e.g. sediment mixing) that assume a cause from fauna or hydrodynamic-related phenomena may serve as a testable hypothesis for future work. A detailed response to the comment on the previous organic carbon flux estimate and the dominant drivers of sediment mixing is provided in the detailed comments section. In short: as outlined below, we state in the discussion (section 5.4, lines 685-690) that we compare a one-point estimate by de Haas et al. (1997) with our high-resolution dataset. The resulting difference in annual organic carbon accumulation is best explained by the uncertainty from model assumptions where they had less observational data. Regarding the dominant driver of sediment mixing: we defined the dominant drivers of sediment mixing in the study area based on our observational data and estimated bottom trawling activity in the HMA. Our approach is discussed in section 5.5.1 of the manuscript and a detailed response to this issue is provided below.

**Recommendation**

Moderate revision (no new analysis or figures needed but textual changes required)

**Detailed Comments**

1. Abstract: The abstract amounts to 28 lines of text while the Conclusions only hold 16 lines. This is not an abstract, this is an Introduction. A true abstract is short while capturing the set-up of the work and the main take home messages.

Author's response: We have condensed the abstract. The introduction statement in the abstract comprises now 4 lines of text.

2. Introduction: I miss an overview here of shelf processes which store carbon long-term, and a reference to studies that examine them like Legge et al (2020). This would provide more context for the presented work: for the North Sea area, how much carbon is stored in the sediments compared to for instance off-shelf transport? And therefore, how much does this particular site add to this on-shelf storage?

Author's response: We thank the reviewer for this comment. We have added a brief overview of the processes on the shelf, the balance between the mechanisms and their contribution to the Northwestern European Shelf, which includes the North Sea accordingly (lines 82-87). The contribution of the HMA in this context is subject of the discussion in section 5.4.

3. Line 64: "Sedimentation rate is one of the most important factor controlling the preservation of OM in the sediment" is followed by 4 references. But this is also presented as a conclusion of this manuscript in line 24 (albeit as simple the most important factor). I think the authors should make it more clear that their work confirms the earlier work with respect to the process controlling carbon burial.

Author's response: We have edited the statement in the abstract in line 23 accordingly. The part of the abstract reads now "High sedimentation rates are known to limit oxygen exposure time thereby enhancing OM preservation. Our data support this finding, demonstrate and confirm that sedimentation rate is the key factor determining organic carbon burial efficiency and long-term sedimentary carbon storage." (lines 23-25).

4. Lines 76-83: fine coastal sediments may have a low O2 penetration depth but they are also characterised by high anthropogenic activity (dredging/dumping, shipping, fishing/trawling, offshore renewable energy, cabling) and biotic activity (more nutrients close to the coast). I miss a discussion on how for instance increased use by offshore wind farms will affect the processes described here or their interpolation to the larger area. It doesn't have to be in-depth but any extrapolation to the North Sea scale should mention this at least.

Author's response: We have included off-shore windfarms as an anthropogenic activity, that can alter the preservation of OC in marine sediments in line 62. This topic is certainly of interest, however, we did not further expand our discussion regarding this topic, since with our dataset an interpretation of large-scale impacts of off-shore windfarms would (if possible at all) be beyond the scope of the manuscript.

5. Lines 91-95: The analysis is focussed on the HMA, which is by no means a common environment type for the North Sea, as shown by the referenced Bockelmann et al (2018). So I don't quite see why the authors claim to determine the main depositional drivers for carbon burial in the North Sea.

Author's response: Thank you for this comment. We are using the HMA as a natural laboratory to assess the controls on organic carbon burial in fine-grained sediments of the North Sea shelf sediments, with insights that can be applied to muddy sediments beyond the HMA. We have amended the statement in the introduction accordingly to make this point clear (lines 100-101).

6. Figure 1: The abbreviation MUC should be explained in the caption. And I would prefer more detail in the bathymetry and possibly a different colour scheme) so that the Elbe channel is visible in more detail.

Author's response: We have added the abbreviation "multi-corer (MUC)" in the caption of Fig. 1 accordingly. We discussed the colouring of the bathymetry in Fig. 1a and decided to keep the

blue colouring, as we did not want to overinterpret the generally small differences in bathymetry and give the impression of large valleys instead of moderate differences.

7. Line 102: the reference here is to Hagen et al (2021), but this study solely relates to the German Bight, not to the whole of the Southern North Sea.

Author's response: We changed the spatial reference in the text accordingly and line 108 reads now "The German Bight of the southern North Sea is characterised by […]".

8. Line 134: why can't the site be classified as such? At the very least an indication of the complex hydrodynamic conditions should be provided here, to sketch the situation and aid interpretation of the results. This should also include a few lines on biotopes (indicating biological activity on or in the sea bed), riverine influences and interannual variability. How far does the influence of the Elbe and Weser reach in general? This could be indicated by salinity gradients over the area or by studies such as Lenhart & Große (2018). Does the muddy Ems influence this region at all? How about the storm floods, did this increase terrestrial material input to the area? The storms are now mentioned at the start of the methods section, for me it would be more logical if presented here with the site overview. I realise the authors use a steady-state model for the derivation of some quantities, but the marine environment is not steady state and the direct observations require an indication of the dynamic setting.

Author's response: We have edited the statement on riverine-influenced deposition in line 143 accordingly. In the study area (lines 108-113), the complexity of tidal and wave energy and its impact on the sediments are mentioned. As mentioned in the manuscript, a follow-up study within the project will address this issue in much greater detail. Regarding the distribution of biological data: to our knowledge, no study has been conducted at a spatial resolution (see e.g., Wrede et al., 2017) that allows differentiation of biological habitats within the HMA. As we focus on sediment biogeochemistry and not biology in this manuscript, we believe that a shift to the new aspect of biology is beyond the scope of this study. As mentioned above, we are aware of the interesting insights of coupling biogeochemistry and biology in this respect, which will be investigated in a forthcoming PhD project.

We have added a statement on the influence of the Elbe river on the sediments of the HMA accordingly (lines 124-126). We agree with the reviewer that the marine system is not in a steady state, but as mentioned above, the advantage of the sediment record in this respect is that the seasonal signals (e.g., riverine discharge, SPM, storm events) are filtered or smoothed by the sampling of various years within one sediment layer.

The occurrence of storm events in the German Bight is mentioned in line 124 of the study site description as a feature of the location. In the materials and methods section, we refer to the specific winter storms before our research cruise. We keep this separation to ensure that the general insights are in the site description and the specifics are given below.

9. Table 1: can the trawling pressure from Figure 1b be included here for each station? Because it seems to me that in the figure site W has a higher (or equal) trawling activity than site NW, but the text later claims that W has a lower trawling activity than NW (line 556). Or are you

referring to the proximity of higher trawled areas? In that case I would like to see residual current patterns for the area.

Author's response: We refer later in the discussion to the combination of the bottom trawling pressure (the swept area ratio as a proxy, Fig. 1b) and our modelled sediment mixing rates to identify the dominant source of mixing (section 5.1.1) and then compare this in the following discussion. Swept area ratios as a proxy for bottom trawling activity are known to have uncertainties and are only given for a certain period of time. Therefore, we compare the spatial overlap of the pattern bottom trawling activity estimates with our calculated sediment mixing rates based on our radionuclide measurements (lines 489-491). This prevents us from overemphasising the mapped estimated bottom trawling activity by using observations from the sediments.

10. Line 170: what language is the GRADISTAT application in? Python, R, Matlab, IDL, Julia, …?

Author's response: The program is supplied as the Microsoft Excel spreadsheet package. For further information see the corresponding publication by Blott and Pye (2001). The reference is cited in the manuscript in line 190.

11. Line 210: "1 sigma uncertainty", do you mean one standard deviation?

Author's response: Yes, one standard deviation. We have changed "1 sigma uncertainty" to the requested expression accordingly (line 225).

12. Line 253: the zone of rapid remineralisation is not specified. Is this the oxic zone, which varies per site but is usually 0.5-1 cm? It seems so, as the input flux is reconstructed using the integral over the aerobic remineralisation. Why is anaerobic remineralisation disregarded? Please provide a reasoning and/or evidence for doing so here.

Author's response: As we do not observe the typical depth-decreasing TOC profiles – indicating rapid organic carbon remineralisation – we use integrated aerobic remineralisation to calculate the influx to the sediments. We have modified the statement in section 3.4 accordingly to match the formulation used by Burdige (2007). More details on the methodology are provided in the published response to reviewer #2. A detailed description of anaerobic remineralisation can also be found there. In short: by using total oxygen uptake/aerobic remineralisation rates, we take into account not only aerobic respiration but also oxygen used for the re-oxidation of reduced metabolic products. This has been shown to be a valid approximation in cohesive sediments by e.g., Glud (2008). A statement has been added in lines 266-268 accordingly.

13. Table 2: this baffles me, why present a table with only presence/absence information when you can insert the observational values themselves? Most of the table 2 information is also include in table 3. Please create one table, either using the depth-mean values for parameters with a depth-profile (O2, DIC, TOC, ect.) or a marker to state the analysis was performed for that site. Figure S1 could be inserted here if more figures are allowed.

Author's response: This is a case where reviews really differed. In the original version of the manuscript, Fig. S1 was part of the main text and Table 2 was not part of the manuscript as now suggested by reviewer #3. Figure S1 has been moved from the main text to the supplementary material and Table 2 has been created at the request of reviewer #2. Either way, all the required information is available in the main text or supplementary material, so the position of the tables would not really affect the completeness of the description. We have therefore decided not to change the table/figure back to the original version.

14. Section 4.2 and onwards: please refer to figure 8 (spatial maps) when discussing the observational evidence.

Author's response: We have deliberately refrained from referring to the spatial maps in the results section. In our opinion, the interpolation of the data across the study area required for showing a map is an essential step in the interpretation of the results and is therefore only referred to in the discussion.

15. Table 3: if SE has no sediment mixing rate due to model failure then please do not include a value in the table. Just a marker will do.

Author's response: The value has been replaced by the marker in Table 3 accordingly.

16. Lines 305-308: I'm not an expert on sediment age modelling, but this part seems very subjective to disturbance for me. According to Figure 1b all sites are subject to trawling activity. Is this visible in Figure 3, either by increased sedimentation because a trawl happened nearby or a direct physical disturbance? Is a storm event much different from a direct trawl, apart from the origin of the particles settling afterwards? And how do burrowing animals affect the age calculations?

Author's response: Here in the results section we present the results of the calculations performed and avoid interpretation of the data at this point. The reviewer is correct in assuming that disturbance or mixing will affect the sedimentation rate results with depth, showing increasing sedimentation rates at the top of the core. The direct influence of sediment mixing on the results of the CRS age model and its application is discussed in section 5.1. In short: the CRS age model has been shown to be very robust for average sedimentation rates/MAR (Arias-Ortiz et al., 2018), using the inventory of $^{210}Pb_{xs}$ and not the activity at z = 0, which can be strongly influenced by sediment mixing (Arias-Ortiz et al., 2018).

Regarding the impact of a trawling event on sedimentation rates/MAR: as the North Sea has been heavily trawled for centuries, the total particle load in the water column is likely affected by bottom trawling, but we assume that the impact of a specific trawling event is not as significant as the overall trawling activity in the German Bight/southern North Sea contributing to the overall particle load. A statement regarding the overall particle load has been added to the introduction in lines 137-139, as suggested by reviewer #1.

In section 5.1 we discuss that we can identify sediment mixing using $^{210}Pb_{xs}$ profiles, but that we cannot distinguish between different drivers of the mixing (biological or physical) from the model alone. Therefore, we use the modelled sediment mixing rates and the spatial overlap with

the swept area ratio map to identify what might be the dominant driver of sediment mixing. The average sedimentation rates/MAR derived from the CRS age model are not affected by the mixing, as mentioned above (Arias-Ortiz et al., 2018).

17. Figure 3: please refer to the spatial maps in Figure 8 here as well.

Author's response: We have deliberately refrained from referring to the spatial maps in the results section. Please find the explanation in comment 14.

18. Line 328: if the model was unable to reproduce the profile for site SE, then why are results for SE included in the table? Shouldn't this site be discarded in terms of derived quantities like sedimentation rate, MAR and mixing rate?

Author's response: As requested in comment 15, the value has been exchanged in Table 3 accordingly. The average sedimentation rates and MAR are not influenced, since those were produced using the inventory of $^{210}Pb_{xs}$ and the CRS model. Please see the response to comment 16 and line 418 in the manuscript.

19. Line 340: I would not call the TOC profile of site Cdeep constant below.

Author's response: We have changed the description to "[…] downward decrease and a constant or slight scattering around a constant TOC content below" accordingly (line 358).

20. Figure 6: here the pore-water mixed layer is not used for the linear fit for station NW, even though only stations NNW, NE, E, SE, SC were listed as having a top layer where pore water mixed with the overlying water column (table 4) and for those stations all points are used. Shouldn't all these sites be treated identically?

Author's response: Using the Miller-Tans plots we have calculated the signature of the degraded OM as DIC production with the corresponding isotopic signature of remineralised OC. As neither the DIC content nor its isotopic signature changes in the upper part of the sediments, we did not want to include these points in the regression analysis. The rather constant values at site NW could be caused by different mechanisms, e.g. mixing of sediments, bioirrigation or lower OC remineralisation in the upper part of the sediments.

21. Figure 7: this is in pore water I assume

Author's response: We have changed the figure caption accordingly. The figure caption starts now: "Pore-water dissolved oxygen […]".

22. Line 393: larger than what? You cannot expect to find identical sedimentation rates across the area, so what has set these stations apart? Are the sedimentation rates large compared to previous estimates, for instance?

Author's response: Thank you for this comment to clarify this statement. The changes in sedimentation rates refer to changes in the sedimentation rate within a sediment core. We have edited the statement to make this clear in the text (lines 411-412).

23. Line 394: "it needs to be evaluated", I would hope the authors mean re-evaluated here, as the initial evaluation should have been done before applying the model.

Author's response: Yes, this is a re-evaluation. We have edited the expression accordingly (line 413).

24. Line 397: I don't see the use of "however" here, as this is exactly what you refer to in the previous sentence.

Author's response: We have deleted the word accordingly (line 416).

25. Line 411: Figure S4 shows not very good fits for stations E and SE, can you comment on this in the manuscript? How does this affect the sedimentation rate results for these stations.

Author's response: The resulting not-very-good fits for sites E and SE show that the pattern of $^{210}Pb_{xs}$ profiles at these sites cannot be perfectly described by steady-state sedimentation and steady-state sediment mixing. As the reviewer pointed out earlier in the review, processes in the marine system cannot always be expected to be described by a steady state. Due to the output of the sediment mixing model for site SE, we decided not to use the results below. We have added a statement to make this clear (lines 346-347).

The sedimentation rates used in the manuscript for further calculations are based on the CRS model, not the sediment mixing model. This has the advantage that e.g. the signal resulting from non-steady sediment mixing, will not affect the $^{210}Pb_{xs}$ inventory and hence the average sedimentation rates. For this reason, we only use average sedimentation rates in the manuscript, as too many processes can alter $^{210}Pb_{xs}$ profiles (see e.g., Arias-Ortiz et al., 2018) and make it impossible to reasonably discuss changes in sedimentation rates or MAR within one core. The effects of sediment mixing are given in the manuscript in lines 424-426 We have added "as it uses the $^{210}Pb_{xs}$ inventory" and further information can be found in the cited reference Arias-Ortiz et al. (2018) in lines 418-419.

26. Line 427: "to fill a significant gap in the understanding of depositional processes in the area", so what gap is that? The authors state that future work will address this, but an indication would be nice here. Or are they referring to the dominant driver again of carbon burial? This would not be a significant gap as others have already indicated sediment deposition rates to be very important in carbon sequestration (line 64).

Author's response: Thank you for this comment. We agree that there is not so much a „knowledge gap" but rather conflicting approaches and results for the sedimentary conditions, as mentioned in the study area description (lines 128-134). Studies with varying resolutions and methods operating on different timescales have yielded a wide range of sedimentation/MAR in the HMA. To resolve this unresolved issue of conflicting descriptions, we provide an overall more conclusive approach that will also help to describe the depositional conditions. We have changed the statement in lines 446-447 to make this clear in the text.

27. Line 432: the authors state that the highest sedimentation rates occur in the southern HMNA, but station S has a relatively low sedimentation rate. The discussion here would benefit from a

brief overview of (residual) current patterns in the area and the marine footprint of the Elbe and Weser in the site description.

Author's response: We have corrected the statement in lines 453 to "southeastern". To our knowledge, a detailed description of residual currents explaining the deposition in the HMA is yet missing. We have added a description of the Elbe river footprint to the study area accordingly (lines 123-126).

28. Figure 8: the dark blue colour makes the station identification hard to read.

Author's response: We believe this would be very important in a printed publication. As the online publication has the great advantage of high-resolution images and the ability to easily zoom in and out, we have decided to retain the uniform colour scheme within Fig. 8. In addition, the locations of the sites are shown in Fig. 1a.

29. Line 445-446: this sentence could use some comma's to improve readability.

Author's response: Commas have been added to improve readability accordingly.

30. Line 447: is their any reference that could support this?

Thank you for this comment. As the impact of the storm event on sediments in the shallow eastern HMA is discussed in detail in the following paragraph, we have decided to delete this sentence.

31. Line 448: this really should have been done before the current analysis was presented, to avoid speculation now present in the manuscript.

Author's response: Please see the detailed response to the comment at the beginning of the review.

32. Line 494: "to speculate" → for speculation

Author's response: The change has been made in line 513.

33. Line 494-499: the authors suggest here that because the sediments are classified as impermeable that wave pumping is not a likely explanation of the mixing of pore waters with bottom waters. I would rather question the classification.

Author's response: We have changed the statement in lines 516-518, to make it clearer that this could also be a result of the classification.

34. Line 545: I would say strong terrestrial influence on sediments in the southeastern HMA, and as terrestrial organic matter is already more degraded than marine organic matter this result is fully expected given the geometry of the location and the size of the Elbe river. It would have been strange to find a different result.

Author's response: We agree with the reviewer's statement, as we also conclude in our manuscript that these are the conditions and controls for the preservation of OC in the sediments of the HMA (lines 564-566).

35. Line 553: trawling occurs throughout the whole area it seems, but without further evidence you cannot attribute the mixing rate differences solely to them. Especially as no information on biological habitats and ecosystems is provided for the area.

Author's response: Detailed analyses of biological habitats are not sufficiently detailed for the HMA (e.g., Wrede et al., 2017, for the entire German Bight). For our classification of the dominant driver of sediment mixing (bioturbation vs. bottom trawling), we compared the spatial overlap of radionuclide-derived sediment mixing rates (Fig. 8c) with the estimated distribution of bottom trawling activity (Fig. 1b), as described in lines 489-491. We do not claim that mixing rate differences are solely determined by trawling.

36. Line 558: if more sites were used this result would be more supported. Now the results could be due to other differences between the stations (physical, biological, chemical).

Author's response: We agree with the reviewer that more data will always improve the interpretation of observations and that the future will provide helpful observational data on the influence of bottom trawling on sediments. We are aware of the data density in this area in our study, as we stated in lines 573-574. However, as these observations are rare and, in terms of bottom trawling activity, restricted to different locations for comparison (see e.g., Paradis et al., 2019), we believe it is justified to present the data in this context. In addition, reviewer #1 and reviewer #2 requested a separate section heading for this topic.

37. Line 565: the term "massive" requires a context, is this in relation to other reported values or just to values within this study?

Author's response: We have specified the text accordingly (line 588).

38. Line 566: Zhang et al (2023) has been published.

Author's response: We have changed the citation accordingly (line 589).

39. Line 638: "with" → while

Author's response: The sentence reads while.

40. Figure 12: no need to repeat the legend in the caption, rather list the studies that provided the extra dots.

Author's response: The figure caption has been condensed accordingly.

41. Line 659-662: repetitive, this can be shortened.

Author's response: The text has been condensed accordingly and now reads: "Based on the distribution of total organic carbon accumulation rates in our study area (Fig. 10b), we

calculated an areal mean TOC accumulation rate of 22.5 g C m$^{-2}$ yr$^{-1}$. The annual organic carbon accumulation for the entire HMA was calculated to amount to 0.011 Tg C yr$^{-1}$." (lines 682-685).

42. Line 663: the value from de Haas et al (1997) for annual organic carbon accumulation is based on reported literature values for the sedimentation rate in the area and own and reported observations for carbon content and dry bulk density. Naturally the current study presents a more detailed estimate by using 14 sites compared to their 1 site, but as that study is based on data from 1994-1996, do the authors think differences since then in current patterns, trawling activity and biological activity may have added to the difference in organic carbon accumulation values between the current study and the one from de Haas et al? In other words, is their value really an overestimate or is it partly a sign of a different era?

Author's response: The uncertainties in the assumptions made by de Haas et al. (1997) are large compared to the data available in the present study, as the reviewer points out in the comment. The resulting difference in annual organic carbon accumulation is best explained by the uncertainty resulting from uncertain assumptions where they had less observational data. For assessing a long-term trend or the influence of changes in ocean currents, trawling activity, biology, etc., error bars would be too large to identify a trend with statistical significance.

43. Line 667: here the authors state their estimate to be conservative, as representing before-bloom conditions. So maybe the overestimation by de Haas et al (1997) was not so much an overestimation at all? And how would seasonal riverine discharges effect the reported burial efficiencies? The spring bloom is not the only seasonal effect in this area.

Author's response: Sediment samples are always integrated over time, as the sampled one-centimetre sediment layers cover several years (see comment above). The sediment smoothes out seasonal patterns such as river discharge. An exception could be the incorporation of higher TOC contents in the uppermost one to two cm of the sediments slightly changing our calculations, as stated in the manuscript in lines 691-694. Even taking this into account, the value estimated by de Haas et al. (1997) is still higher, which is a result of the assumptions made (see comment above).

44. Line 672: what is the value by Diesing for this area, and what is the 0.79% of total annual organic carbon accumulation based on (i.e. what is their reported value for the total North Sea)?

Author's response: In Diesing et al. (2021) the model did not represent the HMA as an area of high TOC accumulation. The model results quantified the HMA as a transition and turnover zone characterised by low organic carbon accumulation – on average less than 2 gC m$^{-2}$ yr$^{-1}$ (Diesing et al., 2021). As also pointed out by reviewer #1, the carbon storage and accumulation potential of the HMA has so far been overlooked. To calculate the relative contribution of the HMA, we used the value by Diesing et al. (2021) for organic carbon accumulation of 1.43 TgC yr$^{-1}$ for the entire North Sea including the Skagerrak.

45. Line 682: TOC is not explained here, though the conclusions should be readable as a stand-alone piece.

Author's response: The abbreviation has been added in the conclusion accordingly (line 705).

References

Amorim, F. D. L. L. D., Balkoni, A., Sidorenko, V., & Wiltshire, K. H. (2024). Analyses of sea surface chlorophyll a trends and variability from 1998 to 2020 in the German Bight (North Sea). Ocean Science, 20(5), 1247-1265.

Bockelmann, F. D., Puls, W., Kleeberg, U., Müller, D., & Emeis, K. C. (2018). Mapping mud content and median grain-size of North Sea sediments–A geostatistical approach. Marine geology, 397, 60-71.

de Haas, H., Boer, W., and van Weering, T. C. E. (1997) Recent sedimentation and organic carbon burial in a shelf sea: the North Sea, Mar. Geol., 144, 131–146, https://doi.org/10.1016/S0025-3227(97)00082-0

Lenhart, H. J., & Große, F. (2018). Assessing the effects of WFD nutrient reductions within an OSPAR frame using trans-boundary nutrient modeling. Frontiers in Marine Science, 5, 447.

Legge, O., Johnson, M., Hicks, N., Jickells, T., Diesing, M., Aldridge, J., ... & Williamson, P. (2020). Carbon on the northwest European shelf: Contemporary budget and future influences. Frontiers in Marine Science, 7, 143.

Pätsch, J. (2024). Daily loads of nutrients, total alkalinity, dissolved inorganic carbon and dissolved organic carbon of the European continental rivers for the years 1977-2022. Report, Inst. für Meereskunde. Available at https://wiki.cen.uni-hamburg.de/ifm/ECOHAM/DATA_RIVER?action=AttachFile&do=view&target=RIVER_Jun_2024.pdf

Schulz, G., van Beusekom, J. E., Jacob, J., Bold, S., Schöl, A., Ankele, M., ... & Dähnke, K. (2023). Low discharge intensifies nitrogen retention in rivers–a case study in the Elbe River. Science of the Total Environment, 904, 166740.

Tian, T., Su, J., Flöser, G., Wiltshire, K., & Wirtz, K. (2011). Factors controlling the onset of spring blooms in the German Bight 2002–2005: light, wind and stratification. Continental Shelf Research, 31(10), 1140-1148. Van Beusekom, J. E., Carstensen, J., Dolch, T., Grage, A., Hofmeister, R., Lenhart, H., ... & Ruiter, H. (2019). Wadden Sea Eutrophication: long-term trends and regional differences. Frontiers in marine science, 6, 370.

Zhang, W., Porz, L., Yilmaz, R., Wallmann, K., Spiegel, T., Neumann, A., ... & Schrum, C. (2023). Intense and persistent bottom trawling impairs long-term carbon storage in shelf sea sediments.

References used in the author's responses

Amorim, F. de L. L. de, Balkoni, A., Sidorenko, V., and Wiltshire, K. H.: Analyses of sea

surface chlorophyll a trends and variability from 1998 to 2020 in the German Bight (North Sea), Ocean Sci., 20, 1247–1265, https://doi.org/10.5194/os-20-1247-2024, 2024.

Arias-Ortiz, A., Masqué, P., Garcia-Orellana, J., Serrano, O., Mazarrasa, I., Marbà, N., Lovelock, C. E., Lavery, P. S., and Duarte, C. M.: Reviews and syntheses: 210Pb-derived sediment and carbon accumulation rates in vegetated coastal ecosystems – setting the record straight, Biogeosciences, 15, 6791–6818, https://doi.org/10.5194/bg-15-6791-2018, 2018.

Blott, S. J. and Pye, K.: GRADISTAT: a grain size distribution and statistics package for the analysis of unconsolidated sediments, Earth Surf. Process. Landforms, 26, 1237–1248, https://doi.org/10.1002/esp.261, 2001.

Burdige, D. J.: Preservation of organic matter in marine sediments: controls, mechanisms, and an imbalance in sediment organic carbon budgets?, Chem. Rev., 107, 467–485, https://doi.org/10.1021/cr050347q, 2007.

Diesing, M., Thorsnes, T., and Bjarnadóttir, L. R.: Organic carbon densities and accumulation rates in surface sediments of the North Sea and Skagerrak, Biogeosciences, 18, 2139–2160, https://doi.org/10.5194/bg-18-2139-2021, 2021.

Glud, R. N.: Oxygen dynamics of marine sediments, Mar. Biol. Res., 4, 243–289, https://doi.org/10.1080/17451000801888726, 2008.

de Haas, H., Boer, W., and van Weering, T. C. E.: Recent sedimentation and organic carbon burial in a shelf sea: the North Sea, Mar. Geol., 144, 131–146, https://doi.org/10.1016/S0025-3227(97)00082-0, 1997.

Paradis, S., Pusceddu, A., Masqué, P., Puig, P., Moccia, D., Russo, T., and Lo Iacono, C.: Organic matter contents and degradation in a highly trawled area during fresh particle inputs (Gulf of Castellammare, southwestern Mediterranean), Biogeosciences, 16, 4307–4320, https://doi.org/10.5194/bg-16-4307-2019, 2019.

Provoost, P., Braeckman, U., Van Gansbeke, D., Moodley, L., Soetaert, K., Middelburg, J. J., and Vanaverbeke, J.: Modelling benthic oxygen consumption and benthic-pelagic coupling at a shallow station in the southern North Sea, Estuar. Coast. Shelf Sci., 120, 1–11, https://doi.org/10.1016/j.ecss.2013.01.008, 2013.

Teeling, H., Fuchs, B. M., Becher, D., Klockow, C., Gardebrecht, A., Bennke, C. M., Kassabgy, M., Huang, S., Mann, A. J., Waldmann, J., Weber, M., Klindworth, A., Otto, A., Lange, J., Bernhardt, J., Reinsch, C., Hecker, M., Peplies, J., Bockelmann, F. D., Callies, U., Gerdts, G., Wichels, A., Wiltshire, K. H., Glöckner, F. O., Schweder, T., and Amann, R.: Substrate-controlled succession of marine bacterioplankton populations induced by a phytoplankton bloom, Science, 336, 608–611, https://doi.org/10.1126/science.1218344, 2012.

Wrede, A., Dannheim, J., Gutow, L., and Brey, T.: Who really matters: influence of German Bight key bioturbators on biogeochemical cycling and sediment turnover, J. Exp. Mar. Bio. Ecol., 488, 92–101, https://doi.org/10.1016/j.jembe.2017.01.001, 2017.

---

## Author Response (AR3)

**Review** of the manuscript *Depositional controls and budget of organic carbon burial in fine-grained sediments of the North Sea – the Helgoland Mud Area as a natural laboratory*, by Müller et al, submitted to **Biogeosciences** (egusphere-2024-1632).

**Manuscript overview**

Given before, this is the second review round for this particular reviewer

**Review overview**

Following the initial presented manuscript (which had been reviewed before) the authors have made textual changes, as suggested. Some of my comments were is direct opposition to previous comments by other reviewers, so I accept the choices made by the authors to keep all/most reviewers happy. I do have some comments on the revised version though:

I know the sedimentation rates do not include seasonality, but my point was more on interannual variability. As the sedimentation rate estimates are based on longer timescales I would expect a shift in values as long term trends in e.g. SPM or DOC loads from the Elbe become visible. This hinders comparison to values obtained decades ago, for instance, if the trends are significant (which was my point for the comparison to de Haas et al (1997), this is nearly 30 years ago). This caveat should be included here, as marine trends have become stronger due to climate change in this region. Not necessarily for SPM, DIC or DOC loads from the Elbe (there is no obvious trend yet in the data from Patsch (2024), which is worth mentioning), but in general I think this point is worth including in the text. The authors have not addressed this point in the revised text.

Author's response: First, we would like to thank the anonymous reviewer #3 for the constructive feedback on our manuscript.
It is common to compare sedimentation rates, also those derived in different years, within the biogeosciences community, and we agree that it is often justified to compare those rates. However, our study does not focus on the interannual comparison (which requires knowledge about the spatial variability), but rather on the comprehensive understanding of the spatial variability itself in the study area. As stated in the manuscript (Chapter 5.1) we cannot assess the changes over time within our study. An assessment on the decadal variability of sedimentation rates would be interesting, but due to the small-scale heterogeneity that we demonstrate here, and the very low data density within the published studies from other sites it is not possible to evaluate a systematic shift with any statistical significance. Changes in sedimentation rates over decades, driven by e.g., SPM load in the Elbe river, which were shown to translate into the sediments of the German Bight (Serna et al., 2010) can potentially be of great importance for the HMA. However, the reviewer correctly states in the comment that the SPM load of the Elbe river shows no obvious trend over time (Pätsch, 2024). Furthermore, as we stated in the study area description, a comprehensive understanding of the sedimentation rates within the HMA cannot be drawn from the existing literature and hence the precision of sedimentation rates needed to compare decadal variability with any significance is not given. In the example of the reviewer: the study by de Haas et al. (1997) could not calculate sedimentation rates using their own [210]Pb data. The authors therefore used a compilation of published sedimentation rates from different years (1969 to 1978), obtained with different

methods, spanning a wide range of sedimentation rates. This makes an interdecadal comparison insignificant and we therefore did not include this aspect in the manuscript.

8. I also understand the focus is not on biology here, yet a few references to relevant work such as Neumann et al (2013), Shojaei et al (2016), Shojaei et al (2021) and particularly Thatje & Gerdes (1997) and Wrede et al (2017, already referenced elsewhere) would not be amiss given the speculation later of benthic activity differences. True, the spatial scale is not the same, but Thatje & Gerdes do classify a large part of the HMA as 1 ecosystem type, which is relevant for the presented work when bioturbation is discussed, and Wrede et al (2017) have quantified bioturbation potential in the area. So mentioning this in the site description seems worthwhile to me.

Author's response: We thank the reviewer for providing additional studies on biology in the German Bight of the North Sea. We have added a statement and the references (Thatje and Gerdes, 1997; Neumann et al., 2013; Shojaei et al., 2016, 2021; Wrede et al., 2017) within the study area description.

27. A reference to Callies et al (2017) would not be amiss here, just for readers who wish to have a better overview of local residual current patterns.

Author's response: The reference has been added in the text.

Purely textual:

Line 91: remove "as a natural laboratory", as the sentence is grammatically wrong this way and it is repetitive with regard to line 95.

Author's response: Done.

Line 553: "this is *likely* a result of", as you cannot be sure of the cause.

Author's response: Done.

Line 651: "sites *are* exceptionally" or replace while with "with" in the line above.

Author's response: Done.

**Recommendation**

Accept after minor revision

**References**

Callies, U., Gaslikova, L., Kapitza, H. and Scharfe, M., 2017. German Bight residual current variability on a daily basis: principal components of multi-decadal barotropic simulations. *Geo-Marine Letters*, *37*, pp.151-162.

de Haas, H., Boer, W., and van Weering, T. C. E. (1997) Recent sedimentation and organic

carbon burial in a shelf sea: the North Sea, Mar. Geol., 144, 131–146,

https://doi.org/10.1016/S0025-3227(97)00082-0

Neumann, H., Reiss, H., Ehrich, S., Sell, A., Panten, K., Kloppmann, M., Wilhelms, I. and Kröncke, I., 2013. Benthos and demersal fish habitats in the German Exclusive Economic Zone (EEZ) of the North Sea. *Helgoland Marine Research*, *67*, pp.445-459.

Shojaei, M.G., Gutow, L., Dannheim, J., Schröder, A. and Brey, T., 2021. Long-term changes in ecological functioning of temperate shelf sea benthic communities. *Estuarine, Coastal and Shelf Science*, *249*, p.107097.

Shojaei, M.G., Gutow, L., Dannheim, J., Rachor, E., Schröder, A. and Brey, T., 2016. Common trends in German Bight benthic macrofaunal communities: Assessing temporal variability and the relative importance of environmental variables. *Journal of Sea Research*, *107*, pp.25-33.

Thatje, S. and Gerdes, D., 1997. The benthic macrofauna of the inner German Bight: present and past. *Archive of fishery and marine research, 45 (2)*, *1997*, pp.93-112.

Wrede, A., Dannheim, J., Gutow, L., and Brey, T.: Who really matters: influence of German

Bight key bioturbators on biogeochemical cycling and sediment turnover, J. Exp. Mar.

Bio. Ecol., 488, 92–101, https://doi.org/10.1016/j.jembe.2017.01.001, 2017.

References used in the author's responses

de Haas, H., Boer, W., and van Weering, T. C. E.: Recent sedimentation and organic carbon burial in a shelf sea: the North Sea, Mar. Geol., 144, 131–146, https://doi.org/10.1016/S0025-3227(97)00082-0, 1997.

Neumann, H., Reiss, H., Ehrich, S., Sell, A., Panten, K., Kloppmann, M., Wilhelms, I., and Kröncke, I.: Benthos and demersal fish habitats in the German Exclusive Economic Zone (EEZ) of the North Sea, Helgol. Mar. Res., 67, 445–459, https://doi.org/10.1007/s10152-012-0334-z, 2013.

Pätsch, J.: Daily loads of nutrients, total alkalinity, dissolved inorganic Carbon and Dissolved Organic Carbon of the European Continental Rivers for the Years 1977 – 2022, Hamburg University, 2024.

Serna, A., Pätsch, J., Dähnke, K., Wiesner, M. G., Hass, H. C., Zeiler, M., Hebbeln, D., and Emeis, K.-C.: History of anthropogenic nitrogen input to the German Bight/SE North Sea as reflected by nitrogen isotopes in surface sediments, sediment cores and hindcast models, Cont. Shelf Res., 30, 1626–1638, https://doi.org/10.1016/j.csr.2010.06.010, 2010.

Shojaei, M. G., Gutow, L., Dannheim, J., Rachor, E., Schröder, A., and Brey, T.: Common trends in German Bight benthic macrofaunal communities: Assessing temporal variability and the relative importance of environmental variables, J. Sea Res., 107, 25–33, https://doi.org/10.1016/j.seares.2015.11.002, 2016.

Shojaei, M. G., Gutow, L., Dannheim, J., Schröder, A., and Brey, T.: Long-term changes in ecological functioning of temperate shelf sea benthic communities, Estuar. Coast. Shelf Sci., 249, https://doi.org/10.1016/j.ecss.2020.107097, 2021.

Thatje, S. and Gerdes, D.: The benthic macrofauna of the inner German Bight: present and past, Arch. Fish. Mar. Res., 45, 93–112, 1997.

Wrede, A., Dannheim, J., Gutow, L., and Brey, T.: Who really matters: influence of German Bight key bioturbators on biogeochemical cycling and sediment turnover, J. Exp. Mar. Bio. Ecol., 488, 92–101, https://doi.org/10.1016/j.jembe.2017.01.001, 2017.